# Annual mass change of the world's glaciers from 1976 to 2024 by temporal downscaling of satellite data with in-situ observations

Inés Dussaillant[1], Romain Hugonnet[2], Matthias Huss[3,4,5], Etienne Berthier[6], Jacqueline Bannwart[1], Frank Paul[1], Michael Zemp[1]

[1] Department of Geography, University of Zurich, Switzerland
[2] University of Washington, Civil and Environmental Engineering, Seattle, WA, USA
[3] Laboratory of Hydraulics, Hydrology and Glaciology (VAW), ETH Zurich, Switzerland
[4] Swiss Federal Institute for Forest, Snow and Landscape Research (WSL), bâtiment ALPOLE, Sion, Switzerland
[5] Department of Geosciences, University of Fribourg, Fribourg, Switzerland
[6] LEGOS, Université de Toulouse, CNES, CNRS, IRD, UPS, Toulouse, France

*Correspondence to* : Inés Dussaillant (ines.dussaillant@geo.uzh.ch)

**Abstract.** Glaciers, distinct from the Greenland and Antarctic ice sheets, play a crucial role in Earth's climate system by affecting global sea levels, regional freshwater availability, nutrient and energy budgets, and local geohazards. Past assessments of regional to global glacier mass changes were limited in spatial coverage, temporal resolution, and/or temporal coverage. Here, we present a new observation-based dataset of glacier mass changes with global coverage and annual resolution from 1976 to 2024. We use geostatistical modelling for the temporal downscaling of decadal glacier-wide elevation change estimates derived from satellite and airborne geodetic data, with glaciological annual in-situ observations. In more detail, we spatially interpolate the annual mass-balance anomalies from sparse in-situ observations and calibrate them to glacier-wide long-term trends from elevation change observations, available for individual glaciers for varying time periods and with global glacier coverage from 2000 to 2019. We then extrapolate the results to yearly time series starting between 1915 and 1976 depending on the regional data availability and extending to 2024. The time series are calculated separately for each of the world's glaciers and then aggregated to gridded (0.5° latitude and longitude), regional and global estimates of annual glacier mass changes. Since 1976, global glaciers have lost $9179 \pm 621$ Gt ($187 \pm 20$ Gt per year) of water, contributing to $25.3 \pm 1.7$ mm ($0.5 \pm 0.2$ mm per year) to global mean sea-level rise. About 41% (~10 mm) of this loss occurred in the last decade, with 6% (~1.5 mm) occurring in 2023 alone, the record-breaking year of glacier loss. We review the strengths and limitations of our new dataset, validate and discuss related uncertainty estimates in a leave-one/block-out cross validation exercise, and compare our results to earlier assessments. The annual mass change time-series for individual glaciers and the derived global gridded annual mass change product are available from the World Glacier Monitoring Service (WGMS) at: https://doi.org/10.5904/wgms-mce-2025-02.

# 1 Introduction

Glacier monitoring has evolved fast since its beginning in the late 19th century, with in situ and remotely sensed techniques allowing to observe detailed changes in area, elevation, volume and mass at first only for single glaciers and recently entire regions and the world (Berthier et al., 2023; The GlaMBIE Team et al., 2025; Thomson et al., 2021; Zemp et al., 2015). In-situ, glaciological observations provide extremely valuable information on the annual-to-seasonal temporal variability of glacier changes, reflecting the impact of atmospheric conditions which can be correlated over several hundred kilometers (Braithwaite and Hughes, 2020; Cogley et al., 2011; Fernández and Somos-Valenzuela, 2022; Kaser et al., 2003; Oerlemans, 2001; Østrem and Brugman, 1991). At present, and thanks to the coordination of the World Glacier Monitoring Service (WGMS) and its global network of contributors, glaciological in-situ observations exist in nearly all glacierized regions of the Randolph Glacier Inventory (RGI, Pfeffer et al., 2014; RGI Consortium, 2017). Most glaciers have been continuously monitored for periods longer than 10 years, with some of the earliest observations reaching back until the early 20th century. While irreplaceable, one major limitation of the glaciological method lies in the logistical hurdles of maintaining continuous field campaigns. At present in-situ observations are limited to approximately 500 glaciers worldwide, representing less than 0.2% of the world's glaciers (WGMS, 2025). Secondly, it is challenging to represent the complex mass balance pattern with individual in-situ point measurements such that potential sampling biases can accumulate in time when interpolating to glacier wide estimates. For this reason, glaciological observations often require reanalysis and calibration with glacier elevation change rates obtained from high-resolution geodetic surveys (Thibert et al., 2008; Thibert and Vincent, 2009; Zemp et al., 2013).

The geodetic or digital elevation model (DEM) differencing method is powerful at providing glacier elevation change observations with high accuracy over large glacierized areas and long periods of time (multi-annual to decadal, Cogley et al., 2011). DEM differencing was initially applied to individual glaciers with DEMs derived from maps (Joerg and Zemp, 2014) and aerial photographs (Belart et al., 2019; Finsterwalder, 1954; Papasodoro et al., 2015; Thibert et al., 2008), but has now evolved to include data from airborne Lidar (Abermann et al., 2010; Echelmeyer et al., 1996) spaceborne altimetry (Jakob and Gourmelen, 2023; Menounos et al., 2024) and satellite derived DEMs from multiple sensors (Berthier et al., 2023; Toutin, 2001). Recent advances in post-processing techniques (Hugonnet et al., 2022; McNabb et al., 2019; Nuth and Kääb, 2011; Rolstad et al., 2009), supercomputing capabilities and automated processing pipelines (Girod et al., 2017; Rupnik et al., 2017; Shean et al., 2016) have further enhanced this methodology enabling its application over entire mountain ranges (Brun et al., 2017; Braun et al., 2019; Dussaillant et al., 2019; Shean et al., 2020) and recently, globally (Hugonnet et al., 2021). The major limitations of the geodetic method lie firstly on the relatively short period since corresponding spaceborne sensors operate (in general after 2000), sensor related issues (e.g., radar signal penetration into snow and ice) and, importantly, on the inability to capture the annual variability of glacier mass changes due to a low signal-to-noise ratio of the elevation changes and the high uncertainties of the volume-to-mass conversion for periods shorter than five years (Huss, 2013).

Glacier change observations using glaciological measurements and the geodetic method therefore complement each other by providing different types of information. Zemp et al. (2019) was the first study to combine the annual variability from the glaciological observations with the long-term trends of the geodetic method in order to estimate annual mass changes for all 19 RGI glacier regions from 1976-2016. Their global estimate was hampered by the limited geodetic observational sample available at the time of the study (only 9% of Earth's glaciers by number) resulting in high uncertainties. The Hugonnet et al.

(2021) global assessment now fills this observational gap by leveraging the repeated acquisitions and global coverage of the Advanced Spaceborne and Thermal Emission and Reflection (ASTER) satellite optical stereo images (Raup et al., 2000). Their assessment provides individual glacier elevation change rates for nearly all glaciers worldwide (97.4% of RGI inventoried glacier area) from 2000 to 2019.

In this study, we provide a global observation-based assessment to estimate annual glacier mass changes at a glacier-specific level by geostatistical modelling, feasible thanks to the now almost complete coverage with glacier elevation change observations in the latest version of the Fluctuations of Glaciers (FoG) database (WGMS, 2025). To achieve this we use glaciological observations from approximately 500 glaciers (0.2% of the world's glaciers) starting between 1915 and 1976 and glacier-wide geodetic observations from approximately 207.000 glaciers (96% of the world's glaciers covering a 96% of

world's glaciated surface) starting in the 1940s (Table 1 and Fig. 1). The DEM differencing observations used here include the multiple local and regional satellite and airborne geodetic glacier change assessments for 30.000 (14%) glaciers plus the 20-year estimates from Hugonnet et al. (2021) available for 205.000 (95%) glaciers in the FoG database. Building upon the methodological foundations laid out in Zemp et al. (2019, 2020), we further develop the approach to spatially interpolate the glaciological annual in-situ observations and calibrate them to the long-term trends derived from satellite and airborne geodetic

data elevation change observations. The time series are calculated separately for each of the world's glaciers using geostatistical modelling and then aggregated to regional and global estimates of annual glacier mass changes. Our results include glacier mass changes with annual temporal resolution for each individual glacier in the RGI (with starting date between the hydrological years 1915-1976 (see methods) depending on the RGI region), and a global observation-based assessment of annual glacier mass changes since the hydrological year 1976, made available as a global gridded product. Our dataset holds

great potential for contribution to internationally coordinated intercomparison exercises such as The GlaMBIE Team et al. (2025).

## 2. Data and Methods

### 2.1 Input Datasets

#### 2.1.1. Glacier inventories

We use the digital glacier outlines from the Randolph Glacier Inventory 6.0 (RGI Consortium, 2017) to spatially locate glaciers, attribute their area and assign the in situ mass balance observations to individual glaciers. RGI version 6.0 represents glacier areas near the beginning of the 21$^{st}$ century. Version 6.0 has been preferred to the more recent version 7.0 for two reasons: first, because glacier elevation change observations are available only for version 6.0 and second, for comparison with previous observation based global assessments, also using this version. RGI outlines are available through the RGI portal (DOI: 10.7265/N5-RGI-60) and the Global Land Ice Measurements from Space initiative, an initiative from the early 2000s to improve glacier inventories using satellite data (GLIMS, DOI: 10.7265/N5V98602). Due to the high number of incorrectly mapped glaciers in the RGI 6.0 Caucasus and Middle East region (region 12), the Hugonnet et al. (2021) geodetic observations were calculated using the latest GLIMS outlines available (Tielidze and Wheate, 2018). The latter inventory is also used in this study for consistency. For the Greenland Periphery (region 5), we excluded glaciers strongly connected to the ice sheet (RGI 6.0 connectivity level 2) similarly to Hugonnet et al. (2021). To spatially constrain glaciers within the world's climatic regions, we use the 19 first-order glacier regions as defined by the RGI and illustrated in Fig. 1. Glacier regions are implemented directly in the RGI dataset and are accessible via the same DOI. The location of the glacier and region outlines is represented in Fig. 1. The full regional hypsometric coverage is illustrated in grey bars in Fig. 1.

#### 2.1.2 Glacier elevation and mass change observations

We use the glacier-wide annual mass change observations from the glaciological method and multiannual trends of elevation changes derived from the geodetic method as available from the latest update of the Fluctuations of Glaciers database (FoG). These glacier change observations are collected by the WGMS in annual calls-for-data through a worldwide network of national correspondents and principal investigators. After integration of the new, homogenized and corrected observations, a new FoG database version is released. Individual glaciers with available observations are identified in the FoG database with a WGMS-Id. Updated versions of the FoG database can be accessed via the WGMS online portal (https://wgms.ch/data_databaseversions/). Results presented here use version WGMS (2025) accessible through: https://doi.org/10.5904/wgms-fog-2025-02. The almost complete observational coverage of the latest FoG database version WGMS (2025) is depicted in Fig. 1, with the geodetic sample covering 206.554 glaciers or 96% of the world's glaciers covering the glacier's entire elevation range. More specifically in this study we use local and regional satellite and airborne glacier-wide DEM differencing observations available for 29.529 glaciers (14% of the world's glaciers) plus the 20-year estimates from Hugonnet et al. (2021) available for 205.120 (95%) glaciers in the FoG database (only glacier-wide estimates from Hugonnet et al. (2021) calculated from elevation change grids covering more than 50% of the glacier area where ingested into the FoG). The key characteristics of glacier mass and elevation change observations are summarized in Table 1. For more details on the

specific input data, auxiliary data, retrieval algorithms and uncertainty estimation of the independent FoG glacier elevation and mass change observations please refer to WGMS (2025). More details on the glaciological method can be found in Østrem and Brugman (1991), Kaser et al. (2003) and Zemp et al. (2013, 2015). For the geodetic method and its error sources see WMO (2023) and about measuring glacier mass changes from space, see Berthier et al. (2023).

**Table 1: Key characteristics of glacier elevation and mass change observations used in this study as available from the Fluctuations of Glaciers database WGMS (2025)**

|  | Glacier elevation change | Glacier mass change |
| --- | --- | --- |
| **Symbol** | $\dfrac{dh}{dt}$ | $B_{glac}$ |
| **Method** | Geodetic method, i.e. DEM differencing | Glaciological method |
| **Platform** | In-situ, airborne, spaceborne | In-situ |
| **Spatial resolution** | Glacier-wide average from DEMs of meter to deca-meter pixel size | Glacier-wide average from interpolated point measurements |
| **Spatial coverage** | Worldwide (~207,000 glaciers, 96%) covering a 96% of world's glaciated area | Worldwide (~500 glaciers, 0.2%) |
| **Temporal resolution** | Multi-annual to decadal | Seasonal to annual |
| **Temporal coverage** | From 1940s (start dates vary by region) until present | From 1915-1976 (start dates vary by region) until present |
| **Unit** | meter (m a$^{-1}$) | meter water equivalent (m w.e. a$^{-1}$) |
| **Required uncertainty*** | 2 m decade$^{-1}$ | 0.2 m w.e. a$^{-1}$ |

*(GCOS, 2022)

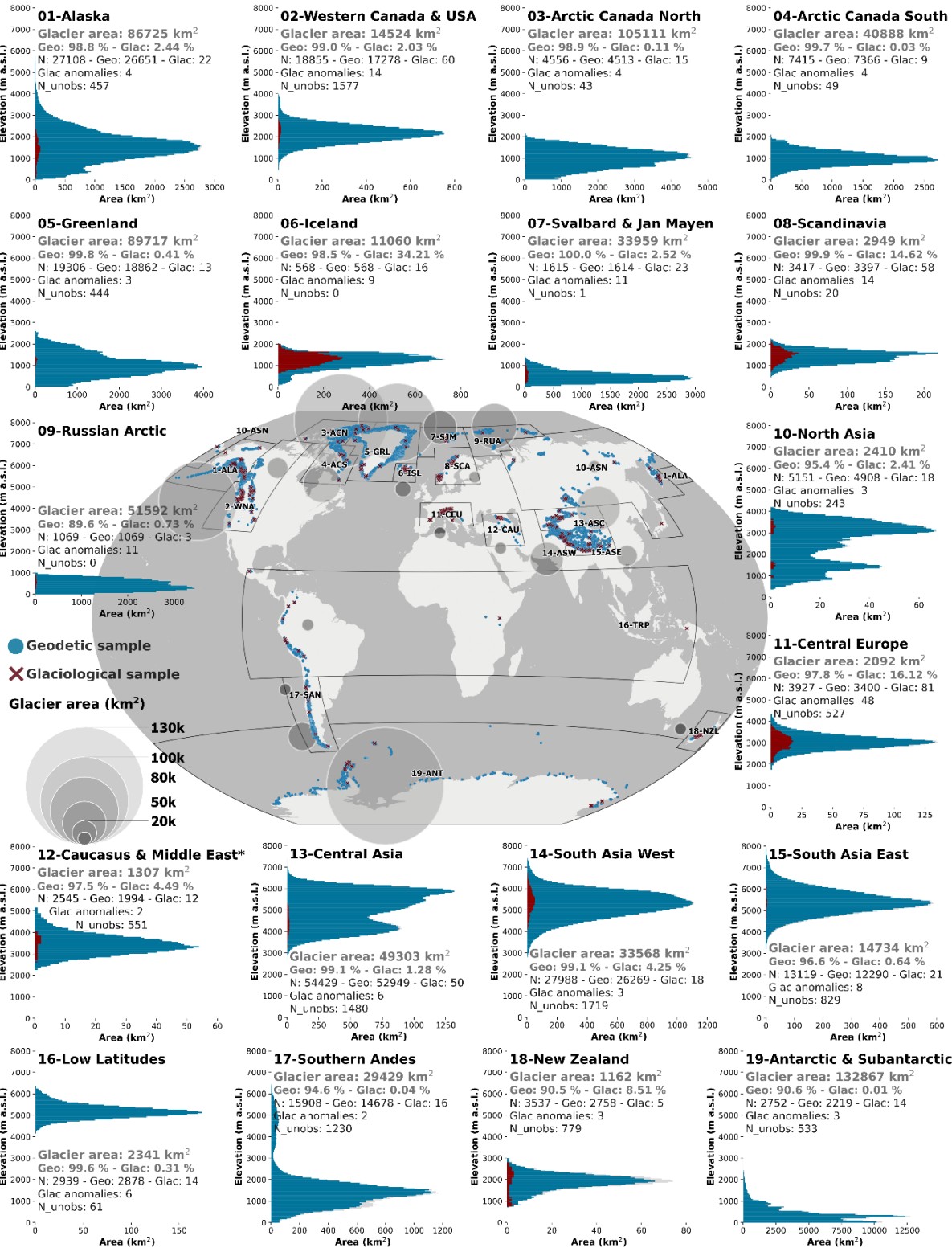

**Figure 1: Spatial and hypsometric coverage of glaciological and geodetic observations for each of the 19 first-order regions.** Glacier hypsometry from RGI 6.0 (grey) is overlaid (and almost hidden) with glacier hypsometry of the geodetic elevation changes (Geo, blue) and the glaciological (Glac, red) samples available from the FoG database WGMS (2025). Values for the glacier area and number (N) of glaciers are given for each region together with the respective percentage area covered, the number of observed and unobserved glaciers (N_unobs). Grey circles overlayed in the map represent the regional glaciated area. The number of used glacier anomalies is noted per region.
* Tielidze and Wheate (2018) inventory available from GLIMS
** Glaciers strongly connected to the ice sheet excluded

## 2.2 Methods

To prepare the data for the main processing, glacier-wide records with available mass and elevation change observations identified by a WGMS-Id are selected from the FoG database and related to their corresponding RGI outline identifier (RGIId) using a link-up table. We exclude geodetic records with survey periods shorter than five years in view of their large uncertainty for the volume-to-mass change conversion (Huss, 2013). For simplicity, throughout this work hydrological years are represented as the last year of the hydrological cycle (e.g. 1976) starting on the $1^{st}$ October to $30^{th}$ September in the Northern Hemisphere, and from $1^{st}$ April of the previous year (e.g. 1975) to $31^{st}$ March of the year (e.g. 1976) in the Southern Hemisphere. For the Low Latitudes region, we assume the hydrological year to be equal to the calendar year from $1^{st}$ January to $31^{st}$ December.

Our processing algorithm is summarized in three key steps, described in the following sections and in Fig. 2. First, focusing on a specific glacier in the RGI-6.0 inventory, we estimate the detrended temporal variability of annual mass change for the glacier, referred here as the glacier mean annual mass-balance anomaly, using the interannual variability of nearby glaciological time series (Section 2.2.1 and Fig. 2a). Secondly, we calibrate the mean annual mass-balance anomaly to the long-term trends from the geodetic sample available for the respective glacier (Section 2.2.2 and Fig. 2b). Third, we integrate all these calibrated time series into a single weighted average, producing a data-fused mean calibrated annual mass-change time series unique for every individual glacier (Section 2.2.3 and Fig. 2c). On a fourth step, the mean calibrated annual mass-change time series are integrated into larger regions (e.g. grid-point, glacier regions, global) to obtain a region-specific calibrated annual mass-change time series, accounting for spatial correlation of error sources. All given uncertainties in the tables, figures, main text and reported in the dataset files are at the one σ level (68% confidence interval), unless stated otherwise.

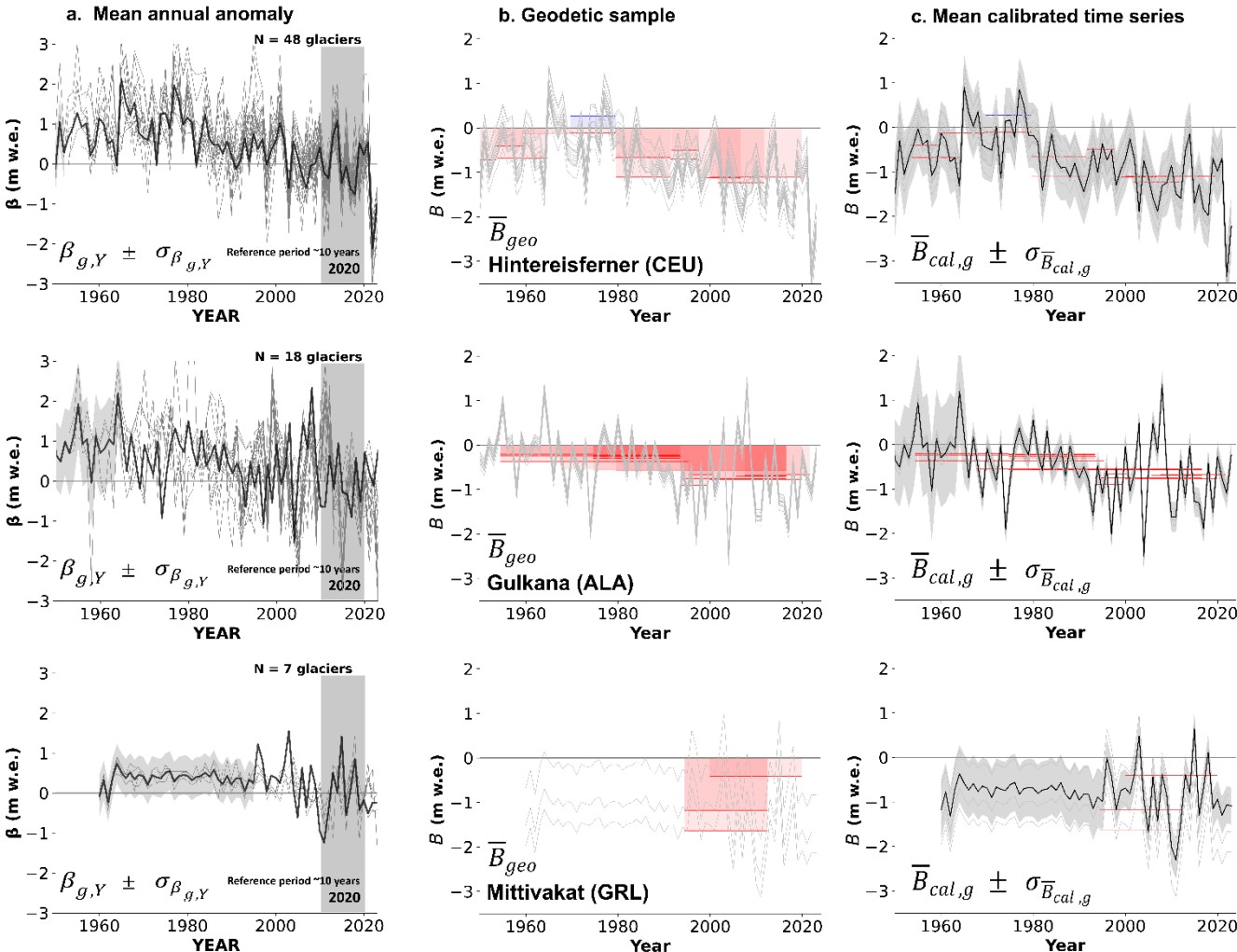

**Figure 2: Methodological steps illustrated at Hintereisferner (Central Europe, upper row), Gulkana (Alaska, middle row) and Mittivakat (Greenland Periphery, lower row).** (a) Mean annual mass-balance anomaly (black line) and uncertainty (grey).. Grey lines correspond to the spatially-selected individual glacier annual mass-balance anomalies. The mean annual mass-balance anomaly is calculated with Kriging spatial correlation function. (b) Calibration of the mean annual mass-balance anomaly over geodetic mass balance observations available for each glacier (red and blue lines for negative and positive change rates, respectively). Grey lines correspond to the individual calibrated time series for each geodetic mass change observation. (c) Mean calibrated annual mass-change time series (black line) and uncertainty (grey). We note that the full time series for Hintereisferner starts in 1915, and for Gulkana in 1946, for visualization purpose plots start 1950. Notations for the plots equations are defined in the text.

### 2.2.1 Computing the mean annual mass-balance anomaly from the neighbouring glaciological observations

Direct annual glaciological observations $B_a$ are reported to the FoG database with their relative uncertainties $\sigma_{B_a}$ in meters water equivalent (m w.e.) as:

$$B_a \pm \sigma_{B_a} \quad (1)$$

In cases where a glaciological series is missing an uncertainty estimate for a given year, we assume it to be equal to the mean of the annual uncertainty estimates within the series. In cases where a glaciological series has no uncertainty estimate for the entire period, we assume it to be equal to the mean annual uncertainty for all glaciological series from glaciers belonging to the same region.

From the individual glacier annual glaciological time series, we estimate an individual glacier annual mass-balance anomaly as the glaciological mass change value at year Y minus the mean mass change during the reference period from 2011 and 2020. Choosing this recent reference period allows exploiting a larger glaciological sample, thus obtaining a better representativeness of glacier temporal variabilities across all regions. We allow a threshold of at least 8 years of glaciological observations within the 10-year reference period to calculate a glacier annual mass-balance anomaly. This means that a glacier
needs to have at least 8 years of glaciological in-situ observations within the 10-year-reference period to calculate their annual mass-balance anomaly. At this step we remove low confidence glacier anomalies from the processing for not being representative of their regional mass balance variability (Table 2). We use a total of 158 individual glacier anomalies for the assessment.

$$\beta_Y = B_{a,Y} - \bar{B}_{a,2011-2020} \quad (2)$$


Starting from a given glacier g belonging to the RGI 6.0 glacier inventory (e.g. Hintereisferner, Gulkana, Mittivakat in Fig. 2), we use the sample of neighbouring glacier annual mass-balance anomalies to capture the annual temporal variability of its changes. The mean annual mass-balance anomaly of glacier g $(\overline{\beta_{g,Y}})$ is then calculated by kriging all glacier annual mass-balance anomalies located near the glacier (Fig. 2a).


$$\bar{\beta}_{g,Y} \pm \sigma_{\bar{\beta}_{g,Y}} = K(\beta_{i,Y}, \rho_{\beta,Y}(d)) \quad (3)$$

Where $\rho_{\beta,Y}(d)$ is the spatial correlation of the annual mass-balance anomaly (see further below) and $K$ the function applying ordinary kriging to $\beta_{i,Y}$ (i.e., kriging with constant mean, Pykrige python package, https://doi.org/10.5281/zenodo.3738604). When interpolating with kriging, the weight between each pair of glaciers is based solely on the distance between them, using a spatial correlation function which can be constrained from an empirical variogram. Additionally, the predicted kriging
uncertainty $\sigma_{\bar{\beta}_{g,Y}}$ grows with distance, from the measurement error of the inputs $\sigma_{B_a}$ at close distances from a measured glacier, to the signal variability (spread of $\beta_Y$) at distances far away from any measured glacier, where the prediction is more poorly constrained. Kriging is a core method of spatial statistics (Cressie, 2015), often coined as the 'best linear unbiased interpolator' due to its non-parametric nature and empirical variance minimization. It emerged in mining applications (Matheron, 1965), and has since become ubiquitous for spatial interpolation across many fields (Webster and Oliver, 2007).

In glaciology, kriging has been for instance used to spatially interpolate sparse ablation measurements (Hock and Jensen, 1999) or ice thickness measurements (Fischer, 2009). Recently, the rise of machine learning methods has extended kriging concept to any kind of dimension through Gaussian Processes (Rasmussen and Williams, 2006), which have also found applications in glaciology, from remote sensing time series interpolation (Hugonnet et al., 2021) to model error emulation (Edwards et al., 2021).


    In order to estimate the spatial correlation of the annual mass-balance anomaly $\rho_{\beta,Y}(d)$ to constrain the kriging, we sampled empirical variograms for both local-scale modelled annual mass-balance anomalies (Huss and Hock, 2015) and for observational 5-year anomalies (Hugonnet et al., 2021), the latter validating the spatial correlation patterns observed in the modeled estimates (Fig. 3). The 5-year anomalies are used only to validate annual anomalies. As climatic patterns driving

correlations in regional anomalies should have a size that is largely consistent in time, we expect 5-year anomalies to be spatially correlated at similar distances than annual anomalies but with a lesser amplitude due to the cancelling of positive and negative anomalies over time. We indeed identify that both anomalies have significant spatial correlation up to 5000 km, with a smaller amplitude for 5-year anomalies (Fig. 3). The empirical variograms were computed independently for each contiguous region (High Mountain Asia, North America and South America considered larger regions) and for each sub-period of 2000–

2020 (annual or 5–year) using up to 10,000 glaciers. In total, all variograms sample more than 10 billion pairwise differences between glacier annual mass-balance anomalies to obtain this average spatial correlation function. We then modelled the spatial correlation in annual mass-balance anomaly $\rho_{\beta,Y}$ as a sum of two exponential models:

$$\rho_{\beta,Y}(d) = s_1 e^{-\frac{3d}{r_1}} + s_2 e^{-\frac{3d}{r_2}} \; if \; d > 0, else \; 1 \qquad (4)$$

where $d$ is the distance between two glaciers, $s_1 = 0.37$, $s_2 = 0.59$ are the partial sills and $r_1 = 200 \; km$ and $r_2 = 5000 \; km$

are the correlation ranges. With this correlation function, we estimate that two directly neighboring glaciers have an annual mass-balance anomaly correlated at 96%, while correlated at 72% if separated by 60 km, at 51% by 250 km, at 32% by 1000 km and at 10% by 3000 km and less than 3% after 5000 km.

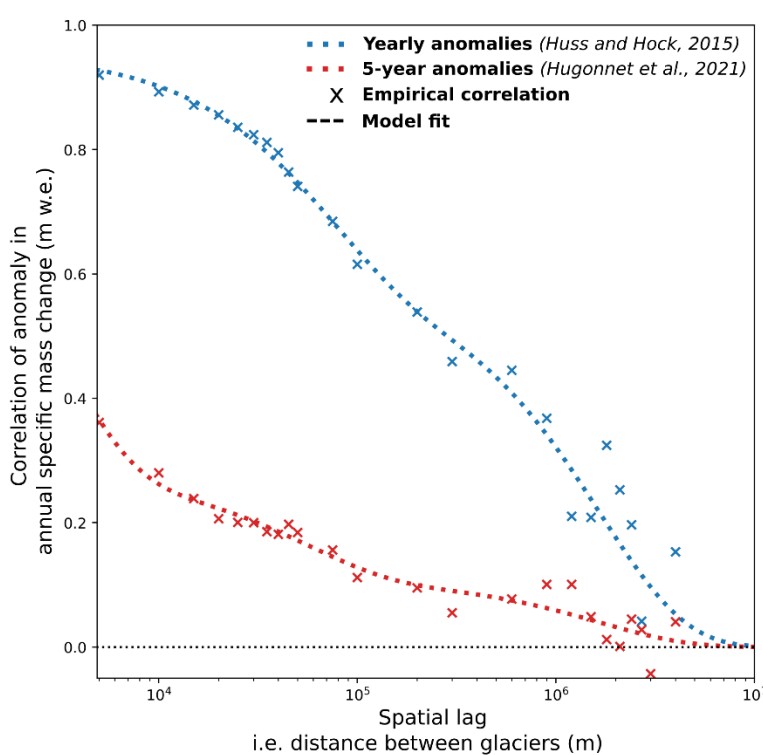

**Figure 3:** Spatial correlation of anomalies in specific mass change as used for kriging. The correlation is a global average estimated from regional empirical correlation sampled from all glaciers worldwide, for 5-year mass-balance anomalies with observational estimates based on surface elevation changes (Hugonnet et al., 2021) and for annual mb-anomalies for modelled estimates from Huss and Hock (2015). The former is used as validation and the latter for kriging in this study. For example, two directly neighboring glaciers have an annual mass-balance anomaly correlated at 96%, while correlated at 72% if separated by 60 km, at 51% by 250 km, at 32% by 1000 km and at 10% by 3000 km and less than 3% after 5000 km (Blue dotted line).

We note that, because kriging is a non-parametric interpolation method, its prediction primarily depends on the observations themselves, so uncertainties in the correlation function stemming from the modelled estimates of Huss and Hock (2015) have little influence on our results. Furthermore, because our correlations span multiple orders of magnitudes (from 10 km to 5000 km), the choice of functional form of the correlation has been shown to have minimal impact on the prediction (Hugonnet et al., 2022). To exemplify this, we compared kriging with inverse-distance weighting, a different interpolation method altogether, and found almost equal regional estimates as those are primarily driven by the input data. Differences between kriging and inverse-distance weighting only showed at the glacier-scale, where kriging allows to further refine anomalies and derive empirical uncertainties.

In under-sampled regions (Arctic Canada South, Russian Arctic, Asia South East, Asia South West and New Zealand), we added complementary glacier anomalies from neighbouring regions to calculate the mean glacier anomalies (Table 2, Fig. 4).

As a rule, all glaciers mean annual mass-balance anomalies should cover at least the period between the hydrological years from 1976 to 2024. For glaciers with mean annual mass-balance anomalies not arriving back to 1976, the best correlated glaciological series from neighbouring regions (i.e. climatically similar) are used to fill in the past years only (Table 2, grey sections in Fig. 4, see metadata file, decision supported by Zemp et al., 2019, 2020; Braithwaite and Hughes, 2020; Fernández and Somos-Valenzuela, 2022). To reduce the effect of possible climatic differences within the neighbouring regions, the amplitude of the complementary glacier anomalies is normalized to the amplitude of the mean glacier anomaly during the reference period.

**Table 2: Regional overview of (i) excluded glacier annual mass-balance anomalies, (ii) complementary glacier annual mass-balance anomalies added in under sampled regions to calculate the mean annual mass-balance anomalies, and (iii) complementary annual mass-balance glacier anomalies, normalized and used to fill up past years only to extend the series back in time until at least 1976.**

| RGI Region | (i) Excluded glacier annual mass-balance anomalies | (ii) Complementary glacier annual mass-balance anomalies | (iii) Complementary glacier annual mass-balance anomalies (Normalized) |
|---|---|---|---|
| **02-Western Canada US** | | | Taku (ALA) |
| **04-Arctic Canada South** | | ACN anomalies | |
| **05- Greenland Periphery** | | | Meighen and Devon Ice Caps (ACS) |
| **06-Iceland** | | | Storbreen, Aalfotbreen and Rembesdalskaaka (SCA) |
| **07-Svalbard** | | | Storglacieren (SCA) |
| **09-Russian Arctic** | | SJM anomalies | Storglacieren (SCA) |
| **10-North Asia** | Hamagury yuki (ASN) | | |
| **12-Caucasus Middle East** | | | Hintereisferner, Kesselwand (CEU) |
| **13-Central Asia** | Urumqi East and west branches (ASC) | | |
| **14-South Asia West** | | Ts. Tuyuksuyskiy (ASC) | |
| **15-South Asia East** | | Ts. Tuyuksuyskiy, Urumqi (ASC) | |
| **16-Low Latitudes** | Yanamarey | | Echaurren Norte (SAN-02) |
| **17-Southern Andes Patagonia** | All except Martial Este | | Echaurren Norte (SAN-02) |
| **17-Southern Andes Central** | All except Echaurren Norte | | Echaurren Norte (SAN-02) |
| **18-New Zealand** | | Martial Este (SAN-01) | Echaurren Norte (SAN-02) |
| **19-Antarctic & subantarctic** | Dry valley glaciers | | Echaurren Norte (SAN-02) |

### 2.2.2. Calibrating the mean annual mb anomaly on the glacier geodetic sample

Geodetic observations are reported to the FoG database with their relative uncertainties as glacier wide mean rates of elevation change ($\frac{dh}{dt}$) in meters during a specific period of record (PoR). Glaciers may contain multiple individual geodetic observations for different time periods depending on the dates of the DEMs used (red and blue bars in Fig. 2b). To obtain the geodetic mass

change rate, we convert elevation changes to glacier specific mass change rates in m w.e. by applying a density conversion factor $f_\rho \pm \sigma_{f_\rho} = 850 \pm 60 \ kg \ m^{-3}$ (Huss, 2013). At this step we exclude from the processing unpublished, and therefore low confidence, DEM differencing estimates available from the FoG.

$$\bar{B}_{geo,PoR} = \overline{dh}_{PoR} \cdot f_\rho \qquad (5)$$

To calculate the uncertainty in the geodetic mass balance rate in m w.e., we propagate the uncertainty in the elevation change rate $\overline{dh}_{PoR}$ and the uncertainty in the density conversion factor $\sigma_{f_\rho}$, considering that they are uncorrelated. We justify this by the fact that elevation change errors stem from instrument noise or spatiotemporal prediction, while density conversion errors stem from modelling errors and lack of knowledge on surface conditions, which are independent. We use reported uncertainties for elevation change rates, and we use $\sigma_{f_\rho} = 60 \ kg \ m^{-3}$ (Huss, 2013) for the uncertainty in the density conversion factor.

$$\sigma_{\bar{B}_{geo,PoR}} = \left| \bar{B}_{geo,PoR} \right| \sqrt{\left( \frac{\sigma_{\overline{dh}_{PoR}}}{\overline{dh}_{PoR}} \right)^2 + \left( \frac{\sigma_\rho}{f_\rho} \right)^2} \qquad (6)$$

In a data-fusion step, we calibrate the mean annual mass-balance anomaly (obtained from the glaciological sample) of glacier g to a given geodetic mass change rate k (i.e. geodetic sample) belonging to glacier g. We obtain a "k-calibrated" annual mass-change time series for every geodetic observation available for glacier g. The k-calibrated annual mass-change time series is then calculated as the sum of the geodetic mass change rate k and the mean annual mass-balance anomaly over the period of
record of the geodetic mass change rate k (grey lines in Fig. 2b). Due to the large uncertainties related to the volume-to-mass-change conversion factor over short periods of time (Huss, 2013), only geodetic observations longer than 5 years are considered for calibration.

$$B_{cal,k,Y} = \bar{B}_{geo,k,PoR} + \left( \bar{\beta}_{g,Y} - \bar{\beta}_{g,PoR} \right) \qquad (7)$$

The uncertainty in the k-calibrated annual mass-change time series is then propagated from the uncertainty in the individual
geodetic mass change rates and the uncertainty in the mean annual-anomaly. We consider these two uncertainties uncorrelated as they originate from independent measurements (remotely sensed and in situ) that do not share any similar error sources (including density conversion).

### 2.2.3. Combining the resulting time series into a mean calibrated annual mass-change time series

The mean calibrated annual mass-change of glacier g (Fig. 2c) is finally calculated as the weighted mean of the K available k-
calibrated annual mass-change time series (K is equal to the number of geodetic observations longer than 5 years available for the glacier). Two different weights are applied: First, a weighting relative to the uncertainty in the geodetic mass change $(W_{\sigma_{\bar{B}_{geo,k,PoR}}})$, where the k-calibrated annual mass-change time series are weighted to the inverse of that squared uncertainty. And second, a weight relative to the time lag of the k-calibrated annual mass-change time series $(W_t)$ in number of years to

the initial ($y_0$) and final ($y_1$) years of the period of record of the k-geodetic rate. Here, the point is to give more weight to a time series calibrated to a geodetic observation that is temporally close to the given geodetic period of record.

$$\bar{B}_{cal,g,Y} = \frac{\sum_{k=1}^{K} B_{cal,k,Y} \cdot W_{\sigma_{\bar{B}_{geo,k,PoR}}} \cdot W_t}{K} \qquad (8)$$

Where, $W_{\sigma_{\bar{B}_{geo,k,PoR}}} = \frac{1}{\sigma^2_{\bar{B}_{geo,k,PoR}}}$

and, $W_t = \left(\frac{1}{t_Y}\right)^p$ { $t = 1$, $y_0 < Y < y_1$ $t = Y - y$, $Y > y_1$ $t = y_0 - Y$, $Y < y$ , $p = 1$ to give temporally distant years a moderate weight.

For later error propagation, we separate the three sources of errors for elevation change, density conversion and anomaly calculation, given that those are largely independent between themselves, as previously justified, but have different scales of spatial correlations. To propagate uncertainties to the mean calibrated annual mass-change time series $\bar{B}_{cal,g,Y}$, we consider that each source is entirely correlated with itself during the averaging of the different calibrated series. This is justified because there is a single density conversion and anomaly estimation referring to a same period and glacier, which thus have the same errors. For elevation changes, there are sometimes multiple estimations referring to a same glacier, but that often share errors from similar instruments (e.g., ASTER) or estimation methods, and so we conservatively assume that their errors are fully correlated. Based on previous equations, we separate error propagation for elevation change, density conversion and anomaly calculation:

$$\bar{\sigma}_{dh,\bar{B}_{cal,g,Y}} = \frac{1}{N}\sum_{k=1}^{N} \sigma_{\overline{dh}_{k,PoR}} \cdot f_\rho \qquad (9)$$

$$\bar{\sigma}_{f_\rho,\bar{B}_{cal,g,Y}} = \frac{1}{N}\sum_{k=1}^{N} \sigma_{f_\rho} \cdot \overline{dh}_{k,PoR} \qquad (10)$$

$$\bar{\sigma}_{\beta,\bar{B}_{cal,g,Y}} = \frac{1}{N}\sum_{k=1}^{N} \sigma_{\bar{\beta}_{k,Y}} \qquad (11)$$

And the total uncertainty in the mean calibrated annual mass-change time series for a certain glacier is:

$$\sigma^2_{\bar{B}_{cal,g,Y}} = \bar{\sigma}^2_{dh,\bar{B}_{cal,g,Y}} + \bar{\sigma}^2_{f_\rho,\bar{B}_{cal,g,Y}} + \bar{\sigma}^2_{\beta,\bar{B}_{cal,g,Y}} \qquad (12)$$

### 2.2.4. Integrating glacier mass changes into larger regions

Every glacier with available geodetic observations has a mean calibrated annual mass-change (approximately 207.000 glaciers covering 96% of the world's glaciated area). The remaining unobserved glaciers ($g_{unobs}$) are assumed to behave as the regional mean of the observed sample. The individual glacier mean calibrated annual mass-change time series can be integrated into any larger region R containing multiple glaciers. The region-specific calibrated annual mass-change time series (i.e. grid cell or glacier region) $B_{cal}$ at year $Y$ and in region $R$ is calculated as the area weighted mean of the individual calibrated annual

mass-balance time series of the sample of observed glaciers belonging to region R (or the specific grid point R for the gridded product, Fig 6ii).

$$B_{cal,R,Y} = \frac{\sum_{g=1}^{N} \bar{B}_{cal,g,Y} \cdot A_g}{\left(\sum_{g=1}^{N} A_{g,Y}\right)} \qquad (13)$$

To derive the uncertainty in the region-specific calibrated mass-balance, we need to account for spatial correlations between the uncertainties of per-glacier mean calibrated annual mass-change. Indeed, our three error sources, elevation change, density conversion and mean annual mass-balance anomaly calculation, are significantly correlated spatially. For elevation change, we use the spatial correlation in elevation change error $\rho_{dh}(d)$ estimated in Hugonnet et al. (2021), as it is the main data source in the FoG database. These spatially correlated elevation errors are largely due to instrument noise and temporal interpolation to match an exact period of estimation.

$$\sigma^2_{dh,B_{cal,R,Y}} = \frac{1}{A_{tot}} \sum_{g_1} \sum_{g_2} \rho_{dh}\left(d_{g_1,g_2}\right) \cdot \sigma_{dh,\bar{B}_{cal,g_1,Y}} \cdot \sigma_{dh,\bar{B}_{cal,g_2,Y}} \cdot A_{g_1} \cdot A_{g_2} \qquad (14)$$

where $d_{g_1,g_2}$ is the distance between glaciers, $A_g$ is the area of glacier $g$, and $A_{tot} = \sum_g A_g$ is the regional glacier area.

For density conversion, we estimated a spatial correlation function of the uncertainty in the density conversion $\rho_{\sigma_{f\rho}}(d)$ by performing a similar variogram analysis as detailed for annual anomalies (Section 2.2.1) but instead applied to modelled estimates of annual density of volume change for all glaciers globally. These estimates were obtained by pairing a mass balance model (Huss and Hock, 2015) with a firn densification model (Huss, 2013), calibrated on geodetic mass balances (Hugonnet et al., 2021). We find a spatial correlation function of:

$$\rho_{\sigma_{f\rho}}(d) = s_1 e^{-\frac{3d}{r_1}} + s_2 e^{-\frac{3d}{r_2}} \text{ if } d > 0, else \text{ } 1 \qquad (15)$$

Where $d$ is the distance between two glaciers, $s_1 = 0.12$, $s_2 = 0.72$ are the partial sills and $r_1 = 200 \text{ } km$ and $r_2 = 5000 \text{ } km$ are the correlation ranges. These spatially correlated density errors are due to large local and regional variations in precipitation and firn densification, resulting in spatially correlated errors from the average value.

$$\sigma^2_{f\rho,B_{cal,R,Y}} = \frac{1}{A_{tot}} \sum_{g_1} \sum_{g_2} \rho_{\sigma_{f\rho}}\left(d_{g_1,g_2}\right) \cdot \sigma_{f\rho,\bar{B}_{cal,g_1,Y}} \cdot \sigma_{f\rho,\bar{B}_{cal,g_2,Y}} \cdot A_{g_1} \cdot A_{g_2} \qquad (16)$$

For mean annual mass-balance anomalies, we assume that errors to the real values are completely correlated at regional scales, and thus propagated as:

$$\sigma^2_{\beta,B_{cal,R,Y}} = \frac{1}{A_{tot}} \sum_g \sigma^2_{\beta,\bar{B}_{cal,g,Y}} \cdot A^2_g \qquad (17)$$

Finally, following the assumption that correlation between sources has a negligible impact compared to the spatial correlation of errors within the same source, we combine all sources of error propagated at the regional-scale as independent:

$$\sigma^2_{B_{cal,R,Y}} = \sigma^2_{dh,B_{cal,R,Y}} + \sigma^2_{f\rho,B_{cal,R,Y}} + \sigma^2_{\beta,B_{cal,R,Y}} \qquad (18)$$

The regional mass change in Gt of water is then obtained by multiplying the specific mass change by the region's (or grid point) glacierized area $S_R$, corrected to the year 2000 using the area change rates updated from Zemp et al. (2019). We propagate the uncertainty in the specific regional mass change, the uncertainty in the regional area (Paul et al., 2015) and the uncertainty in the area change assuming them uncorrelated. Errors in the area stem mostly from remote sensing delineation errors, while errors in area change stem from a lack of multi-temporal outlines to constrain area change. They are largely uncorrelated with error sources described above for elevation change, glaciological measurements and anomalies. However, elevation change estimates usually already consider errors in area at the scale of each glacier, so we might conservatively be double counting these.

$$\Delta M_{R,Y} = B_{R,Y} \cdot (S_R + \Delta S_{R,Y}) \qquad (19)$$

$$\sigma_{\Delta M_{R,Y}} = |\Delta M_{R,Y}| \sqrt{\left(\frac{\sigma_{B_{R,Y}}}{B_{R,Y}}\right)^2 + \left(\frac{\sigma_{S_R}}{S_R}\right)^2 + \left(\frac{\sigma_{\Delta S_{R,Y}}}{\Delta S_{R,Y}}\right)^2} \qquad (20)$$

Where $\frac{\sigma_S}{S} = 5\%$ (Paul et al., 2015) and $\frac{\sigma_{\Delta S_R}}{\Delta S_R}$ is updated from Zemp et al. (2019)..

The global annual (Y) and cumulative mass change (in Gt) and sea level equivalent for any given period of record is finally calculated as the sum of the regional mass change, assuming that the regional mass loss uncertainties are independent and uncorrelated between every region. To simplify the combination of annual values into long term trends or cumulative annual values, we assume the yearly uncertainty to be independent of other years. This is true for glaciological measurement, having an independent uncertainty estimation for each individual year of the time series, but not for the elevation change measurements, where uncertainties are correlated over the years of the survey period.

$$\Delta M_{Glob,Y} = \sum_{R=1}^{19} \Delta M_{R,Y} \qquad \text{and} \qquad \Delta M_{Glob,PoR} = \sum_{R=1}^{19} \Delta M_{R,PoR} \qquad (21)$$

$$\sigma_{\Delta M_{Glob,Y}} = \sqrt{\sum_{R=1}^{19} \left(\sigma_{\Delta M_{R,Y}}\right)^2} \quad \text{and} \quad \sigma_{\Delta M_{Glob,PoR}} = \sqrt{\sum_{R=1}^{19} \left(\sigma_{\Delta M_{R,PoR}}\right)^2} \qquad (22)$$

$$SLE_Y = \frac{\Delta M_{Glob,Y}}{S_{ocean}} \cdot 10^6 \pm \sigma_{SLE_Y} \qquad \text{and} \qquad SLE_{PoR} = \frac{\Delta M_{Glob,PoR}}{S_{ocean}} \cdot 10^6 \pm \sigma_{SLE_{PoR}} \qquad (23)$$

$$\sigma_{SLE_Y} = \sqrt{\sigma_{\Delta M_{Glob,Y}}^2 + \sigma_{S_{ocean}}^2} \qquad \text{and} \qquad \sigma_{SLE_{PoR}} = \sqrt{\sigma_{\Delta M_{Glob,PoR}}^2 + \sigma_{S_{ocean}}^2} \qquad (24)$$

Where $S_{ocean} = 362.5 \times 10^6 km^2$ and $\sigma_{S_{ocean}} = 0.1 \times 10^6 km^2$ (Cogley, 2012)

One strength of producing per-glacier mass change time series is the possibility to integrate them as an area-weighted mean at any given spatial resolution (i.e. regular grid, subregions, regions, basins, etc). In this study we integrate glacier mass changes

in three spatial resolutions: Regionally by the 19 RGI 1[st] order regions and globally to allow direct comparison with previous global observation-based assessments by Zemp et al. (2019) and Hugonnet et al. (2021). Further, taking advantage of the per-glacier annual time series, we generate a global gridded product of annual glacier mass changes for the Copernicus Climate Change Service (C3S, https://climate.copernicus.eu/) Climate Data Store (CDS). For consistency with other climate observation datasets (e.g. C3S), we provide glacier changes at a global regular grid of 0.5° latitude longitude. For temporal consistency within all regions, we extend the global time series only as far as the hydrological year 1976, in contrast to Zemp et al. (2019) who reached back to 1962. This adjustment is due to the absence of annual observations in the Southern Hemisphere regions prior to 1976 (evidenced in Zemp et al., 2019, Fig. 10). Regional time series start from the date of the first year of mass change records available for the region (see Table 5). Importantly, our fully operational approach allows producing yearly updates as soon as new glacier observations are ingested into the FoG database of the WGMS.

### 2.2.5. Methodological progress in data fusion of glaciological and geodetic data

The specific methodological improvements on data fusion of glaciological and geodetic data of the present assessment with respect to Zemp et al. (2019) are detailed in Table 3.

Table 3: Specific methodological improvements in data fusion of glaciological and geodetic data with respect to Zemp et al. (2019)

| | Zemp et al. (2019) | This study |
|---|---|---|
| **Extraction of the temporal variability from the glaciological sample** | | |
| **Selection strategy of glaciological time series** | By spatial clusters defined from 1[st] and 2[nd] order regions | Automatically selected with respect to the distance to the glacier. Manual removal of low confidence glaciological series from FoG-WGMS, 2025 (Table 2) |
| **Combination strategy of glaciological anomalies** | Variance decomposition model | Kriging spatial correlation function |
| **Selection of complementary glacier anomalies from neighboring regions for under sampled cases** | Selected by arbitrary expert choice | Best correlated with regional time series (Table 2) |
| **Normalized amplitude of the complementary glacier anomalies** | None | Normalized to the amplitude of the regional series during the reference period (Table 2) |
| **Calibration of the mean annual mass-balance anomaly on the glacier geodetic sample** | | |

| | | |
|---|---|---|
| **Selection strategy of DEM differencing observations** | All DEM differencing estimates from FoG-WGMS, 2018 used | Removal of low-confidence DEM differencing estimates from FoG WGMS (2025) |
| **Calibration strategy** | Regional anomaly calibrated to geodetic rates of available observations, averaged per glacier and combined with estimates for sample without observations | Glacier anomaly calibrated over every individual geodetic rate and then combined by weighting mean considering geodetic uncertainty and distance to geodetic survey period |
| **Uncertainty estimation and validation** | | |
| **Time-dependent uncertainty accounting for area-change rates** | Mean regional annual change rates | Mean regional annual change rates |
| **Spatial correlation of uncertainties** | Assuming no correlation for samples larger than 50 glaciers | Spatial correlation following an empirical function in density conversion error $(\rho_{f_\rho}(d)$, EQ 15) |
| **Validation of results** | Comparison with estimates in IPCC AR5 | Leave-one-out and leave-block-out cross-validation over independent reference and benchmark glacier time series |
| **Special cases** | | |
| **Special treatment in the Southern Andes region 17** | Considered as a whole | Subdivided into two RGI $2^{nd}$ order regions, due to the scarcity of glaciological time series in the Southern Andes region and to better account for the distinct climatic conditions of the Central and Patagonian Andes (Garreaud et al., 2013, 2009) |
| **Correction of Echaurren Norte glaciological time series** | None | Past period (1976-2000) normalized with respect to present period amplitude due to suspicious values. |

## 2.2.6. Description of the datasets

The datasets are described in Table 4. The main dataset, Dataset 1, corresponds to individual glacier annual mass change time series provided in .csv files by RGI first order regions. Glaciers are identified by their RGIId, centroid latitude and centroid longitude corresponding to the RGI60 glacier outline geometry. For Dataset 1, the start of the time series is region-dependent corresponds to the date of the first year of mass change records available for the region. We also provide for every region and on a glacier-by-glacier basis a .csv file with additional metadata information and a README. The second

Dataset 2 stands as a by-product from Dataset 1, it corresponds to an integration of the individual glacier timeseries on a global grid of 0.5° latitude longitude. We chose this resolution and a netCDF file format to make the glacier change product consistent and easily usable by other climate observation datasets (e.g. C3S). To ensure global completeness of annual

glacier mass changes, this dataset spans the period from 1976 to 2024.

**Table 4. Details of the annual glacier mass change output datasets**

| | Dataset 1 | Dataset 2 |
|---|---|---|
| **Dataset name** | **Individual glacier annual mass change time series** | **Global gridded annual glacier mass changes** |
| **Dataset access** | https://doi.org/10.5904/wgms-mce-2025-02 | https://doi.org/10.5904/wgms-mce-2025-02 |
| **File format** | Comma delimited file (.csv) One file per RGI 1st order Region | NetCDF (.nc) One file per hydrological year And one file with all hydrological years |
| **Data format** | **Columns:** **RGIId:** Glacier identifier from RGI60 (value of GLIMS_ID for Caucasus region 12). Unob_gla for unobserved glaciers. **CenLat:** Glacier centroid Latitude extracted from the RGI60 glacier outline geometry. (GLIMS outlines for Caucasus region 12) **CenLon:** Glacier centroid Longitude extracted from the RGI60 glacier outline geometry. (GLIMS outlines for Caucasus region 12) **YYYY:** Hydrological year named as last year of the hydrological cycle<br><br>**Variable:** glacier change time series and uncertainty in m w.e. | **Variables :** Glacier change (Gt) Glacier change uncertainty (Gt) Glacier change (m w.e.) Glacier change uncertainty (m w.e.) Glacier area per grid point ($km^2$)<br><br>**Dimensions:** Time Latitude Longitude<br><br>**Grid point naming convention**: Latitude, longitude at the middle of the grid point |
| **Data file names** | **Mean calibrated annual mass-change time series:** RRR_gla_MEAN-CAL-mass-change-series.csv<br><br>**Elevation change error (**one σ level**):** RRR_ gla_mean-cal-mass-change_DH-ERROR.csv<br><br>**Mean annual mass-balance anomaly error (**one σ level**):** RRR_gla_mean-cal-mass-change_ANOM-ERROR.csv | **Mean calibrated annual mass-change time series and total error:** global-gridded-annual-glacier-mass-change-YYYY.nc<br><br>One file per hydrological year YYYY, named as last year of the hydrological cycle |

**Density conversion error (**one σ level**):**
RRR_ gla_mean-cal-mass-change_RHO-
ERROR.csv

**Mean calibrated annual mass-change total error (**one σ level**):**
RRR_ gla_mean-cal-mass-change_TOTAL-
ERROR.csv

**Metadata:**
RRR_RGI-region-long-name_metadata.csv
README_metadata.txt

One file per RGI 1st order region, where RRR corresponds to the RGI-region code RGI-region-long-name to the RGI first order region long name

Exception RGI-region-long-name for Southern Andes related to the RGI second-order-region: SAN-01 as SouthernAndesPatagonia, and SAN-02 SouthernAndesCentral

| | | |
|---|---|---|
| **Spatial Coverage** | Global | Global |
| **Spatial resolution** | Individual glaciers | 0.5° (latitude - longitude) regular grid |
| **Temporal coverage** | Time series starting hydrological year is region dependent (see Table 5) until hydrological year 2024 | Hydrological years from 1976 to 2024 |
| **Temporal resolution** | Annual, hydrological year | Annual, hydrological year |
| **Conventions** | n/a | NetCDF convention CF-1.8 |
| **Projection** | Geographic Coordinate System: WGS 84 – EPSG:4326 | Geographic Coordinate System: WGS 84 – EPSG:4326 |
| **Projection identifier** | Centroid of glacier geometry (RGI60, GLIMS outlines for Caucasus, region 12) | Centroid of grid point |

# 4. Results

## 4.1 Global glacier mass changes

Our results provide revised global annual glacier mass changes extending back to the hydrological year 1976 (annual regional glacier mass changes further back in time depending on the region) at various spatial levels: per-glacier (Fig. 6i), regional and

on a global scale (Fig. 4, 5 and 6, Table 5). Globally, glaciers have lost 9179 ± 621 Gt of water (or 187 ± 20 Gt year$^{-1}$), contributing to 25.3 ± 1.7 mm of sea level rise since 1976. About 41% of the total loss, equivalent to 10 mm of sea level rise, occurred during the last decade from 2015-2024 only (Table 5). The strong acceleration of global glacier mass loss is evident over the last years of observations, where five (2019, 2020, 2022, 2023 and 2024) out of the last six years present the strongest global glacier mass loss ever recorded, exceeding 430 Gt/yr (or 1.2 mm/yr of sea level rise contribution), and the record year 2023 at 540 ± 69 Gt of water. During 2023 alone, glacier mass loss was about 80 Gt larger than any other year on record, corresponding to 6% of the total loss since 1975/76. In only this one year, glacier melt rose sea levels by 1.5 ± 0.2 mm.

Regionally, glacier mass losses in 2023 range from -0.3 m w.e. in the less impacted regions to up to -2.6 ± 0.6 m w.e. in Western North America. Strong glacier mass loss rates (lower than -1.0 m w.e.) were also reported in Central Europe, Alaska, New Zealand, the Southern Andes, Iceland and Scandinavia. In 2024, it was Scandinavian (-2.3 ± 0.4 m w.e.) and Svalbard (-1.6 ± 0.3 m w.e.) glaciers that suffered the most, followed by the Russian Arctic, Western North America, New Zealand, Southern Andes, South Asia East, Low Latitudes, Central Europe and Asia North with strong mass loss rates lower than -1.0 m w.e. Noteworthy years 2022, 2023 and 2024 are the first -and consecutive- years where all 19 glacier regions experienced average mass loss.

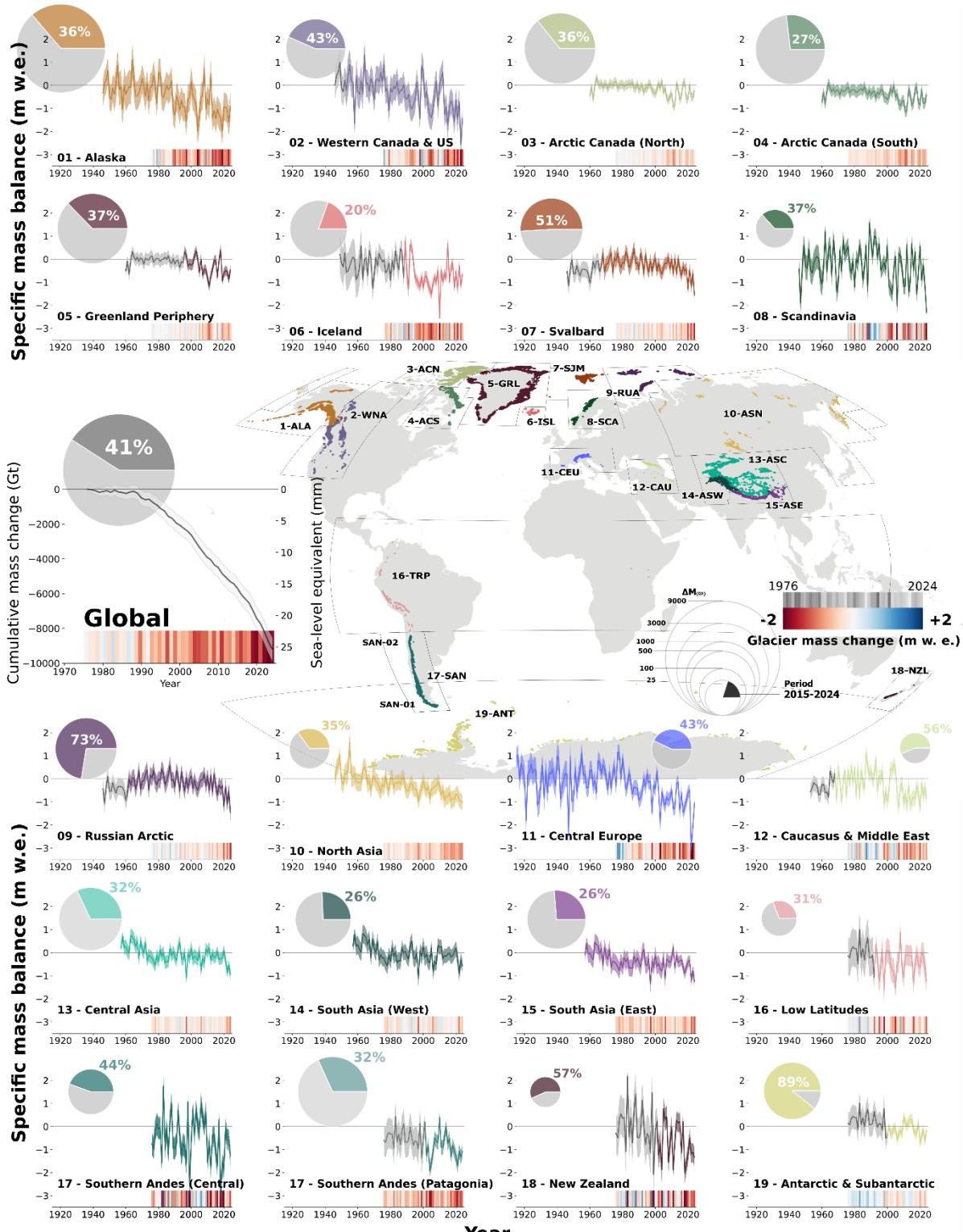

**Figure 4: Specific annual mass change time series for the 19 regions, with Southern Andes separated by 2nd order regions, with respective uncertainties.** The size of the circle in the pie charts represents the mass lost (in Gt) by region and the globe since the hydrological year 1976. Coloured sections represent the mass lost during the last decade only (2015-2024). Heat maps represent regional and global glacier mass changes in m w.e. for every hydrological year over the common period from 1976 to 2024. Global results from 1976 to 2024 are represented as cumulative mass changes in (Gt), the global heat map stretches from ±500 Gt. Regional time series start from the date of the first year of mass change records available. Grey sections show the periods where annual mass-balance anomalies from neighbouring regions are used to capture the temporal variability.

## 4.2 Regional glacier mass changes

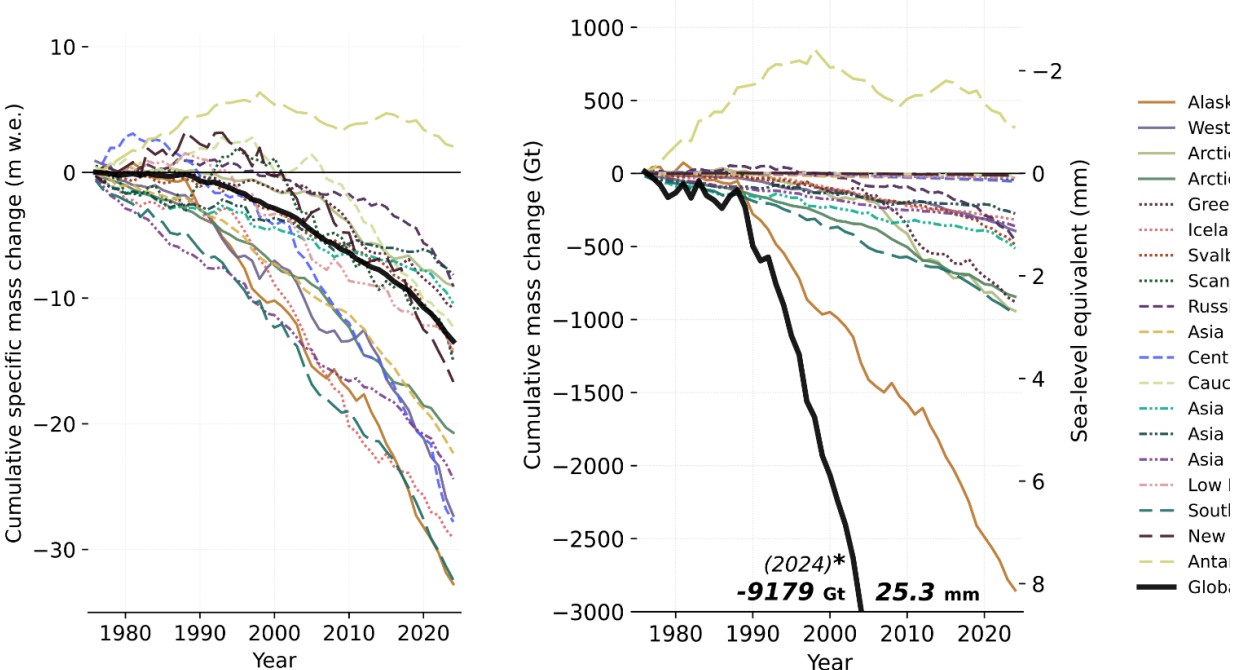

**Figure 5: Cumulative regional glacier mass changes from hydrological year 1976 to present for the 19 regions.** Specific mass changes in m w.e. indicate the mean height of the water layer lost over a given glacier surface, large negative values suggest regions where glaciers have suffered the most. By multiplying by the regional glacier area in km$^2$ we obtain the mass-change in Gt of water. Cumulative glacier mass changes in Gt correspond to the volume of water lost (1 km$^3$ w.e. = 1 Gt) and are related to the regional contributions to global mean sea-level rise in mm.

Since year 2000, all glacier regions have lost ice (Fig. 5). Alaska, Western Canada US, Svalbard, Russian Arctic, North Asia, Caucasus, Central Asia, Asia East, Southern Andes, New Zealand and Antarctica have experienced increased mass loss during the last decade (2015-2024) compared to the full period since 1976. Extremely negative regional decadal rates reaching lower than to -1.2 m w.e. yr$^{-1}$ are observed in Alaska, Western North America and Central Europe during the last decade. Record-breaking annual rates (more negative than -2.5 m w.e.) occurred in Western North America (2023), Iceland (2010), the Central Andes (1999, 2018) and Central Europe, the latter experiencing in 2022 the most negative regional mass balance ever recorded

at -3.1 m w.e. Mass losses larger than 100 Gt of water in a single year are present in only in Antarctica (2023) and Alaska

(twelve years in total, five of them during the last decade) with two regional yearly record mass loss of $176 \pm 52$ Gt and $163 \pm 48$ Gt in 2004 and 2019 respectively.

**Table 5: Annual rates of regional glacier mass changes in Gt and m w.e. from 1976 to 2024 and the last decade from 2015 to 2024. Mean area is calculated from the annual changes in area estimated with change rates updated from Zemp et al. (2019).**

| RGI Region Number-long name (code) | Mean area (km²) 1976-2024 | Start year | Mass change rate (Gt yr⁻¹) | | Mass change rate (m w.e. yr⁻¹) | |
|---|---|---|---|---|---|---|
| | | | 1976-2024 | Last decade 2015-2024 | 1976-2024 | Last decade 2015-2024 |
| 01-Alaska (ALA) | 90,507 | 1946 | -58.2 ± 55.5 | -103.5 ± 50.5 | -0.67 ± 0.61 | -1.26 ± 0.61 |
| 02-Western Canada US (WNA) | 15,045 | 1946 | -8.0 ± 8.5 | -17.1 ± 7.7 | -0.56 ± 0.56 | -1.26 ± 0.56 |
| 03-Arctic Canada North (CAN) | 105,119 | 1960 | -19.2 ± 20.7 | -33.7 ± 20.4 | -0.18 ± 0.20 | -0.33 ± 0.20 |
| 04-Arctic Canada South (ACS) | 40,892 | 1960 | -17.2 ± 10.1 | -22.7 ± 10.0 | -0.42 ± 0.25 | -0.56 ± 0.25 |
| 05-Greenland Periphery (GRL) | 89,978 | 1960 | -18.0 ± 21.8 | -32.9 ± 16.3 | -0.22 ± 0.24 | -0.44 ± 0.21 |
| 06-Iceland (ISL) | 11,080 | 1949 | -6.4 ± 3.8 | -6.2 ± 2.4 | -0.59 ± 0.33 | -0.60 ± 0.23 |
| 07-Svalbard (SJM) | 34,205 | 1946 | -10.0 ± 10.3 | -24.9 ± 9.4 | -0.30 ± 0.30 | -0.75 ± 0.28 |
| 08-Scandinavia (SCA) | 2,969 | 1946 | -0.9 ± 1.1 | -1.6 ± 1.0 | -0.31 ± 0.36 | -0.57 ± 0.35 |
| 09-Russian Arctic (RUA) | 51,965 | 1946 | -9.4 ± 18.1 | -33.3 ± 17.3 | -0.18 ± 0.35 | -0.66 ± 0.34 |
| 10-North Asia (ASN) | 2,529 | 1957 | -1.1 ± 0.9 | -1.9 ± 0.8 | -0.46 ± 0.34 | -0.82 ± 0.35 |
| 11-Central Europe (CEU) | 2,159 | 1915 | -1.1 ± 0.7 | -2.3 ± 0.5 | -0.57 ± 0.31 | -1.29 ± 0.28 |
| 12-Caucasus Middle East (CAU) | 1,317 | 1953 | -0.3 ± 0.4 | -0.8 ± 0.4 | -0.25 ± 0.32 | -0.70 ± 0.33 |
| 13-Central Asia (ASC) | 49,742 | 1957 | -10.5 ± 13.6 | -16.5 ± 11.7 | -0.21 ± 0.27 | -0.35 ± 0.25 |
| 14-South Asia West (ASW) | 33,849 | 1957 | -5.6 ± 11.6 | -7.1 ± 10.1 | -0.17 ± 0.34 | -0.22 ± 0.31 |
| 15-South Asia East (ASE) | 14,877 | 1957 | -7.4 ± 5.0 | -9.5 ± 3.5 | -0.50 ± 0.33 | -0.68 ± 0.25 |
| 16-Low Latitudes (TRP) | 2,335 | 1976 | -0.6 ± 1.5 | -0.9 ± 0.9 | -0.29 ± 0.61 | -0.51 ± 0.48 |
| 17-Southern Andes Patagonia (SA2) | 25,863 | 1976 | -17.5 ± 13.5 | -27.3 ± 7.9 | -0.68 ± 0.52 | -1.09 ± 0.31 |
| 17-Southern Andes Central (SA1) | 3,978 | | -2.1 ± 1.8 | -4.5 ± 1.8 | -0.53 ± 0.46 | -1.18 ± 0.45 |
| 18-New Zealand (NZL) | 989 | 1976 | -0.3 ± 0.7 | -0.8 ± 0.4 | -0.34 ± 0.69 | -0.98 ± 0.42 |
| 19-Antarctic Subantarctic (ANT) | 129,100 | 1976 | 6.4 ± 52.3 | -28.1 ± 28.1 | -0.04 ± 0.40 | -0.23 ± 0.23 |
| **GLOBAL** | **708,498** | 1976 | **-187.3 ± 19.8** | **-375.3 ± 15.6** | **-0.27 ± 0.12** | **-0.57 ± 0.10** |

## 5. Discussion

## 5.1 Multi-spatial dimensions of annual mass change series

The revised dataset claims several strengths, primarily related to the enhanced temporal resolution, providing glacier changes at annual resolution, and the multiple spatial dimensions for data integration from individual glacier to global. We remind here

that, by construction, nearby glaciers share a large fraction of the variance in mass balance variability and are thus not

independent. In a visualization example for selected years in Iceland, Fig. 6 depicts the multiple spatial dimensions from the dataset: Individual glacier annual time series (Fig. 6i) and annual time series aggregated into any larger scale region encompassing multiple glaciers, such as regular grid cells of 0.5-degree latitude longitude (e.g. Fig. 6ii or any user-specified resolution) and regional to global specific mass changes (Fig. 6iii and Fig. 4) and mass changes (Fig. 6iv, Fig. 10). This

versatility enables identification of individual years marked by significant glacier changes and the detection of zones with varying impacts. For instance, it allows us to pinpoint regions and subregions that were affected by specific annual climate variations (e.g. droughts, floods, heat waves, etc.), as well as those with a larger or smaller influence on the yearly contribution to hydrology and annual sea level rise.

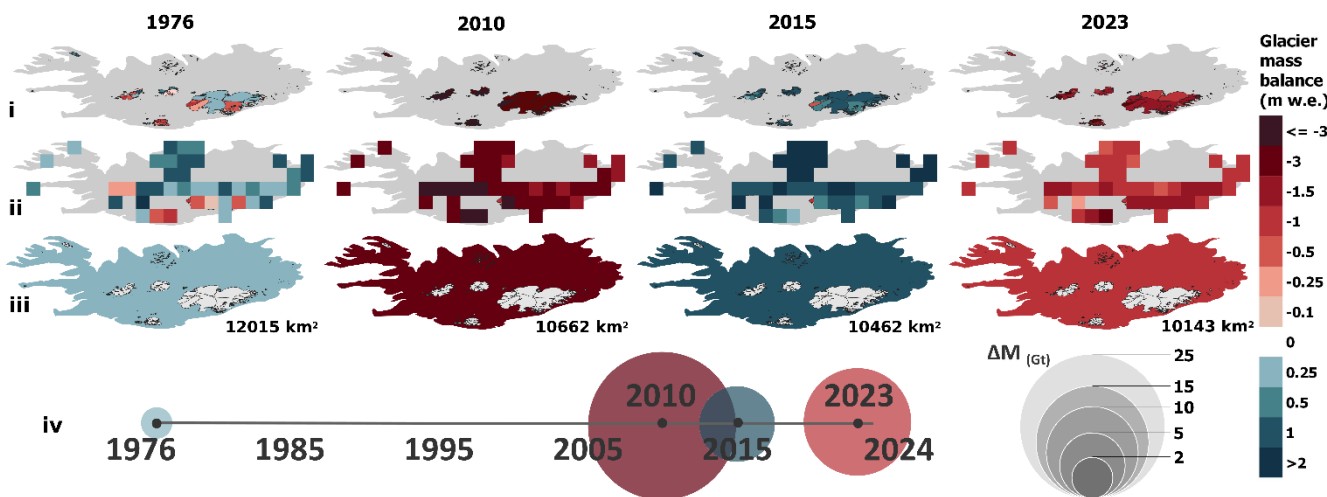

**Figure 6: Illustration of the multi spatial dimensions of the global annual mass change series, example for the Iceland region during selected years.** (i) Individual glacier annual mass change series, (ii) Gridded annual mass change series, (iii) Regional specific annual mass change in m w.e. and (vi) Regional mass change in Gt (note that all the previous dimensions i, ii and iii may also be represented in Gt).

Represented years in Fig. 6 are chosen arbitrarily: the initial hydrological year considered in the global assessment, the well-
470 known extremely negative mass change hydrological year 2010, attributed to the eruption of the Eyjafjallajökull volcano (Aðalgeirsdóttir et al., 2020; Belart et al., 2020; Möller et al., 2019), the most positive year of the time series in 2015 and the global extreme mass loss year 2023. This example allows us to illustrate the rather spatially homogeneous glacier mass loss of year 2010 with larger Icelandic glaciers all losing more than 2 m w.e. and some smaller glacier losing between 1.5 and 2 m w.e., whereas other years show larger variabilities between glaciers and grid points. This example demonstrates the richness
of the dataset for interpretation of glacier mass changes at different spatial scales and a deeper analysis of the spatial and temporal impact of known glaciological trends and anomalies like, for example, the Andes Megadrought (Dussaillant et al., 2019; Garreaud et al., 2017, 2020; Gillett et al., 2006) or the Karakoram anomaly (Farinotti et al., 2020; Gao et al., 2020;

Ougahi et al., 2022) at an unprecedented yearly temporal resolution. We note that the annual mass-balance anomalies are extracted from a handful of glaciers in each region and thus, in each region, individual glaciers share a large fraction of these variabilities. This limitation is, however, well evidenced by large uncertainties on under sampled regions and periods.

## 5.2 Leave-one-out cross validation

Due to the lack of independent measurements available to compare and validate our glacier change assessment, we applied a leave-one-out cross validation exercise over selected reference and benchmark glaciers. Reference and benchmark glaciers are selected considering their fluctuations to be mainly driven by climatic factors. They provide a reliable and well-documented sample of globally distributed long-term observation series, with more than 10 (benchmark) and 30 (reference) years of continuous and ongoing glaciological mass balance measurements. Noteworthily, glaciological time series can be subject to biases inherent to the glaciological method (e.g. Thibert et al., 2008) and are encouraged to be periodically reanalyzed and calibrated with long term trends derived from high resolution elevation change surveys (Zemp et al., 2013). To reduce the risk of validating over potentially erroneous "truths" we do not use all available glaciological time series in this experiment. We select a sample of 74 reference and benchmark glaciers for the leave one out cross validation and then repeat the analysis over a selection of 32 glaciers knowingly reanalyzed.

For each selected glacier, we compare the original 'reference' mass balance time series (reference Ba) as available from the FoG database, with the estimated leave-one-out mean calibrated annual mass-change time series (Leave-one-out Ba). The latter is obtained as described in the original methodology by calibrating the mean annual mass-balance anomaly of the glacier over its geodetic sample, only that this time we exclude the selected glacier anomaly from the processing. Reference and benchmark glaciers are usually highly monitored and contain multiple sources of geodetic observations for different time periods. However, more than 80% of the world's glaciers present only one source of geodetic observations, i.e., the 20-year elevation change rates from the Hugonnet et al. (2021). To make our validation exercise relevant for these under-sampled glaciers, we only consider for calibration the elevation change rates from Hugonnet et al. (2021), excluding other sources of geodetic observations. For each glacier, the Mean Error (ME) and the Standard deviation of the residual (S) between the reference and leave-one-out Ba are estimated as metrics to quantify potential systematic errors and the magnitude of random errors, respectively.

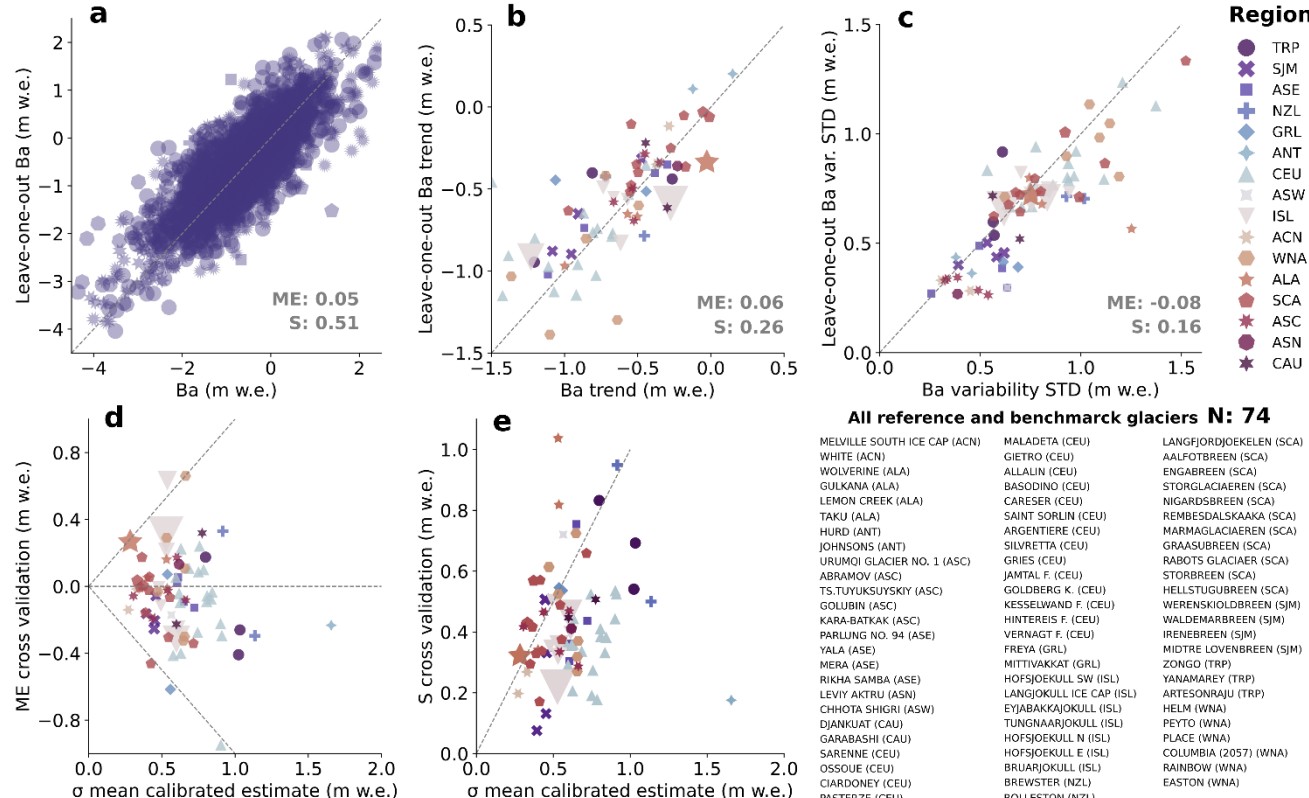


**Figure 7: Leave-one-out cross-validation results and statistics over 74 glaciological time series from reference and benchmark glaciers.** Mean error (ME) and standard deviation of the residuals (S) between the estimated **Leave-one-out Ba** cross validation time series and the 74 reference and benchmark glacier reference Ba time series. (a) Annual values from the estimated Leave-one-out Ba against reference/benchmark glaciers reference Ba. Every dot corresponds to a yearly observation. (b) Long term trends (period 1976-2024) from
the estimated Leave-one-out Ba time series against long term trends for the reference and benchmark glaciers reference Ba time series. (c) Amplitude of the annual variability measured as the time series variability STD (not to be confused with the standard deviation of the residuals noted S) for the period 1976-2024 from the estimated Leave-one-out Ba time series against the reference and benchmark glaciers reference Ba time series. (d) ME and (e) S of residuals for each reference and benchmark glacier against the estimated uncertainty of the mean calibrated annual mass-change estimate for the same glacier at one σ. In a, b, c and d, each value corresponds to one of the 74 reference
and benchmark glaciers used for cross-validation, symbols correspond to the glacier regions to which they belong. The size of the symbol is related to the area of the glacier.

Validation results are shown for all 74 selected reference and benchmark glaciers in Fig. 7. As verified by the low ME of 0.05 m w.e. for annual values and 0.06 m w.e. for long-term trends (Fig. 7a and 7b), we find no systematic error on the estimated Leave-one-out Ba. This means random errors are the largest error source, at 0.51 m w.e. for the annual values and 0.26 m w.e.
for long-term trends. Differences in annual values and the long-term trends may come from the different geodetic datasets used for reanalysis. Reference and benchmark glaciers, if reanalyzed, use high quality local elevation change observations for calibration, whereas here, our estimated leave-one-out Ba is calibrated over Hugonnet et al. (2021) elevation change only.

There is a slight underestimation of the glacier annual mass-balance variability amplitude, as shown by the slightly negative ME of -0.08 m w.e. between the Leave-one-out Ba and the reference Ba time series amplitude. This bias may come from a

smoothing of the mean annual mass-balance anomaly extreme values when averaging the nearby glaciological times series. This effect is clear for e.g. glacier Afoltbreen (Fig. 8g) where the extreme years in the reference mass balance series are less extreme in the leave-one-out series.

In general, regions with a rich glaciological sample, like Central Europe, Scandinavia, Svalbard, Iceland and Arctic Canada North perform well. Specifically, Hintereisferner located in the well-sampled Central Europe, present residual S as low as 0.19

m w.e. and a ME of -0.09 m w.e. The largest random errors are observed in glaciers Brewster in New Zealand (S: 0.94, ME: 0.32 m w.e.), Gulkana (S: 0.82, ME: -0.01 m w.e.) and Wolverine in Alaska (S: 1.03, ME: 0.16 m w.e.) and Zongo in the Low Latitudes (S: 0.83, ME: 0.17 m w.e.). The largest systematic errors are observed in glaciers Columbia in North America (S: 0.37, ME: 0.66 m w.e.), Tungnaarjokull in Iceland (S: 0.35, ME: 0.63 m w.e.), Mittivakkat in Greenland (S: 0.54, ME: -0.61 m w.e.) and Sarenne in Central Europe (S: 0.54, ME: -0.95 m w.e.) (Fig. 7 and Fig. 8 for selected reference and benchmark

glaciers). Still large differences may occur between reference and predicted values for individual years (outliers in Fig. 7a), also evident on the few annual values where the reference annual mass balance is not included within the uncertainties of the leave-one-out annual estimates (e.g. Mittivakkat, Fig. 8j and Djankuat, Fig. 8n).

Fig. 7d and 7e show the ME and S of residuals for each reference and benchmark glacier against the uncertainties of the mean calibrated annual mass-change time series $\sigma_{\bar{B}_{cal,g,Y}}$ calculated by our kriging predicting method for the same sample of glaciers.

In most cases, the predicted error is larger than both the leave-one-out cross-validation ME and S residuals for all glaciers. This means that the kriging method is cautious, predicting larger uncertainties than what is observed in the residuals. Thus confirming our uncertainties to be within a conservative range, with no significant systematic errors, and able to well capture random errors introduced by the prediction method.

The leave-one-out cross-validation shows that our method introduces only small systematic errors and has a per-glacier random

error that is conservative. However, most reference and benchmark glaciers are located in regions with a high density of mass balance observations able to compensate for their exclusion. Due to this, our leave-one-out exercise does not necessarily represent a typical glacier in a data scarce region. To explore the validity of our methodology in under sampled regions, where the mean glacier anomaly may come from distant glaciological time series, we present below a leave-block-out experiment over the full sample of reference and benchmark glaciers.

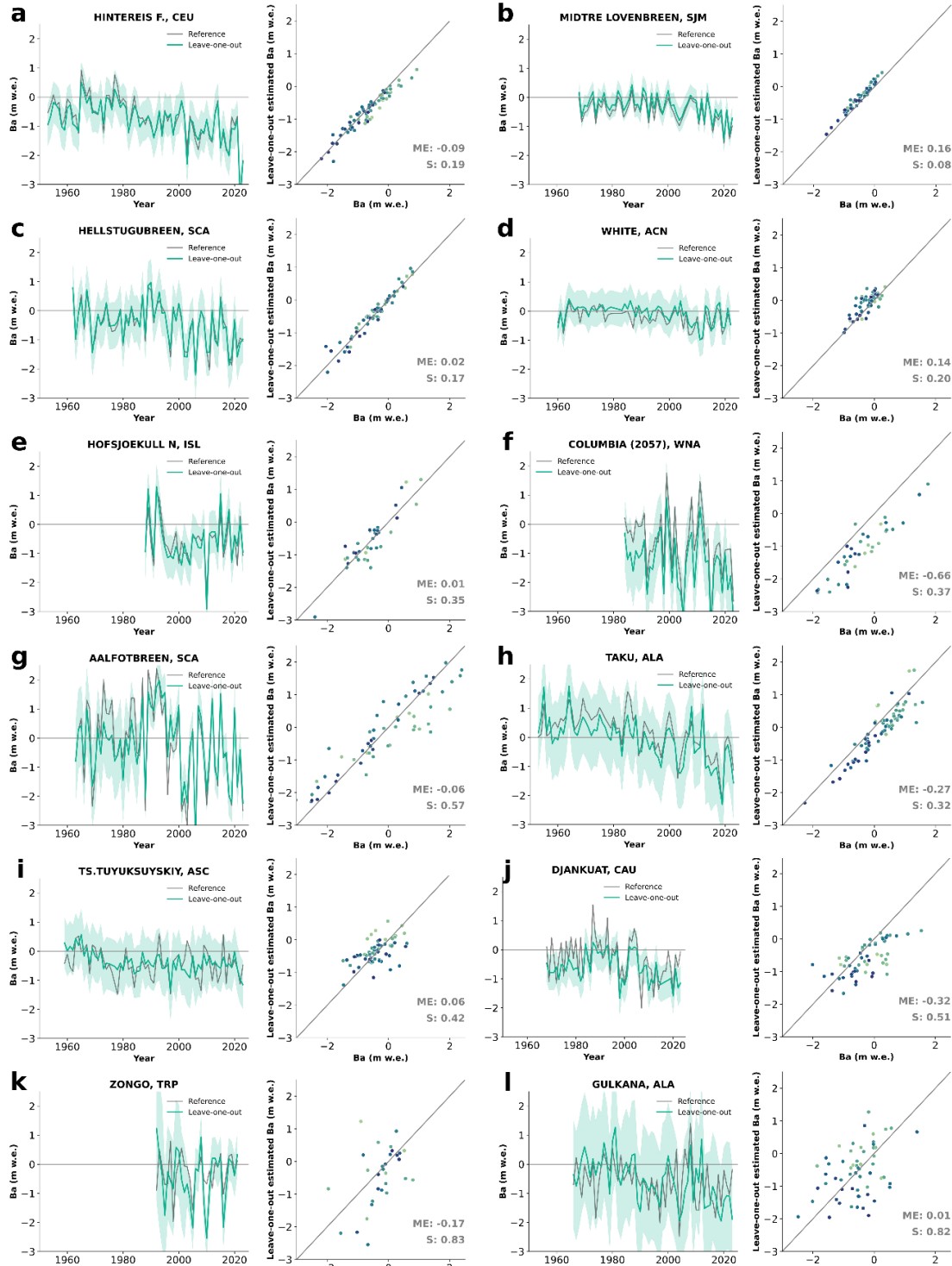

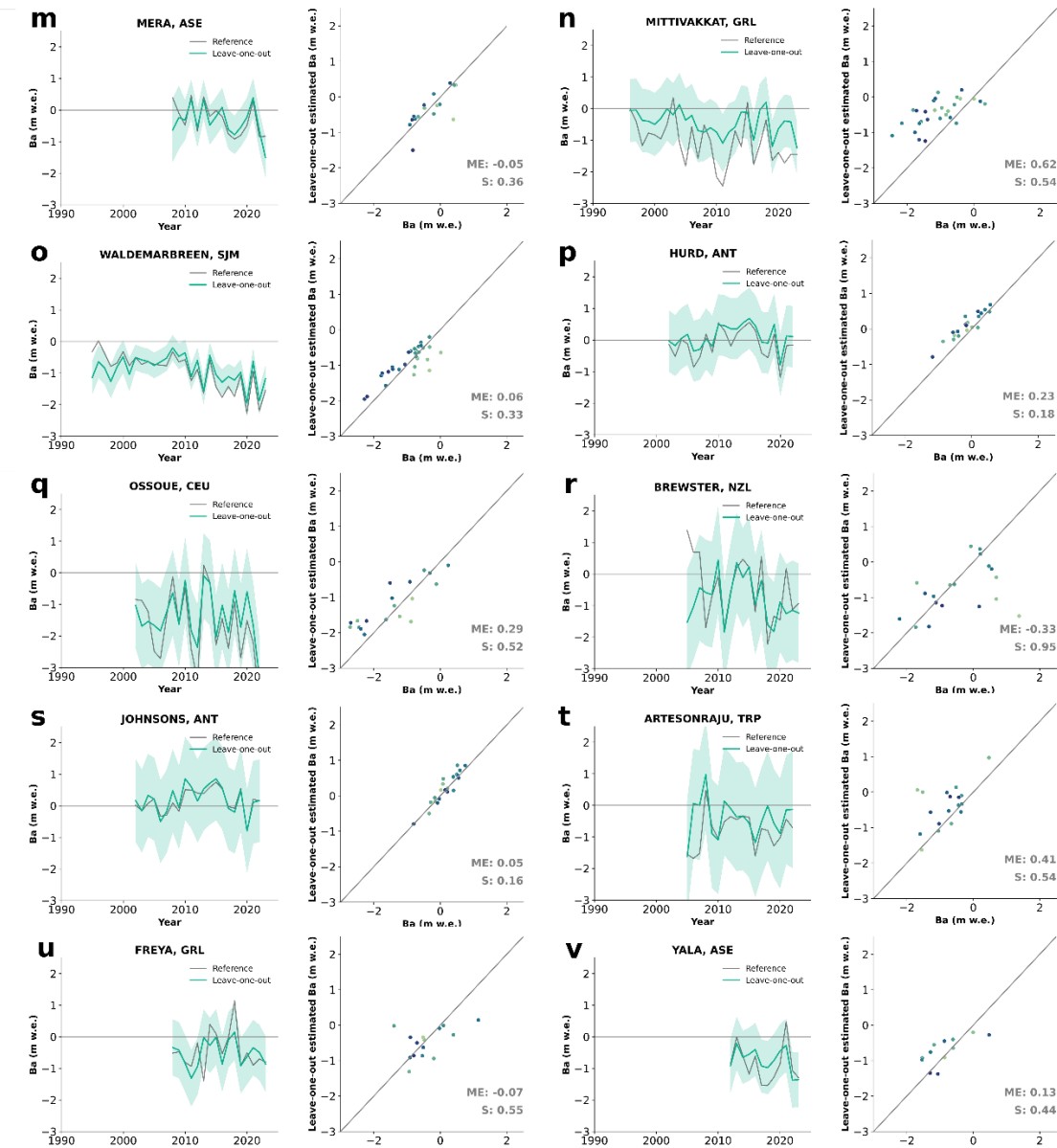

**Figure 8: Selected examples of the leave-one-out cross validation results on individual reference glaciers (a-l, more than 30 years of observations) and benchmark glaciers (m-v, more than 10 years of observations).** (Left plots) reference/benchmark glacier reference annual mass-balance time series (grey) and the leave-one-out estimated mean calibrated annual mass-change time series and uncertainties at 2σ (95% confidence interval, lightblue). (Right plot) Annual values from the estimated Leave-one-out Ba against reference/benchmark glaciers reference Ba. Every dot corresponds to a yearly observation.

## 5.3 Leave-block-out cross validation

The experiment is similar to the leave-one-out cross validation, only this time we remove from the processing not one but all the glacier anomalies surrounding the reference/benchmark glacier for increasing distances: first, we consider all anomalies further than 1 km (i.e. as in leave one out cross validation, only the glacier's mass-balance series is excluded), then, we remove the closest glacier's annual mass-balance anomalies at different distance thresholds of 60, 120, 250, 500, 800 and 1000 km. At every step a new mean glacier mass-balance anomaly is calculated for the reference/benchmark glacier from the evolving sample and calibrated to the 20-year elevation change rates from the Hugonnet et al. (2021). The ME and S of the residuals between the estimated leave-block-out mean calibrated annual mass-change time series (Leave-block-out Ba) and the reference annual mass-balance (reference Ba) is estimated at every distance threshold.

Considering all reference Ba values against the Leave-block-out Ba for the six different distances, systematic errors (ME) appear to stay stable between 0.06 and 0.08 m w.e. until 500 km, and then increase to 0.20 m w.e. above 500 km. Random errors (S) appear to increase gradually as the distance gets larger (Table 6).

**Table 6: Leave-block-out cross validation ME and S residuals for the yearly values (i.e. Fig. 7a) and for the amplitude of the annual variability during the 1976 – 2024 period (i.e. Fig. 7c) of the estimated Leave-block-out Ba calculated from samples at different distance thresholds.**

| Distance threshold of sample (km) | Yearly values | | Amplitude of the annual variability | |
|---|---|---|---|---|
| | ME | S | ME | S |
| > 1 | 0.05 | 0.52 | -0.08 | 0.16 |
| > 60 | 0.06 | 0.61 | -0.10 | 0.20 |
| > 120 | 0.07 | 0.65 | -0.10 | 0.22 |
| > 250 | 0.08 | 0.71 | -0.14 | 0.24 |
| > 500 | 0.19 | 0.92 | -0.01 | 0.39 |
| > 800 | 0.19 | 0.96 | -0.02 | 0.41 |
| > 1000 | 0.20 | 1.01 | 0.00 | 0.41 |

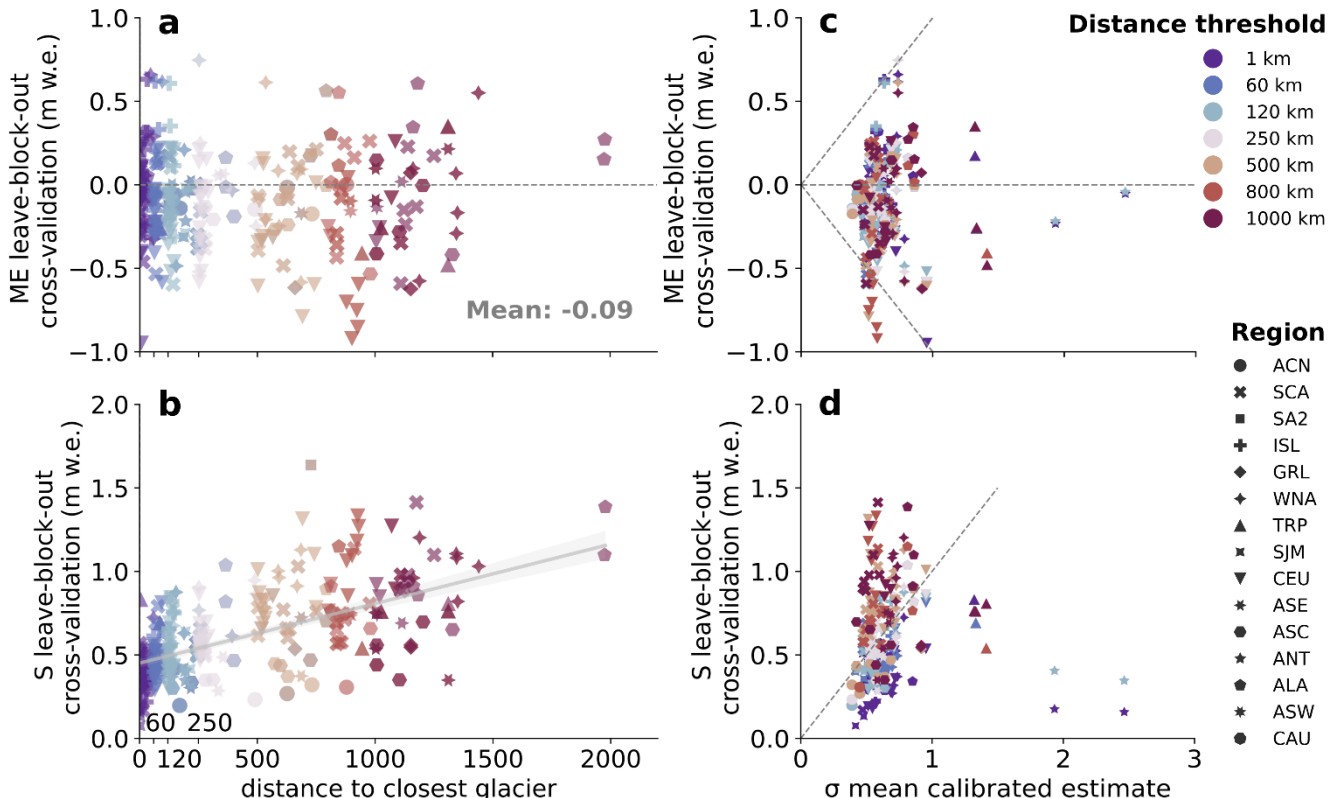

**Figure 9: Leave-block-out cross-validation results and statistics over 74 glaciological time series from reference and benchmark glaciers.** (a) Mean error of the leave-block-out residuals (ME) and (b) Standard deviation of the leave-block-out residuals (S) as a function of the distance to the closest glacier anomaly used to calculate the mean mb anomaly. (c) ME of residuals and (d) S of residuals at different distance thresholds against the estimated uncertainty of the mean calibrated annual mass-change estimate for the same glacier at $\sigma$. Symbols correspond to the region where the benchmark or reference glacier belongs.

Fig. 9 illustrates the leave-block-out glacier-wide residual results as a function of the distance to the closest glacier annual mass-balance anomaly considered. There is no apparent influence of the distance on systematic errors in the estimated glacier-wide Leave-block-out Ba justified by absence of trends in Fig. 9a. In these cases, the slight systematic errors will mostly depend on whether the reference series are reanalysed or not, and the quality of the elevation change used for calibration. As expected, random errors (residual S) increase as the glacier mean annual mass-balance anomaly is calculated from a more distant sample (Fig. 9b), from 0.5 m w.e. for nearby time series up to 1 m w.e. for series located farther than 2000 km. Importantly, in most cases systematic errors are not significant given our sample size, and random errors are well captured by the mean calibrated annual mass-change uncertainty at one σ independent of the distance of the sample (Fig. 9c and 9d). This means that our predicted uncertainties reflect the true variability in the residuals, and that our model is providing realistic confidence intervals for the mean annual mass-balance anomaly predictions. On average S starts to become larger than σ with distances to the closest glacier larger than 500km, but the large spread suggests this is coming from the randomness of the predictions.

We can conclude that the spatial extrapolation of distant glaciers does not introduce clear systematic errors but increases the random errors. However, these increased random errors are well predicted by our uncertainty assessment, showing larger uncertainties over glaciers in under-sampled regions. Both the leave-one-out and the leave-block-out cross validations show that our algorithm can capture the annual variability of individual glacier mass changes on glaciers not presenting glaciological time series (99% of the global glaciers) with realistic and conservative uncertainties.

## 5.4 Improvements with respect to earlier assessments

A general overview of the improvements with respect to previous assessments by Zemp et al. (2019) and Hugonnet et al. (2021) is provided in Table 7. Regional and global glacier mass change results for the three observation-based estimates are compared in Fig. 10 (in Gt).

**Table 7: General methodological ameliorations with respect to Zemp et al. (2019) and Hugonnet et al. (2021)**

| | Zemp et al. (2019) | Hugonnet et al. (2021) | This study |
|---|---|---|---|
| **Spatial coverage** | 9% | 97.4% of glacier area | 96% of glacier area |
| **Spatial resolution** | RGI Region | Individual glaciers RGI-6 Regular global grid (5, 10, 20 year resolution) | Individual glaciers RGI-6 Regular global grid (annual resolution) |
| **Temporal coverage** | 1960 – 2016 (with global annual resolution 1976 – 2016) | 2000 - 2020 | 1976 – 2024 Globally (regionally <1976, see Table 5) |
| **Temporal resolution** | annual | Pluri-annual (5,10 and 20 years) | annual |
| **Uncertainty sources for individual glaciers time series** | N/A (only regional time series) | **σ dh** **σ density** mean (Huss, 2013) | **σ dh** **σ density** mean (Huss, 2013) **σ glac. anomaly** |
| **Uncertainty sources regional time series** | **σ dh** and **σ glac** propagated assuming uncorrelation for samples larger than 50 glaciers **σ density** mean (Huss, 2013) **σ area change rate** from regional annual mean | **σ dh** propagation (Hugonnet et al, 2022) | **Empirical spatial correlation function** **σ dh** propagation (Hugonnet et al, 2022) **σ glac. anomaly** propagation **σ density** propagation (empirical function in density conversion error, $\rho_{f_\rho}(d)$, EQ 15) **σ area change rate** from regional annual mean |
| **Annual updates** | No update | Not updated yet | Yearly updates since 2022 (C3S CDS) |

| **Estimates capture subaqueous mass loss** | No, assumes all mass loss occurs above sea level | No, assumes all mass loss occurs above sea level | No, assumes all mass loss occurs above sea level |
|---|---|---|---|
| **Glacier inventory Caucasus region 12** | RGI-6 | GLIMS Tielidze and Wheate (2018) | GLIMS Tielidze and Wheate (2018) |

Broadly, our new approach effectively corrects the negative bias in the long-term trends observed in Zemp et al. (2019) thanks

to the integration of the glacier elevation changes from Hugonnet et al. (2021), which translates into a significant reduction in both regional and global uncertainties, largely noticeable for the more recent years (Fig. 10, Table 8). Globally, glacier mass change rates between 1976 and 2016 are less negative (-148 ± 20 Gt yr$^{-1}$) than previously estimated by Zemp et al. (2019) (-204 ± 45 Gt yr$^{-1}$, Table 7). The resulting 6069 ± 783 Gt of cumulative mean mass loss for the period is 35% smaller than the 9290 ± 7698 Gt predicted loss by Zemp et al. (2019) for the same period. The differences mainly come from the Southern

Andes, Alaska, Russian Arctic, Antarctic and Subantarctic Islands, Greenland Periphery (Table 8) – all regions with limited geodetic coverage in the previous assessment. Both regionally and globally, the years after 2000 are well aligned to the Hugonnet et al. (2021) trends as consequence of the calibration to their geodetic trends with global coverage.

Alaska, Arctic Canada North, Western North America, the Russian Arctic, Caucasus, Low Latitudes, New Zealand and the Southern Andes exhibit less-negative trends compared to Zemp et al. (2019). In contrast, Asia North and Arctic Canada South

trends are more negative in the past. Overall, the regional trends agree well with Hugonnet et al. (2021) trends during the overlapping period 2000-2019 (Table 8). Deviations of more than 5 Gt yr$^{-1}$ are found in Alaska, Greenland Periphery, and Antarctic and Subantarctic Islands.


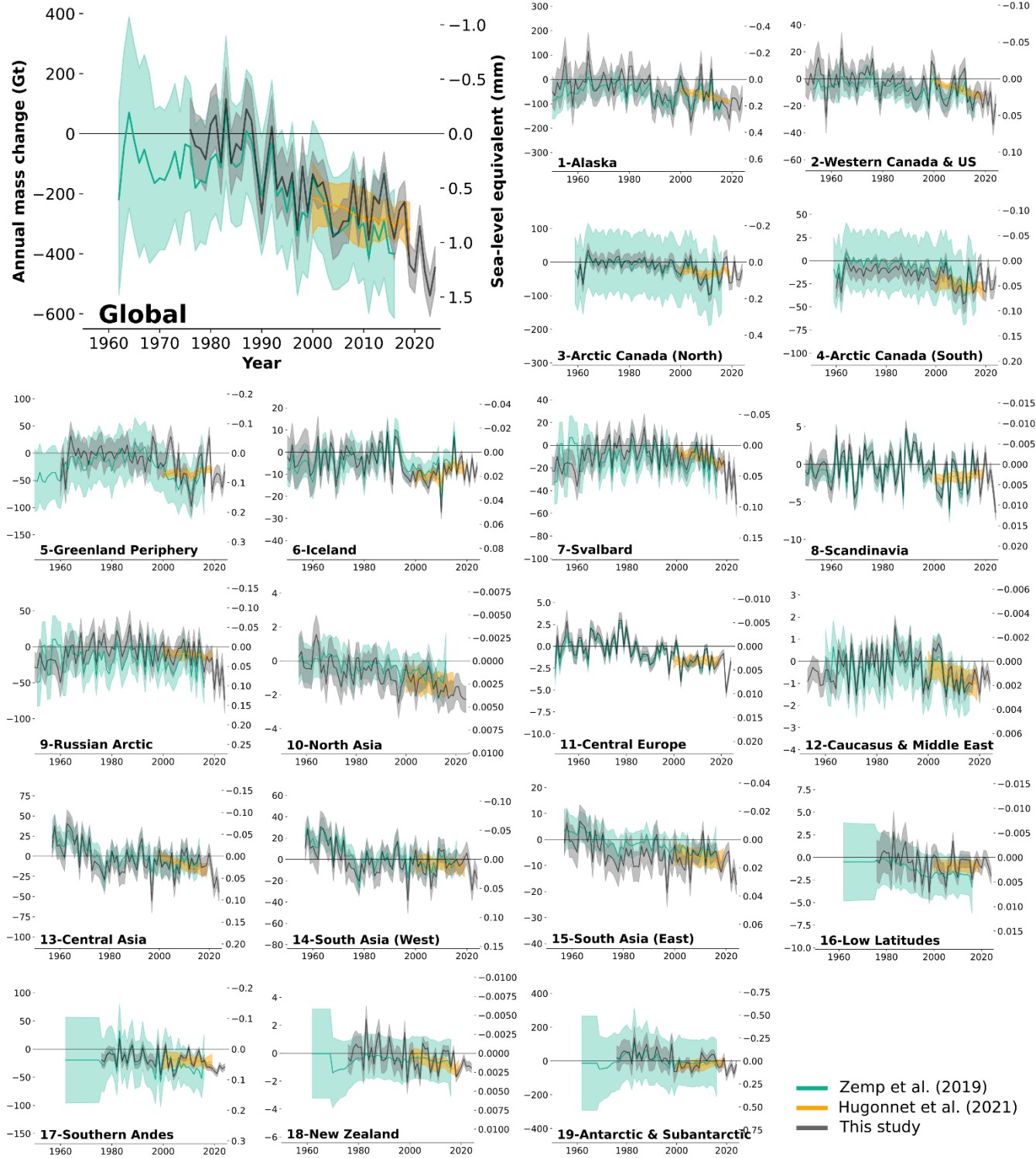

**Figure 10: Annual glacier mass change (Gt) from this study compared with results from Zemp et al. (2019) and Hugonnet et al. (2021).**

Differences in Greenland Periphery, Alaska and the Russian Arctic can be explained by the removal of unpublished geodetic observations from our processing chain, which may have biased the trends in Zemp et al. (2019). The new mean calibrated annual mass-change time series in these regions rely only on the 2000-2019 Hugonnet et al. (2021) estimates as calibration reference (and the 1985-2000 Huber et al. (2020) for the Greenland Periphery past period). The Antarctic and Subantarctic Islands is a region prone to large uncertainties in all studies. Because all estimates agree within (large) uncertainties no difference can be interpreted. The Antarctic and Subantarctic Islands region presents no glaciological or geodetic measurements before the year 2000. The signal for the annual variability before 2000 is driven purely by the very-distant Echaurren Norte (Central Andes) normalized time series, and past trends are only calibrated over the 2000-2019 Hugonnet et al. (2021) series, which are very likely biasing the period before 2000 towards more positive values. There is insufficient evidence to support the glacier mass gain observed before 2000 in both our assessment and the Zemp et al. (2019). We assume results in the Antarctic and Subantarctic Islands region to be very likely biased by the lack of observations, and therefore highly uncertain, as reflected in our large error bars. However, we still include them to provide global glacier mass changes back to 1976.

Most regions display increased interannual variabilities when compared to both previous studies. The large uncertainties associated with the ASTER DEM time series in Hugonnet et al. (2021) prevent the detection of variability at short time scales (typically less than 3 years, Fig.10). Similarly in Zemp et al. (2019), the variance decomposition model (Eckert et al., 2011; Krzywinski and Altman, 2014) employed to extract the temporal mass change variability for each region has shown to contribute to a slight smoothing of the annual amplitude signal (Zemp et al., 2020). Our approach allows us to better represent the interannual variability at the individual glacier level, supported by the Leave-one-out cross validation exercise and the effect is inherited to the regional and global level.

Importantly, our assessment reduces global and regional uncertainties compared to Zemp et al. (2019) (Table 8). This reduction is achieved firstly, because our approach benefits from the almost complete-observational sample from Hugonnet et al. (2021), which highly reduces uncertainties after year 2000. Secondly, multiple algorithm improvements, such as the spatial search of glacier anomalies, the normalization of neighbouring glacier time series with original regional amplitudes. Notably, considering error propagation using empirical functions of spatial correlation for the different sources of error provides more realistic uncertainties than previous work. All these enhancements are further supported by the leave-one-out-cross validation over reference glaciers, confirming that our uncertainties remain still on the conservative side. Compared with Hugonnet et al. (2021), long term trends agree within uncertainties, but our estimated errors are consistently larger throughout all regions. This comes from the propagation of the mean mass-change anomaly uncertainties, which is the largest source of error in our assessment, and a source that was not considered in the Hugonnet et al. (2021) annual results. These differences are likely

explained by different assumptions taken in the temporal propagation of uncertainties, rather than actual errors arising from the estimations.

Like Zemp et al. (2019), one of the largest strengths of our method is the ability to provide glacier mass changes back in time up to 1976 at a global level and even further back in regions with longer observational records (e.g. 1915 for Central Europe). Thanks to geostatistical modelling for temporal downscaling, we can capture now the annual temporal variability of glacier changes for each of the world's glaciers and then aggregate them into revised regional and global estimates of annual glacier mass changes.

**Table 8: Annual rates of regional glacier mass change for the period 1976-2016 from this study compared with results from Zemp et al. (2019) for the same period and annual rates of regional glacier mass change for the period 2000-2019 from this study compared with results from Hugonnet et al. (2021) for the same period.** Mean area is calculated from the annual changes in area estimated with change rates updated from Zemp et al. (2019).

| RGI Region | Mean area (km²) (1976-2016) | Mass change (Gt year$^{-1}$) | | Mean area (km²) (2000-2019) | Mass change (Gt year$^{-1}$) | |
|---|---|---|---|---|---|---|
| | | This study (1976-2016) | Zemp et al. (2019) (1976-2016) | | This study (2000-2019) | Hugonnet et al. (2021) (2000-2019) |
| 01 Alaska | 91,980 | -49.4 ± 53.6 | -65.2 ± 41.0 | 86,299 | -72.3 ± 50.3 | -66.7 ± 10.9 |
| 02 Western Canada US | 15,303 | -6.3 ± 8.7 | -9.6 ± 10.0 | 14,311 | -9.1 ± 8.1 | -7.6 ± 1.7 |
| 03 Arctic Canada North | 105,387 | -17.5 ± 20.8 | -24.5 ± 92.0 | 104,354 | -32.0 ± 20.6 | -30.5 ± 4.8 |
| 04 Arctic Canada South | 40,996 | -16.6 ± 10.1 | -9.6 ± 32.0 | 40,594 | -22.2 ± 10.1 | -26.5 ± 4.3 |
| 05 Greenland Periphery | 92,409 | -15.0 ± 22.9 | -24.9 ± 46.0 | 83,464 | -27.8 ± 18.8 | -35.5 ± 5.8 |
| 06 Iceland | 11,219 | -6.3 ± 4.0 | -3.9 ± 6.0 | 10,681 | -9.4 ± 2.6 | -9.4 ± 1.4 |
| 07 Svalbard | 34,372 | -6.8 ± 10.0 | -13.2 ± 12.0 | 33,767 | -10.9 ± 8.5 | -10.5 ± 1.7 |
| 08 Scandinavia | 2,997 | -0.7 ± 1.1 | -0.9 ± 1.0 | 2,889 | -1.7 ± 1.0 | -1.7 ± 0.4 |
| 09 Russian Arctic | 52,219 | -4.3 ± 16.9 | -20.0 ± 24.0 | 51,300 | -10.6 ± 16.0 | -10.4 ± 1.9 |
| 10 North Asia | 2,565 | -1.0 ± 0.9 | -0.5 ± 1.0 | 2,426 | -1.3 ± 0.8 | -1.3 ± 0.4 |
| 11 Central Europe | 2,227 | -0.8 ± 0.7 | -1.0 ± 1.0 | 1,966 | -1.6 ± 0.6 | -1.7 ± 0.4 |
| 12 Caucasus Middle East | 1,342 | -0.3 ± 0.4 | -0.5 ± 1.0 | 1,248 | -0.6 ± 0.4 | -0.7 ± 0.2 |
| 13 Central Asia | 50,237 | -8.9 ± 13.4 | -5.9 ± 15.0 | 48,327 | -7.9 ± 12.6 | -9.6 ± 2.1 |
| 14 South Asia West | 34,173 | -5.2 ± 11.3 | -1.9 ± 10.0 | 32,924 | -4.0 ± 10.5 | -4.6 ± 1.7 |
| 15 South Asia East | 15,034 | -6.9 ± 5.0 | -3.2 ± 5.0 | 14,428 | -6.7 ± 4.1 | -6.9 ± 1.4 |
| 16 Low Latitudes | 2,430 | -0.6 ± 1.6 | -1.5 ± 3.0 | 2,069 | -0.8 ± 1.0 | -0.9 ± 0.2 |
| 17 Southern Andes | 30,009 | -16.7 ± 10.3 | -22.8 ± 40.0 | 29,420 | -21.3 ± 6.4 | -20.7 ± 4.1 |
| 18 New Zealand | 1,017 | -0.2 ± 0.8 | -0.4 ± 2.0 | 909 | -0.5 ± 0.5 | -0.7 ± 0.2 |
| 19 Antarctic & subantarctic | 130,356 | 15.2 ± 56.3 | 5.5 ± 152.0 | 125,513 | -11.1 ± 28.9 | -20.9 ± 4.9 |
| **GLOBAL** | **716,272** | **-148.0 ± 20.1** | **-203.8 ± 45.4** | **686,889** | **-251.8 ± 15.8** | **-267 ± 8.0** |

## 5.5 Known limitations

### 5.5.1. Scarcity of the glaciological in-situ observations

The scarcity of glaciological data stands as the primary limitation in assessing the variability of glacier changes with our methodology. In sparsely observed regions like High Mountain Asia, the Southern Andes, Arctic Canada South, the Russian Arctic, Greenland and Antarctica, annual variations in glacier changes depend on limited and distant or neighbouring regions timeseries, which may not necessarily be representative of the local glacier annual variability. As a consequence, the annual glacier mass change time series exhibit high uncertainties in these regions, realistically estimated by our method.

We note a significant observational gap in the Southern Hemisphere where the glacier mass change variability before 2000 is driven by the single and very distant Echaurren Norte glacier. This results in a disproportionate importance at the global scale. The best way forward while still maintaining the independence and observation-based nature of the present assessment -crucial for calibration and validation of glacier models- is to bolster in-situ glacier monitoring programs in these regions. In the short term, efforts must be directed towards ensuring the continuity of glacier in situ monitoring in the Southern Hemisphere and possible correction of past long-term series.

### 5.5.2. Availability of past geodetic observations

The lack of geodetic observations for the period before 2000 is consistent for most glacier regions, and critical for accurate results of our assessment in less sampled regions, as shown for the Antarctic and Subantarctic Islands. The best way to correct possible deviations in past time series is to calibrate them against accurate long-term geodetic glacier elevation changes. Geodetic observations can be temporally enriched in all regions by unlocking historical United State spy satellite archives (e.g. KH-9 Hexagon and Corona declassified satellite imagery) and national historical airborne image archives.

### 5.5.3. Grid point artifact in polar regions

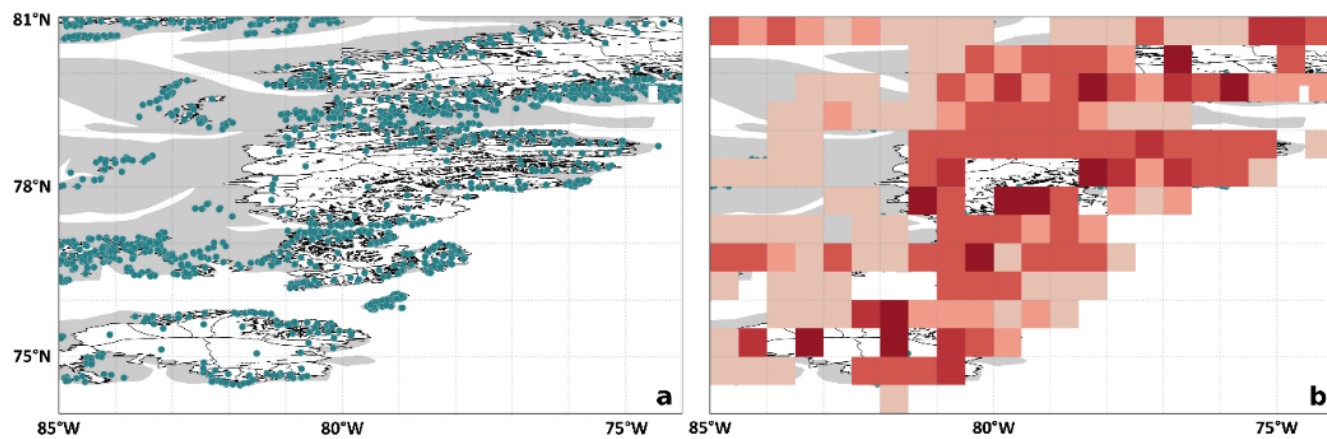

**Figure 11: Mockup example of the grid point artifact in polar regions.** (a) Glacier outlines (white) and their centroids (blue points) in the region Arctic Canada North under 1° grid cells. (b) Visual illustration of the 0.5° gridded annual mass-change integration.

To assess glacier mass changes, it is crucial to treat a glacier as a single and indivisible entity. Therefore, in order to properly integrate glacier-wide changes into a regular grid system, we consider a glacier belonging to a grid point if its centroid falls within that grid point's boundaries. If the grid cell is sufficiently large, it will encompass multiple glaciers at their full extension within the grid cell and the grid-point mean mass change will be determined accordingly. However, in cases where the grid cell is smaller than the glacier's surface area, the grid point containing the glacier's centroid will represent the mass change of the entire glacier, despite not all its extension is contained within the grid point (Fig. 11a). This discrepancy is particularly evident in polar regions above 60° latitude when integrating mass changes at a 0.5° global grid resolution. Polar grid cells are relatively smaller in area compared to the large polar glaciers. Consequently, this leads to a biased estimate of mass change at the grid point containing the glaciers centroid and consequent neighbouring glacierized grid points lack a mass change estimate (Fig. 11b).

This issue might be especially critical for deconvolving the glacier signal for gravimetry (GRACE, e.g. Blazquez et al., 2018; Chen et al., 2022) or other applications in polar regions due to coarse resolution of the ancillary datasets (usually not smaller than 0.5°). A potential solution for larger scale applications with coarser spatial resolution would be an area-weight per tile glacier area, but this would bring an additional bias related to the divisibility of the glacier signal.

### 5.5.4. Calendar years versus hydrological years

Our results present regional glacier mass changes spanning the hydrological years from 1976 to 2024. In glaciological terms, it is widely accepted that the hydrological year starts in winter with the onset of the accumulation season and concludes at the end of the summer or ablation season (Cogley et al., 2011). Consequently, the hydrological year varies across regions (South

and North Hemispheres and Tropics) and does not align to the calendar year. Gridded annual glacier mass change values for a given hydrological year will not be fully consistent. For grid-points located in the northern hemisphere glacier mass changes correspond to the period from the 1st October of the previous year to 30th September of the given year. Whereas grid-points in the southern hemisphere will represent glacier mass changes from the 1st April of the previous year to 31st March of the given year. It is important to note that this discrepancy, stemming from the input data, introduces inconsistencies and uncertainties on the gridded global assessments that users should acknowledge. For cumulative values over longer periods, these differences become less significant. Addressing this issue would need an increase in temporal resolution of the input data to monthly observations, which is not feasible at the global level purely relying on observations.

### 5.5.5 Glacier-specific area-change rates

We took into account the impact of glacier area changes in time for our regional mass change estimates as done by Zemp et al. (2019). In the former study, the evolution of area change rates is calculated for each first-order glacier region independently represented in two-time steps: a past period with no change rates and then a linear change rate calculated from a regional sample of glaciers with observations. This assumption is strong since glacier area changes are far from being linear. Still is the best possible guess considering the available observations. In this study, we further assume the glacier specific area-change rates behave as their regional mean change rates. This may introduce an additional bias since glacier specific area change rates strongly depend on the size of the considered glaciers, with an observed decreasing mean and increasing variability of relative area changes towards smaller glaciers (Fischer et al., 2014; Paul et al., 2004). This also implies that the average change obtained for a greater region ultimately depends on the glacier size distribution considered in a specific sample, which may or may not be representative of the full regional glacier size range present in the RGI60 inventory used here.

### 6. Data availability

The annual mass change time-series for individual glaciers and the derived global gridded annual mass change product at a spatial resolution of 0.5° latitude and longitude are available from the World Glacier Monitoring Service website at https://doi.org/10.5904/wgms-mce-2025-02. Earlier versions of the gridded product are available from the Copernicus Climate Data Store (CDS) web-based service (Dussaillant et al. 2024, DOI 10.24381/cds.ba597449). The present version will follow in the next C3S phase. FoG database version used here (WGMS, 2025; https://doi.org/10.5904/wgms-fog-2025-02) is available for download from the WGMS website at https://wgms.ch/data_databaseversions/. RGI version 6.0 is available from the National Snow and Ice Data Center (NSIDC, RGI consortium 2017; https://doi.org/10.7265/4m1f-gd79).

### 7. Code availability

The code is available at https://github.com/idussa/global_mb_fusion.

## 8. Conclusions

Building on the insights from previous assessments, we present a new dataset of glacier mass changes with global coverage and annual resolution from 1976 to 2024 based on an approach that combines the strengths from glaciological in-situ measurements and geodetic satellite observations. Our results offer critical insights into the acceleration of glacier melt at a global scale during recent years. While glaciers featured moderate mass loss rates in the 1970s and 1980s, the rates strongly increased in the 1990s, reaching about $375 \pm 16$ Gt per year in the last decade (2015-2024) and a record annual loss of $540 \pm 69$ Gt alone in 2023. Since 1976, glaciers globally have lost $9179 \pm 621$ Gt ($187 \pm 20$ Gt per year) of water, contributing $25.3 \pm 1.7$ mm ($0.5 \pm 0.2$ mm per year) to global mean sea-level rise. Nearly half (41%) of this loss, amounting to $10 \pm 0.2$ mm ($1.0 \pm 0.2$ mm per year) of sea level rise, occurred in the last decade (2015−2024), with 6% (1.5 mm) occurring in the global record year 2023 alone.

Compared to earlier assessments, our approach allows us to combine the global coverage from geodetic satellite observations with the annual variability and temporal coverage from glaciological in-situ measurements, globally for the period 1976-2024 and for some regions even further back in time. As a result, our new dataset provides long-term trends and interannual variability of glacier mass changes, both based on glacier observations, that can be aggregated at any regional to global scale. Geostatistical modelling allowed the interpolation and extrapolation of observational gaps and uncertainties in space and time. Our uncertainty assessment combines estimates of observational errors from glaciological and geodetic methods, considering spatial autocorrelation, density conversion and glacier area changes. We note that the mean annual mass-balance anomalies are extracted from a handful of glaciers in each region and thus, in each region, individual glaciers share a large fraction of these variabilities. Importantly, our glacier-wide uncertainty estimates are discussed in a leave-one/block-out cross validation exercise.

Given the available input data, the primary limitation of the new dataset is the scarcity of glaciological in-situ measurements, especially in the Southern Hemisphere, and the strongly reduced spatial coverage of geodetic observations before the year 2000. Our dataset would highly benefit with the support of in-situ monitoring programs in regions with low density of observations, the exploitation of historical archives such as airborne stereoscopic imagery and maintaining future spaceborne missions for cryosphere observation. Our approach would also benefit from scientific advances concerning volume-to-mass conversion for geodetic estimates and the mapping or modelling of glacier area changes over time.

**Author contribution**

I.D. and M.Z. initiated and coordinated the study. I.D. compiled the data, performed all the processing and analysis, wrote the manuscript and produced all figures. R.H. supported the uncertainty assessment algorithm. J.B. supported as project manager for the Copernicus Climate Change Service data product. All authors supported the analysis and commented on the manuscript.

**Competing interests**

The contact author has declared that none of the authors has any competing interests

**Acknowledgements**

The present method was developed within the World Glacier Monitoring Service and the Global Gravity based Groundwater product (G3P) project as contributions to G3P, The Glacier Mass Balance Intercomparison Exercise (The GlaMBIE Team et al., 2025) and the Copernicus Climate Change Service Climate Data Store (C3S-CDS). EB acknowledges support from the French Space Agency (CNES) and the Dispositif Institutionnel National d'Accès Mutualisé en Imagerie Satellitaire (DINAMIS).

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
