# Peer review of "Annual mass change of the world's glaciers from 1976 to 2024 by temporal downscaling of satellite data with in-situ observations"

_Earth System Science Data, 2024_

## Referee Comment (RC1)

**Review of *"Annual mass changes for each glacier in the world from 1976 to 2023"* by *Ines Dussaillant et al.**

Earth System Science Data: essd-2024-323

This study combines in-situ glaciological annual mass-balance observations with remotely sensed surface elevation data to provide an annual time series of individual glacier mass changes over the last decades to a century. Based on the assumption that glacier mass-balance anomalies are similar between neighbouring glaciers, annual observed glaciological mass-balance anomalies are extrapolated to all glaciers globally. These anomalies are combined with several geodetic samples to calibrate a mean annual mass change time series and its respective uncertainties.

This extensive data merging and extrapolation study presents an interesting analysis. However, major revisions are necessary before the publication of this dataset to avoid potential misuse by future users.

First of all, there are several major issues, questions and obscurities in the approach of how the uncertainties were estimated (see Sect. 1.1, 1.2) and how the cross-validation was done and analysed (see Sect. 1.3). Another critical aspect is the missing discussion about uncertainties at the glacier scale in the paper itself. The large per-glacier uncertainties become apparent only by checking the dataset itself. Many data users might not be able to use the per-glacier dataset as the uncertainties surpass the signal in many cases (see Sect. 1.4, Sect. 1.5).

Another essential aspect is to communicate clearly that this dataset is not a purely observed dataset since it was created by extrapolation. When extrapolating, predictions are made about unobserved glaciers/years based on an underlying assumption or rule. This dataset is based upon a model of belief of how the system behaves. Upon publication on the WGMS website, the dataset might be misused and falsely interpreted as observations. Most data users neglect or do not include the uncertainties in their frameworks. Therefore, it is important that data providers clearly state the limitations of their dataset. It may imply, for example, adapting the title and the analysis (see Sect. 1.6), and also "flagging" respective regions or glaciers by adding some "metadata" to the data (see Sect. 1.7). In general, the manuscript should focus much more on the uncertainties that vary regionally and temporally and showcase for what use case the data can be used and for what the data might not be useful.

There are several steps in the manuscript that I find unclear, particularly concerning the statistical analysis. I believe these issues need to be addressed by the author team and eventually reviewed by a statistical expert (if not already done). Additionally, the paper and data require a substantial rewrite before they can be reviewed properly. Consequently, I am only able to partially evaluate the manuscript and the dataset at this time. Only a revised version that incorporates or addresses my comments will enable me to fully assess the study and the dataset's added value.

My major comments are summarised in the 'General Comments' (Sect. 1). Line-by-line comments are in the 'Specific comments' (Sect. 2). After each manuscript section, the respective figures and tables and their captions are commented there as well.

**1 General comments**

**1.1 Standard error**

The manuscript uses two times the standard error as the uncertainty measure for the glacier mass balance anomalies. The standard error of the mean describes the uncertainty in estimating the mean, essentially providing the precision of the mean. In contrast, the standard deviation describes the variability of individual points around the mean, indicating the spread of the mass-balance anomalies. This distinction is visually explained on the following website: `https://seaborn.pydata.org/tutorial/error_bars.html`. In this context, the standard deviation would indicate how much the glacier anomalies deviate from their mean across different locations and years, which is likely of primary interest to data users. Using the standard error because it "allows" years with more observations to have smaller uncertainties is, to my knowledge, an uncommon approach. Additionally, the standard deviation appears to be used later for other uncertainties (e.g., $\sigma_{B_{glac}}$, density conversion factor uncertainty). The uncertainties are combined by adding the standard error and standard deviation together, which is probably also not a standard practice. Are there references that justify the approaches described in this paragraph?

The equation in line 193 is unclear and raises further confusion. The equation states that the uncertainty is two times the sum of the different standard deviations of the individual N selected annual glacier mass balance anomalies, divided by the square root of the number of observations. It appears these uncertainties correspond to the individual lines in Fig. 2b. However, it is unclear why a standard deviation is calculated for each glacier anomaly ($i$), which is then summed. Additionally, the explanation provided in the text (line 188) does not seem to align with the equation on line 193. To clarify, the code was briefly reviewed at `https://github.com/idussa/mb_data_crunching/blob/cgab8e10198583docb2fc1de809eo1e4bd5fbca3/2.1_spatial_anomalies/calc_global_gla_spatial_anom.py#L505`. Based on this, it seems the standard deviation is computed over the observations, not summed, which conflicts with the equation in line 193. The code then appears to calculate a mean over another variable. It also seems that the error is first calculated for every "line" shown in Fig. 2c, leading to an average in the script. However, there does not appear to be any summation applied, suggesting a possible discrepancy in the equation on line 193 or a misunderstanding of the correct line of code. Please clarify this process.

Moreover, the rationale behind using a factor of two for the standard error is unclear. Please clarify the reasoning behind this choice (described further in Sect. 1.3).

Another point of concern is that the current approach results in glacier mass-balance (MB) anomaly uncertainties that only depend on the amount of included glaciers and their differences in the anomaly. I suggest that a mass balance anomaly from a glacier located further away results in larger uncertainties compared to one that is nearer. Is this accounted for in the uncertainty estimates? Do the uncertainties increase if only distant glaciers are available? In some cases, this might occur naturally if the distant glaciers are not clustered, leading to significant differences in MB anomalies and, consequently,

larger uncertainties. However, if the available glaciers with MB time series are far away but clustered closely together, could the assessed uncertainties be underestimated? Is there any algorithm in place to prevent this potential underestimation?

**1.2 Uncertainties/Error propagation**

For the analysis of per-glacier mass balance uncertainties, the law of random error propagation is frequently used. It would be beneficial to explain, in each instance, why it is considered valid to assume that the errors are completely uncorrelated. Specific examples where random error propagation might not be valid or should at least be discussed are noted in the specific comments (e.g., **L191, L255-257, L269, L276, L281**). It may also be necessary to mention that assuming complete independence could lead to underestimating the actual uncertainties.

Regarding **L269, eq. 16**, it is stated that the errors are assumed to be completely correlated at regional scales, but the equation suggests that complete independence is assumed (as indicated by summing the square roots). Which assumption was actually applied in the results? This was not clear from the code.

**1.3 Leave-one-out cross validation**

Applying a leave-one-out cross-validation is crucial, and it is great that this validation is performed by using geodetic data available for all glaciers. However, given the nature of the reference glaciers, there are concerns about the validity of the conclusions drawn, such as the claim in line 452 that the "leave-one-out cross-validation results prove that our algorithm can capture the annual variability of individual glaciers."

As noted in lines 454-456, a major issue arises from the fact that the approach may work well for reference glaciers, often located in regions with nearby glaciers with mass-balance time series. Therefore, evaluating the metrics for these glaciers may not be representative. For example, removing Hintereisferner still leaves the nearby Kesselwandferner, which could skew the results. To provide robust estimates of the method's performance, a "data-denial/blocking" cross-validation approach is necessary. This involves analyzing how well the algorithm performs when assuming that, for instance, Hintereisferner has only one or two randomly selected glacier anomalies located far away, such as in the French Alps. Repeating this analysis across many glaciers and examining how the performance metrics change, as illustrated in Fig. 6, would provide a clearer understanding of the method's robustness. Additionally, evaluating how performance metrics vary with the number of considered glaciers would be valuable.

Another consideration is the selection of glaciers for cross-validation. Why are e.g. Echaurren Norte and other WGMS reference or benchmark glaciers not chosen for the cross-validation? Including all glaciers with at least 10 years of observations could allow for a more comprehensive analysis, even if some glaciers have fewer years of data and are not validated. This inclusion would enable assessment in regions without reference glaciers and ensure that performance metrics are not skewed by a few

well-sampled regions. Please evaluate the approach with a larger glacier sample and the data-denial experiment to better demonstrate the dataset's robustness or non-robustness.

Regarding validation, if direct glaciological mass-balance observations were not included in the calibration due to the lack of data over the baseline period 2010-2019, it would be beneficial to use these observations for additional validation if possible.

Finally, the claim that cross-validation shows the uncertainty estimates are on the "conservative" side and that the dataset has realistic uncertainties needs clarification. The assessment of whether the cross-validation errors are sufficiently small is based on comparing them to the assumed uncertainties of the dataset. However, this approach may allow for "inflating" the uncertainties until they encompass the cross-validation errors.

In relation to Fig. 6d, there is confusion about the comparison presented. If the y-axis represents $\sigma_{var_{\beta Y}}$ from line 193 (i.e., two times the standard error) and the x-axis shows the mean absolute error, there seems to be a comparison of two different types of errors. The metrics being compared are different in nature: the mean absolute error is calculated differently from the standard error. It is unclear whether these two metrics can be directly compared. Should the x-axis not display the RMSE (Root Mean Squared Error, i.e., typically larger than the MAE), as it involves estimating squared differences, which aligns more closely with the standard deviation? The standard deviation is typically used to measure the spread of errors around the mean, and RMSE would be more appropriate for comparing with it. Comparing RMSE on the x-axis with the standard deviation from the calibration on the y-axis would allow for a more consistent evaluation of prediction error (RMSE) relative to the inherent variability or spread of errors (standard deviation). Please verify this approach (if possible with a statistician) and provide a clear explanation for the chosen comparison, including its validity.

**1.4 Limited "glacier anomalies" for specific periods or regions**

The manuscript mentions a threshold of at least three glaciers with mass balance anomalies as necessary. However, it appears that in regions such as the Southern Andes or Subantarctic and Antarctic Islands, only Echaurren Norte is used as a source of MB anomalies before the year 2000, and after 2000, only two to three glaciers are included. Are these sources truly representative for all the RGI regions in these areas?

Similarly, in the Alps, the MB time series are extracted only from Claridenfirn and Silvretta. To my knowledge, these observations are based on very few stakes during the first 40 years (only two stakes?), which likely introduces higher uncertainty compared to more recent MB time series (e.g., Huss et al., 2021, `https://doi.org/10.3929/ethz-b-000474039`; Huss et al., 2017, `https://doi.org/10.3189/2015JoG15J015`). Was this increased uncertainty in the past data accounted for in your analysis? The dataset and the estimated individual glacier MB time series show relatively small uncertainties for Central Europe in the period when anomalies are sourced from only two glaciers. Please clarify how these factors were addressed.

**1.5    Uncertainty analysis - signal to noise ratio**

The manuscript would benefit from a more comprehensive uncertainty analysis that examines how uncertainties vary between regions, glaciers, and time periods. This analysis should include a review of the number of glacier mass balance anomalies used, the covered years, their distances, and the amount of geodetic samples. Such information is crucial for potential data users to assess whether the data are suitable for their purposes.

In addition to this analysis, it would be valuable to include a metadata file for each glacier or grid point. This file should detail these statistics and clarify whether a glacier is "unobserved" and if the regional mean was used instead. Ideally, the metadata file would also list the glacier names used to extrapolate the MB anomaly for any given glacier.

While reviewing the paper and examining the data, several questions arise: Where is the annual time series valuable and usueful, and where should caution. A quantitative analysis with statistical tests would be useful for addressing these questions (more discussion on usage cases stated by the authors is in Sect. 1.6).

One potential approach could be a "signal-to-noise" ratio test, where the standard deviation of the mean interannual MB time series is divided by the mean uncertainties (also represented as a standard deviation). If this ratio exceeds one, it suggests that the data adds value; if below one, it implies that uncertainties might overshadow the signal. While this simple ratio is not a rigorous statistical test, it can provide initial insights into data usability. For most glaciers outside Central Europe, the estimated uncertainties are so large that the interannual variability appears smaller than the uncertainty, indicating a signal-to-noise ratio below one (review Fig. 1 left), which raises concerns about data reliability. A more refined approach could involve detrending the time series and comparing the standard deviation of the residuals to the uncertainties (review Fig. 1 right). Repeating the analysis for different time periods could further clarify the data's reliability. Please check with a statistician if this test or another test is suitable. This type of analysis should be included in the manuscript and referenced in the abstract and data documentation.

**1.6    Usage of the dataset as described by the authors**

Among others, the following usages of the dataset are mentioned by the authors:

- L20: "new baseline for future glacier change modelling assessments and their impact on the world's energy, water, and sea-level budget."
- L376: "This versatility enables identification of individual years marked by significant glacier changes and the detection of zones with varying impacts. For instance, it allows to pinpoint glaciers within a region that were affected by specific annual climate variations (e.g. droughts, floods, heat waves, etc.), as well as those with a larger or smaller influence on the yearly contribution to hydrology and annual sea level rise."
- L391: "spatial and temporal impact of known glaciological trends and anomalies like, for example,

[Figure]

Figure 1: **Signal-to-Noise ratio analysis for the 20 regions of Dussaillant et al. (in review):** (left) Boxplots illustrating the signal-to-noise ratio, calculated as the ratio of the standard deviation of the mean interannual time series to the mean of the estimated total uncertainties for each glacier individually. A ratio below one indicates that the signal (interannual variability) is smaller than the noise (uncertainties). (right) Untrended signal-to-noise ratio, where a linear trend was removed from the time series to isolate the residuals. The ratio compares the standard deviation of these residuals (signal) to the total uncertainties. Values below one suggest that the residual variability is less than the uncertainties. (right) Untrended signal-to-noise ratio where a linear fit was applied to compute a trend, and then the signal was defined as the "residual" only.
In both plots, values below one potentially mean that the signal is smaller than the noise (here assumed to be the uncertainties). The signal-to-noise ratios were estimated from the entire provided time series of each region. The total uncertainties were estimated by assuming complete independence of the three given uncertainty sources.

> the Andes Megadrought (Gillett et al., 2006; Garreaud et al., 2017, 2020; Dussaillant et al., 2019) or the Karakoram anomaly (Farinotti et al., 2020; Gao et al., 2020; Ougahi et al., 2022) at an unprecedented yearly temporal resolution."

- L644: "... vast potential for applications in various fields within and beyond 645 glaciology. These include international cryosphere observation intercomparison exercises; multi-Essential Climate Variable (ECV) products; serving as invaluable resources for calibrating and validating climate models; and advancing our understanding of the broader implications of glacier melt on sea levels, freshwater resources, global energy budgets, and nutrient cycling. This work opens new opportunities for future assessments of global glacier mass changes at increased temporal resolutions, fostering a more detailed examination of their climate and hydrological impacts worldwide."

The manuscript suggests that the dataset can be used for a variety of applications; however, there are concerns about the practicality and reliability of these uses, especially considering the uncertainties involved. Also, some of the examples provided are not sufficiently concrete, and it is unclear how uncertainties are integrated into these applications.

Fig. 5 presents an example from Iceland, but uncertainties are not shown. It raises questions about the reliability of pinpointing individual years when uncertainties are accounted for. Iceland benefits from

relatively good coverage of mass balance time series and has a unique conditions due to the presence of volcaninc eruptions, and is thus not very representative of other regions.

For regions such as the Southern Andes, Subantarctic, and Antarctic Islands, where annual data before 2000 are derived from a single glacier, the added value of the dataset compared to using data from that single glacier (or the few glaciers available) needs clarification. The dataset's ability to represent these regions accurately, considering the associated uncertainties, requires a more detailed discussion.

In lines **357-366**, the manuscript discusses mass changes for regions like the Subantarctic Islands and Periphery. Since these estimates are based on extrapolated data from Echaurren Norte and a few other glaciers post-2000, the confidence in these annual estimates may be limited. A more thorough discussion on how uncertainties impact the interpretation of mass changes should be included if these estimates are to be retained in the manuscript.

In the abstract, line 20 states: "...new baseline for future glacier change modelling assessments". Do the authors believe that glacier models should now calibrate their models to match the per-glacier annual anomalies? In my opinion, glacier models should not, because the uncertainties are way too large. Most calibration procedures just completely neglect uncertainties, and in that case, just calibrating to highly uncertain per-glacier annual MB time series would give a false estimate of confidence. While glacier modelers may benefit from having a more detailed MB time series to better constrain model parameters (such as the precipitation factor), the current dataset may not yet provide the level of precision required for direct application in glacier modeling due to its significant uncertainties. Some modeling approaches do incorporate uncertainties, such as the Bayesian calibration framework utilized by Rounce et al. (2023), which includes uncertainties from the 2000-2019 geodetic observations of Hugonnet et al. (2021). Once the uncertainty estimation approach is clarified and cross-validation is repeated with a data-denial approach, the MB time series and associated uncertainties may become valuable for such calibration methods. However, it is noteworthy that Rounce et al. (2023) did not incorporate the 5-year averaged per-glacier mass change observations from Hugonnet et al. (2021) due to the excessive uncertainties associated with these observations. A similar issue may arise with the current dataset.

**1.7   Data and code documentation and availability**

Firstly, it is great that the code and data are made fully available.

I have a few comments first on the provided data:

- Hosting the extrapolated / modeled per-glacier annual data on the WGMS website could potentially lead to misunderstandings. Given that this dataset is not purely observation-based, its direct availability at the WGMS website could result in misleading conclusions. If the decision is made to include the data directly on the WGMS website, it is essential to include a comprehensive "meta"-dataset and a flagging system to highlight glaciers/areas where the uncertainties are too large to extract a signal (as discussed in Section 1.5).

- The type of uncertainty documented in the dataset requires clarification. The term "uncertainty" is used generically, but it is unclear whether this refers to two times the standard error as described in Line 187, or one or two times the standard deviation (related to Sect. 1.1).
- Currently, only individual uncertainties are provided, requiring data users to perform their own aggregation. It is strongly recommended to include a dataset with total uncertainties, as this will likely be the most utilized. Additionally, understanding the different sources of uncertainty and their origins took considerable effort. Enhanced documentation explaining these aspects would be beneficial for users.
- To enforce people, to look into the uncertainties, consider creating a netcdf file that has the mean time series, the total uncertainties, and a "flagging" system
- **Issues found in the per-glacier annual time series**
  - no glacier ID for Greenland, everywhere NaN values as IDs. Please update the glacier IDs for Greenland!
  - a bit confusing to have sometimes GLIMS_ids and sometimes RGI_ids ...

Comments on the github/code:

- It would be beneficial to include a README document in the GitHub repository that provides a brief overview of the functionality of each script. Such a document would guide interested users on where to find specific processes or analyses within the codebase. While the code does not need to be meticulously documented, a general overview in the README would greatly enhance the accessibility and usability of the repository.

**1.8 Terminology**

- The terms "(mean) glacier (annual) anomaly" appear to be unclear and could benefit from clarification. It is recommended to use more specific terminology, such as "(mean) glacier (annual) MB anomaly" or "glaciers with glaciological MB time series". This issue is particularly evident in Figure 1, where the term is not yet explained. The phrase "glacier anomaly" may imply that the glacier itself is unusual or deviates from expected behavior, rather than referring to mass-balance measurements. Including the term "mass-balance" would help clarify the meaning and ensure consistency throughout the manuscript (e.g., line 169 and other mentions).
- What is the difference between GTN-G regions and RGI6? For instance, in Line 102, GTN-G regions are mentioned, yet later references seem to align more closely with the "usual" RGI6 regions, with the exception of the Southern Andes, which is split differently. It would be beneficial to review the references to GTN-G and RGI6 throughout the manuscript to ensure consistency. If possible, it is recommended to use only one of these terms to avoid confusion.

**2 Specific comments**

To maintain relative conciseness in the review, specific comments have been provided without consistently using phrases such as "please reconsider" or "please change." However, many of these specific comments are intended as suggestions to guide improvements and offer constructive feedback, rather than as strict directives. Some comments made in the general comments section may be repeated in the specific comments section, potentially with additional elaboration or different descriptions. In the response to this review, you may disregard specific comments that you have already addressed in the general comments.

**2.1 Title, Abstract & Introduction**

- **L1** The title overstates the precision of the data and does not acknowledge the uncertainties sufficiently. Maybe change to something like "Uncertainties in extrapolating annual glacier mass changes from 1976 to 2023: Estimates for every glacier world-wide based on in-situ and geodetic data". At least the methods of your approach should somehow be incorporated in the title. For example, the phrase "from in-situ extrapolation" could be added to the title.
- **L19-20** In my opinion, you can not yet conclude that from the current leave-one-out cross validation (see Sect. 1.3). Also change: "in the conservative side" to "on the conservative side
- **L32-34** From these 500 glaciers, much less are actually usable time series. I think it would be valuable to rather mention how many you use (the glaciers with "glacier anomalies"). You also say, that nearly all glacier regions are represented. However, from these 500 glaciers, a lot are in Central Europe... related to Sect. 1 and idea of giving a clear overview of amount of used glacier MB anomalies and covered years-
- **L54** define FoG here (it is defined only later)
- **L54-58** long sentence, I don't understand the meaning of the sentence, specifically of "and evident..."
- **L67** "global glacier ice changes": mass or volume?
- **L68-71** Why are the geodetic observations of Hugonnet et al. 2021 giving you information on annual mass changes. For example from where do you get the density conversion information here? Rephrase the paragraph to clarify that you use the glaciological mass-balance observations.
- **L74** define "Fluctuations of Glaciers" at the first usage (L54)
- **L77-87** I am not sure if this paragraph is really necessary. You write here what you did, but not really the results. E.g. L78-80: strange to mention that here, as it does not tell the reader in which regions it works and in which it does not work so well...
  L85-87: the two sentences seem to be very similar, maybe combine in one sentence

**2.2 Data and Methods**

- **L93** do you apply any correction as the RGI dates are often different to year 2000?

- **L109** to which year do these outlines correspond?
- **L104** this should be in the caption, but actually the grey bars are almost not visible
- **L108** FoG already defined in database
- **L114-L115** "throughout their full hypsometry" : what is meant by that?
- **L136** Are these short-term geodetic records also excluded in the statistics of Fig. 1. Please clarify! Similarly, in Fig. 2, you show the short-term geodetic records, I would recommend removing or labelling them as you don't use them for the calibration.
- **L142-148** It is difficult to understand these steps if you haven't read the individual subsections. I would suggest to either put that at the end of Sect. 2.2.3, or instead move the important stuff into the captions, or somehow clarify that the individual steps will be explained later ...
- **L150** Fig. 2 caption: (see Fig. 2 comments below)
- **L160** What is $\sigma_{B_{glac}}$? I assume it corresponds to one standard deviation, but I think it is important to mention that. You often have the $\sigma$ term in your equations, maybe clarify here directly if those represent always one standard deviation.
- **L169** "glacier annual anomaly" $\rightarrow$ see Sect. 1.8
- **L168-170** You choose a threshold for the amount of years within 2011-2020 for a glacier to be chosen to be used for your calibration approach. However, how many years outside of the 2011-2020 years are necessary that a glacier is chosen to be used? I think this is important to mention / analyse to understand the "number of glacier anomalies" of Fig. 1
- **L174** "more than two...": replace with "at least three glaciers with mass-balance anomalies". This threshold is only valid for the search radius, as there are many periods and regions where the anomalies come from less than three glaciers? (related to Sect. 1.4)
- **L180** the uncertainties of the anomaly are computed without any inverse weighting correctly? Maybe clarify that!
- **L182** Do you mean here $\sigma_{B_{glac}}$? If yes, please clarify and maybe refer to eq. 1?
- **L183-186** These are almost the same sentences as in line 162-164! Remove one of them. Also typo: replace "ire" with "are"
- **L186-189** Why do you use two times the standard error here, why not the standard deviation. More about that in Sect. 1.1
- **L191** Why can you assume here random error propagation? Couldn't it be that on a specific year, glacier mass balance is under/over estimated for several glaciers because of a specific "climatic" phenomenon? (see also Sect. 1.2)
- **L194** "low confidence glaciological series" $\rightarrow$ how do you define "low confidence"? Are only high confidence glaciological series included in the statistics of "glacier anomalies" in Figure 1?
- **L196-197** What do you mean by that? I guess you mean by that that you want them to have anomalies until then? This sentence does not say something about how much glaciers with "glacier mass-balance anomalies" are used, but when I first read over the sentence I thought that you want to say that you somehow only select glacier mass-balance anomalies that cover the entire period 1976 to 2023. Please rephrase.
- **L197-200** How do you define "climatically similar" (see comment to Table 2)? Here you describe

gaps in mean calibrated glacier MB annual-anomalies that result from the aspect that no glaciers with observations for these gap years were found within the search radius? But do you also explain how the uncertainties are estimated for these gaps?

- **L212** add the unit of the density conversion factor (maybe best to make it in kg m-3 and then divide by 1000 in eq.5)

  it should also be 0.85+/-0.06 (not 0.60)

- **L215** here the $\sigma_\rho$ is in units of kg m-3, please be consistent. The acronym is also different to line 212.

- **L213** eq.6 is a bit confusing, as it is unclear for what this is done. Maybe add that this uncertainty is later used for the weighting algorithm of the geodetic samples in eq. 8 (if I got it correctly).

- **L218-222** I had to read this several times to hopefully understand it. You create for every geodetic sample of >= 5 years, an individual time series which goes over the entire considered period (not only the period of record of the geodetic sample), correct?

- **L222** here you write "only geodetic observations larger than 5 years" are used (also in L230), but in L155, you wrote longer or equal to 5 years. Which option did you choose?

- **L224-226** You explain the uncertainty estimate, but there is no equation to it. I have the feeling that exactly this is already done in eq. 9-12, maybe? If yes, please merge this sentence with the explanations there? Or is this where you basically convert $\sigma_{\bar{beta}_{g,y}}$ to $\sigma_{\bar{beta}_{k,y}}$?

- **L231-234** Maybe add the respective acronyms (such as $W_t$ for the second uncertainty part) at the end of the descriptive sentences. Like that it is easier to understand eq. 8, and otherwise the acronyms of eq. 8 are not explained.

- **L239** replace considering with "assuming", because you do not show that,or?

- **L241** I don't really understand why you call it "error" separation. You aggregated the errors beforehand for the weighting, but you have the errors already indivudally, so you don't need to separate them again, or? What eq. 9-11 do is basically averaging the uncertainties from the individual k mass change time series of the individual geodetic samples. If yes, maybe instead add something like that instead of saying error separation. You could also maybe explain everything from line from 239 to 244 before explaining the weighting, and then at the end explain how you get to the total uncertainties (L245). Then you don't need to say you look again at the errors separately...

- **L245** Why can you assume independence and add up the square roots of the different squared standard deviation uncertainty sources to get to the total uncertainty? If I get it right these three uncertainties are the ones available in the per-glacier files. In the data files the total uncertainty is not available, maybe consider adding the total uncertainty as dataset, or somehow clarify that in the manuscript

- **L240-246** What kind of errors do these uncertainties represent? Standard errors or standard deviation? I am just wondering because at least one of them apparently represents the standard error (i.e. the one calculated via line 192?)

- **L248-249** I find this sentence a bit confusing. Maybe clarify that this is what you did, i.e. rephrase by saying.... We calibrated a mean annual mass change for ...

- **L249** Explain what you mean with unobserved. If I understood correctly, it is a glacier that is not available in Hugonnet et al. 2021 (and also does not have any other geodetic observations or in-situ observations). Correctly?
- **L250** replace "Individual" with "individual"
- **L254** why is it only of the observed glaciers? Don't you use all glaciers where you created calibrated data and then in addition the "unobserved" glaciers?
- **L255-257** It is great that you account here for spatial correlations, and have identified that some error sources are significantly correlated spatially, such as elevation change, density conversion and annual anomaly prediction. Are there any figures/analysis for that?
  I am not an expert here, but I am wondering if you need to account for a similar spatial correlation when estimating the total uncertainty of an individual glacier mass balance time series? The reason is that the glacier anomalies are estimated from time series that are coming from different glaciers.
- **L263** Huss et al., in preparation: It would be good to add some details here. When looking into the density uncertainties of individual glaciers, they seem to be quite small.
- **L268** "errors to the real values": what do you mean by that
- **L269, eq.16** see Sect. 1.2
- **L270** "as independent". Why can you assume that?
- **L273** Fig. 2f does not exist
- **L276** Same as L268. Why can you assume that? you use different expression for the same aspect, i.e. assuming that the errors are independent, then write law of random error propagation... maybe stick with one thing
- **L279** Zemp et al. 2019: "et" is written in different text style
- **L281** regional mass loss uncertainties independent and uncorrelated : from where do you know that you can assume that...
- **L285** do you assume that all mass loss is above sea level? eq. 21-23 are not explained
- **L315** starts at the first year of mass change records, is a single mass change record sufficient to start from that year, or is is three glacier MB anomalies?

**Table 1**

- **L5** maybe add another row with the differences in the Uncertainty?

**Fig 1**

- It it strange that you show the location of all glaciological samples, although in this study you only use a fraction of these (i.e., just those with glacier anomalies). I think it makes much more sense to visualise those glaciers with a MB time series. I would prefer to see the hypsometry of those glaciers with "glac anomalies" instead of those from all glaciers with measurements.
- For me, the focus of this figure at the moment is to show the hypsometry of the glaciers of that region. For me it would be more interesting to see the statistics of the glaciers that are used for

the "glacier mass-balance anomalies" visually. For example, how many glacier MB anomalies are actually used for the individual years of the time series?

- maybe remove duplicate labels to make the figure less busy
- it is very difficult to see the hypsometry of the "glaciological" sample as the red is difficult to see
- you give the % of observed glaciers in terms of glaciological or geodetic observations. Is a single observation in one year that the glacier is here "observed"?
- The red shows the hypsometry of glaciers with any kind of glaciological observations (I count 468 glaciers there)? If I understand correctly the amount of glaciers with observed mass-balance time series are described by the "glac anomalies"? How many observations are necessary to be such a glacier? Maybe add that to the caption (it is later described in the text, but maybe good to explain it also here). Also related to that: change the wording of "glac anomalies" and describe that in the Fig. 1 caption, see Sect. 1.8).
- the grey "RGI6" glacier hypsometry is almost not visible (specifically if you print it)
- explain in caption the meaning of the circles (glacier region area). Location of the circles is sometimes far away from the region's glaciers. For example in CEU, SCA, NZL. In 17-SAN, there are two circles, probably from the two subregions, this needs to be documented further and if the two circles are kept the region should also be split up via the "black" lines.

**Table 2**

- Why did you exclude these specific glaciers?
- How did you choose the complementary glacier mass-balance anomalies?
- How did you choose the complementary normalized glacier mass-balance anomalies?
- typo: Hinteeisferner → Hintereisferner

**Fig. 2**

- It is a bit strange that you show the method for one of the best measured glaciers. Hintereisferner has an annual time series for over 60 years. I think it is very necessary to show at the same time the method for a less well sampled glacier, i.e. a glacier with no in-situ observations, with less and shorter available annual MB anomalies, and with only the Hugonnet et al. 2021 geodetic sample data. This second glacier corresponds better to most glaciers world-wide, I guess? You could add the other glacier in the same figure to have the comparison. Or move the Hintereisferner example in the supplements and add here another glacier.
- You show here the data of Hintereisferner only from 1952 onwards (i.e., the start of the Hintereisferner observational period). In the dataset that you want to publish, however, the new calibrated time series begins already in 1915. I think you should either mention this in the caption or show it in the figures. From the perspective of Hintereisferner, the period from 1915 to 1952 is the most interesting, as this is the period where your method actually creates new data.
- a: There is a big red cross in North Africa with the text $B_{glac}$ search. I find that rather confusing.
- b: you mention 10 glaciers with anomalies, I guess one of the 10 is HEF itself, correctly? So, what we see are 10 thin lines together with the inverse-distance weighted average and the uncertainties around it? And if I understand correctly, some anomalies are over the entire time period, and others are just over a period of time. It would be interesting to somehow visualise that. Probably this gets easier with a glacier with less glacier anomalies around.

I think it would be good to color the line showing the in-situ observed glacier mass-balance anomaly of Hintereisferner, I guess it will be near to the mean annual anomaly? At least it should be clarified in the caption or subplot that one of the lines represents the anomaly of Hintereisferner.

You do not explain what the grey shading is. You added the equation and from that I assume that the grey shading are the uncertainties from eq. 4, but I believe, both the grey shading and the equation need to be explained (e.g. by refering to eq. 4, and saying in caption that the shaded area corresponds to "two standard errors from the glacier MB anomalies and the glaciological sample uncertainties"?

- c: you write that you only use geodetic data with at least five years. However, in the plot, it seems like you also plotted the geodetic sample data for smaller periods? (even for single years, e.g. 2003). As you do not use them in the calibration, I would not include them in the plot, unless you somehow mark them in another color/style to clarify that these are just used for validation? Please also add how many grey lines there are, i.e. how many geodetic mass change observations were used

  From Sect. 2.2.3, I understood that every of these "k" lines have their own uncertainty estimated from (b). Although too complex to visualise, you might mention that in the subplot or caption.

  I would also prefer to see the uncertainties of the geodetic estimates instead of having red/blue filled ares to the zero line.

- d: What are the red and blue lines? I guess this is the same ones as in c? Not sure if it is necessary to keep them, but in any case, you need to describe them in the caption or in a legend.

  You just write, that the grey is "uncertainty". Please clarify what kind of uncertainty it is (see comment in Sect. 1.1).

  I would like to see here how the mean calibration time series changes to the actual in-situ HEF observations with that approach. I guess it is quite near as it is included. I think you should add a colored line with the actual Hintereisferner observations (similar as suggested for subplot b).

- add the corresponding equation numbers to b, c, d; consider adding legends into the subplots to clarify better what the lines mean

- Fig. 2b, d: "mean calibrated time series" and "mean annual anomaly" isn't it mean and "some kind of uncertainty" that you show?

table 3:

- "low confidence" glaciological/geodetic estimates : how is this defined?
- empirical function "of"...Hugonnet et al. 2022; Huss and Hugonnet (in prep) → called differently somewhere else, be consistent!

table 4:

- "uncertainty" : what uncertainty does that describe? Standard error/standard deviation/two times standard deviation?
- Dataset 1: there is no dataset with the total error. Interested people need to aggregate the uncertainties themselves, which is error-prone.
- Dataset 2: is the dataset really that large if you add all years together, it is easier to download just one file instead of many...
- Dataset 2: here you have mean time series and total error : again, what does total error represent?
- see table XX: -> ref. is missing
- time series start "of" hydrological year (of was missing)

**2.3 Results**

- **L326** What do the numbers represent? One standard dev. / std. error?
- **L333** Maybe refer here to Fig. 3; is it "m" or "m w.e", in Fig. 3 it is in m w.e.
- **L356** attention: your estimates are not "observations" anymore, as you apply basically a model to get the individual glacier MB time series. Consider rephrasing the word "observed".
- **L357-366** It is unclear to which figures or tables you refer to in this paragraphs
  You write volumes, but write "GT". I would suggest to replace volume everywhere by mass to coincide with the GT unit.
  Can you really say with confidence that these mass changes occurred on single years? More in Sect. 1.6
- **L359-362** I didn't understand the last part of the sentence, consider rephrasing the sentence.

Fig. 3:

- Global subplot: move that subplot up as the mass change time series axis is very near to the the the Russian Arctic subplot. To make some space, move maybe the global pie to the center left part of the plot
- It is a bit confusing that the global plot is in a different style than the regional subplots
- Some of the timeseries are difficult to see due to the bright colors of that region. Consider using black insted and only colo
- Caption: the area of the pie charts: maybe clarify that you mean the size of the circle

Fig. 4:

- Why does RGI19 and RGI05 regional glacier mass increases from 1979 to 2000? Are there any physical explanations for that or other studies showing the same? It seems like Zemp et al., 2019 had less positive MB on these two regions.
- You describe here the meaning of m w.e. but this concept is used earlier. Maybe rather describe it somewhere in the methods?

**2.4  Discussion**

Fig. 5

- More in Sect. 1.6
- year 1976: (i) and (ii) look very different. On the individual time series, it looks like all glaciers lost mass, while on the gridded dataset it seems rather that they gained mass. Why?

- **L398-400** "...selected considering..": but that means you select glaciers where you know it works. Isn't that kind of a bias towards specific glaciers?
- **L403** How many of the 32 glaciers are in CEU? Does a typical "reference" glacier not have much more "glacier anomalies" in their search radius than a "normal glacier"? see Sect. 1.3
- **L431** "seven glaciers" : In Fig. 2b, you wrote that there are 10 glaciers used to estimate the "Mean annual anomaly" of HEF. If you remove the anomaly of HEF for the cross-validation, there should be still 9 available glaciers, why is it now seven? In Fig. 8a, there are also 9 glaciers listed.
- **L435-436** To my knowledge, the RMSE (combines variance and systematic errors) and STD-diff (variability in the errors) do not directly verify whether there is no systematic error for Ba. Without checking the bias, you cannot confidently rule out systematic errors. I think it would be important to include the bias in Fig. 6,7.
- **L441-444** Maybe clarify by writing sth. like ... For XX out of XX glaciers, the actual standard deviation is >XX larger than the standard deviation estimated by the cross-validation. Do you believe that in general the interannual variability is underestimated by your approach? This is very important to clarify, as e.g. glacier models interannual variability largely depends on the precipitation factor.
- **L448-454** I don't fully agree that you can conclude all of that. I am specifically confused about the comparison of two times the standard error vs MAE. (discussed more in detail in comments about Fig. 6d in Sect. 1.3).
- **L454-456** This is a major problem, and should also be accounted for in your cross-validation and uncertainty analysis. This is one of the main reason why I can not accept the conclusions from this paragraph. See Sect. 1.3 for an idea of a data-denial analysis.
- **L476** are these cumulative mean mass losses within the uncertainties of Zemp et al. 2019? Please add uncertainties to these numbers!
- **L486** Deviations of more than XX : please clarify that you compare here to Hugonnet et al. 2021 2000-2019 period?

  Why do you have these differences in the regional trends w. Hugonnet et al. 2021 in the period 2000-2019? Do they only come from additional geodetic data used over that period and in these regions? Or do the glacier anomalies (in-situ observations) also influence the regional trend over the period 2000-2019?

  In the entire paragraph, the mentioned differences are within the large uncertainty ranges of the regions. If you mention the differences, I believe you should also mention that uncertainties are larger than the differences.

- **L535-537** You apply a "model" by extrapolating the glacier anomalies from reference glaciers to another glacier. For example, you assume that the anomalies are similar for nearby glaciers, you even select a glacier with the most similar climate for glaciers in regions without glacier anomalies. All these choices are somehow like a model. Therefore, you should not call this product a purely observation-based product.
- **L549** From where do you know if the estimated uncertainties are sufficient. The cross-validation that you applied can not tell you that as none of the glaciers of the cross-validation are e.g. in Southern Andes or Subantarctic and Antarctic Islands.
- **L550 onwards** You clearly state the problem, but I am wondering if it is then really valid to still give an annual time series for these regions and periods of extremely high uncertainties
- **Sect. 5.5.1** I think adding an overview of which geodetic data sources were included before year 2000, and which after year 2000 would help. Which kind of additional geodetic observations were used to compare to Hugonnet et al. (2021) in the period 2000-2019? Did you use any photogrammetric data for geodetic observations in the past? L560: I assume, this is future work, so maybe clarify that.
- **Sect. 5.3.2** maybe also discuss potential usage of terminus location (e.g. more used in glacier runoff studies)
- **L592** please remove, your dataset is not purely observational

Fig. 6 (see also Sect. 1.3)

- maybe add in the legend the amount of considered glacier anomalies for each glacier and the region where they are located in!
- FIg. 6a-d: Please adapt all subplots to have the same scale on the x-axis and y-axis, with equal tick labels and lengths. E.g. in python, you can set: $ax.set_aspect('equal')$. I believe this would help a lot to correctly interpret the plots (and would help to understand that all grey dashed lines are 1:1 lines).
- does it work better for those glaciers that have a lot of glacier anomalies nearby?
- Fig. 6d: Can you please clarify clearly the meaning of the y-axis (more details on that in Sect. 1.3).
- caption L419: I find it hard to interpret "std-diff"? Does every glacier count the same? Does it basically describe whether the differences are similarly large for the different glaciers?
- caption L422: mass change "trend" (not std. dev)
- caption L423: mass change "standard deviation" (not trend)
- caption L424: x-y descriptions does not display the actual Fig. 6d. As all other figures represent on the y-axis the leave-one-out cross-validation results, it would be best to exchange x and y-axes in Fig. 6d.

Fig. 7

- maybe add the uncertainties of the "leave-one-out" time series
- caption L459: maybe add the word "observed" here. If I understand it correctly the time series that is actually used in the dataset (i.e., e.g. for HEF Fig. 2d) is not shown here, or? Or is it in case

of these reference glaciers the same?

- caption L459-460: caption description of right and left is reversed! Eventually show the other non-selected reference glaciers in the supplements or appendix

**2.5 Data and Code availability**

see Sect. 1.7

**2.6 Conclusions**

- **L628** Here again, I would prefer to remove that statement of "independence" and of "purely observational nature".
- **L629-630** With the current cross-validation analysis and figures that I have, I can not yet conclude that. Please recheck, once you refined the cross-validation (see Sect. 1.3).
- **L645-end** These are very broad use cases, maybe a bit more concrete and nuanced use cases would help to clarify what this new dataset can do (i.e., where it adds value) compared to other existing datasets.
- **L657** replace "The" with "the

---

## Author Comment (AC1)

**Reply to reviewer #1**

We are very thankful for the anonymous reviewer's detailed and thorough analysis of our dataset and constructive review of our manuscript. Most of their suggestions are feasible within the time frame of the review. We acknowledge that the proposed changes will make the dataset (i) more robust in terms of uncertainty assessment and (ii) better express the strengths and limitations of the dataset to potential users.

Please find here our preliminary answers to their main comments. The resulting improvements and the response to the specific/minor comments will be detailed in our final response letter and a fully revised manuscript if the editor considers that our manuscript is appropriate for Earth System Science Data.

The document is color coded as follows:
**Black: reviewer general comment**
**Green: answers to reviewer**

This study combines in-situ glaciological annual mass-balance observations with remotely sensed sur- face elevation data to provide an annual time series of individual glacier mass changes over the last decades to a century. Based on the assumption that glacier mass-balance anomalies are similar be- tween neighbouring glaciers, annual observed glaciological mass-balance anomalies are extrapolated to all glaciers globally. These anomalies are combined with several geodetic samples to calibrate a mean annual mass change time series and its respective uncertainties.

This extensive data merging and extrapolation study presents an interesting analysis. However, major revisions are necessary before the publication of this dataset to avoid potential misuse by future users.

First of all, there are several major issues, questions and obscurities in the approach of how the un- certainties were estimated (see Sect. 1.1, 1.2) and how the cross-validation was done and analysed (see Sect. 1.3). Another critical aspect is the missing discussion about uncertainties at the glacier scale in the paper itself. The large per-glacier uncertainties become apparent only by checking the dataset itself. Many data users might not be able to use the per-glacier dataset as the uncertainties surpass the signal in many cases (see Sect. 1.4, Sect. 1.5).

Another essential aspect is to communicate clearly that this dataset is not a purely observed dataset since it was created by extrapolation. When extrapolating, predictions are made about unobserved glaciers/years based on an underlying assumption or rule. This dataset is based upon a model of belief of how the system behaves. Upon publication on the WGMS website, the dataset might be misused and falsely interpreted as observations. Most data users neglect or do not include the uncertainties in their frameworks. Therefore, it is important that data providers clearly state the limitations of their dataset. It may imply, for example, adapting the title and the analysis (see Sect. 1.6), and also "flagging" respective regions or glaciers by adding some "metadata" to the data (see Sect. 1.7). In general, the manuscript should focus much more on the uncertainties that vary regionally and temporally and showcase for what use case the data can be used and for what the data might not be useful.

There are several steps in the manuscript that I find unclear, particularly concerning the statistical analysis. I believe these issues need to be addressed by the author team and eventually reviewed by a statistical expert (if not already done). Additionally, the paper and data require a substantial rewrite before they can be reviewed properly. Consequently, I am only able to partially evaluate the manuscript and the dataset at this time. Only a revised version that incorporates or addresses my comments will enable me to fully assess the study and the dataset's added value.

My major comments are summarised in the 'General Comments' (Sect. 1). Line-by-line comments are in the 'Specific comments' (Sect. 2). After each manuscript section, the respective figures and tables and their captions are commented there as well.

**General comments**

**1.1 Standard error**

The manuscript uses two times the standard error as the uncertainty measure for the glacier mass balance anomalies. The standard error of the mean describes the uncertainty in estimating the mean, essentially providing the precision of the mean. In contrast, the standard deviation describes the variability of individual points around the mean, indicating the spread of the mass-balance anomalies. This distinction is visually explained on the following website: https://seaborn.pydata.org/tutorial/error_bars.html. In this context, the standard deviation would indicate how much the glacier anomalies deviate from their mean across different locations and years, which is likely of primary interest to data users. Using the standard error because it "allows" years with more observations to have smaller uncertainties is, to my knowledge, an uncommon approach.

As the reviewer correctly defines, the standard error of the mean describes the uncertainty in estimating the mean, while the standard deviation can refer to any measure of sample variability or spread. However, it seems unclear to the reviewer that the two terminologies are not mutually exclusive. By definition, the standard error is also a standard deviation, namely the standard deviation of all possible means. During uncertainty analysis, a standard deviation implicitly refers to the associated estimate (which can be a mean, or other), and is thus always a measure of uncertainty. We confirm that this is also the case everywhere in our study.

We agree with the reviewer that the justification of our uncertainty methods was sometimes convoluted, and that the language used was inconsistent. We will better explain those in the revised manuscript, either by relating the uncertainties to their physical meaning for a new source, or by justifying how they propagate from other sources.

Additionally, the standard deviation ap- pears to be used later for other uncertainties (e.g., $\sigma B_{glac}$, density conversion factor uncertainty). The uncertainties are combined by adding the standard error and standard deviation together, which is probably also not a standard practice. Are there references that justify the approaches described in this paragraph?

We refer to the answer above on the fact that standard error and standard deviation are not mutually exclusive, and instead refer to the same concept in uncertainty analysis. Standard deviations always refer to the variability of the estimate described (whether it is a means of samples, a modelling estimate, a temporal or spatial sum for aggregating to larger regions).

For instance, the uncertainty in density conversion of Huss (2013) mentioned by the reviewer is a standard deviation of estimates of density conversions, derived from a modeling exercise. As we do not have any other prior knowledge and use a constant mean density conversion globally, also corresponding to Huss (2013), this standard deviation corresponds to our uncertainty in density conversion at the glacier scale.

The equation in line 193 is unclear and raises further confusion. The equation states that the uncertainty is two times the sum of the different standard deviations of the individual N selected annual glacier mass balance anomalies, divided by the square root of the number of observations. It appears these uncertainties correspond to the individual lines in Fig. 2b. However, it is unclear why a stan- dard deviation is calculated for each glacier anomaly ($i$), which is then summed. Additionally, the explanation provided in the text (line 188) does not seem to align with the equation on line 193. To clarify, the code was briefly reviewed at https://github.com/idussa/mb_data_crunching/blob/c9ab8e10198583d0cb2fc1de809e01e4bd5fbca3/2.1_spatial_anomalies/calc_global_gla_spatial_ anom.py#L505. Based on this, it seems the standard deviation is computed over the observations, not summed, which conflicts with the equation in line 193. The code then appears to calculate a mean over another variable. It also seems that the error is

first calculated for every "line" shown in Fig. 2c, leading to an average in the script. However, there does not appear to be any summation applied, suggesting a possible discrepancy in the equation on line 193 or a misunderstanding of the correct line of code. Please clarify this process. Moreover, the rationale behind using a factor of two for the standard error is unclear. Please clarify the reasoning behind this choice (described further in Sect. 1.3).

There was an error in the notation of Equation 4b in the submitted manuscript. We corrected it and simplified it from:

$$\sigma_{var_{\beta_Y}} = 2 \cdot \frac{1}{\sqrt{n_Y}} \sum_{i=1}^{N} Stdev\, \beta_i \qquad \text{to} \qquad \sigma_{IDW_{\beta_Y}} = \frac{\bar{\sigma}_{\beta_{i,n_Y}}}{\sqrt{n_Y}} \qquad (4b)$$

where $\bar{\sigma}_{\beta_{i,n_Y}}$ is the per-glacier mean of the standard deviation of yearly glacier anomalies.

The sum in the previous equation was an error of notation, it was intended to show that we use the mean of the yearly standard deviations. In addition, we changed the subscript "var" for "IDW" to express that this term represents the uncertainty in the IDW spatial interpolation (the way the term is derived, whether it is a standard error of the mean or other, should not be reflected in the terminology as already showed by the equation).

We agree that the use of the factor of two in this equation is not correct. Our uncertainty analysis assumes, as is common practice, that the distributions of errors are normal (i.e. Gaussian). This is what allows us to propagate $1\sigma$ terms throughout, but this is not applicable to $2\sigma$. We therefore modify all uncertainty equations to be at the $1\sigma$ level, and then report results at the $2\sigma$ level (i.e., approximately 95% confidence) in the text, figures and tables, by multiplying any given $1\sigma$ uncertainty estimate by 2. This will be corrected in the revised manuscript text and the code will be changed accordingly.

Another point of concern is that the current approach results in glacier mass-balance (MB) anomaly uncertainties that only depend on the amount of included glaciers and their differences in the anomaly. I suggest that a mass balance anomaly from a glacier located further away results in larger uncertainties compared to one that is nearer. Is this accounted for in the uncertainty estimates? Do the uncer- tainties increase if only distant glaciers are available? In some cases, this might occur naturally if the distant glaciers are not clustered, leading to significant differences in MB anomalies and, consequently larger uncertainties. However, if the available glaciers with MB time series are far away but clustered closely together, could the assessed uncertainties be underestimated? Is there any algorithm in place to prevent this potential underestimation?

We agree with the referee that our methodology lacks a way to capture the varying errors with distance to the measurement used for interpolation. At present, our uncertainty in mean glacier mass-balance (MB) anomaly depends only on the number of glaciers included, their individual uncertainties, the distance of the selected anomalies used (inverse-distance weighting spatial interpolation, for which there is no integrated error propagation) and the differences of the individual glacier anomalies as standard error or a measure of the uncertainty in estimating the yearly means (EQ 4b).

To solve this issue within the timing of this review, the best solution is to maintain the IDW interpolation. A potential solution is to replace the IDW spatial interpolation by a kriging spatial interpolation (that includes error propagation natively). Our only worry is that the kriging implementation might not be sufficiently efficient computationally to run on our dataset in a reasonable time. Moreover, since it is still an exploratory solution, we need to analyze first if it really makes a difference on the uncertainty assessment. This is something to explore in the future. For the sake of time we propose adding this in the discussion of the dataset limitations.

As the reviewer suggests, we will make sure users are aware of where our dataset is better and less constrained, not only through the uncertainties but also through adding the necessary metadata. Additionally, we will provide a clear illustration of the results in the manuscript figures and discussion

to flag the periods and regions where the dataset is less constrained. As an example: the reduced robustness of the mean calibrated time series during gap-filled years where neighboring glaciers have been used, will be evident not only because of their larger uncertainties, but also clearly delineated in the manuscript figures with dashed lines.

**1.2 Uncertainties/Error propagation**

For the analysis of per-glacier mass balance uncertainties, the law of random error propagation is frequently used. It would be beneficial to explain, in each instance, why it is considered valid to assume that the errors are completely uncorrelated. Specific examples where random error propagation might not be valid or should at least be discussed are noted in the specific comments (e.g., **L191, L255-257, L269, L276, L281**). It may also be necessary to mention that assuming complete independence could lead to underestimating the actual uncertainties.

We agree the text was not fully clear in showing where assumptions of correlation or no correlation are applied between sources, as we instead primarily focused on spatial correlations. The reason behind this difference in focus is that the assumption of correlation between sources has negligible impact compared to the spatial correlation of errors within the same source. This is because we only have 3 main sources of uncertainties, while we have 200,000 glaciers distributed spatially. For instance, for a sum (of volume changes), assuming for the sake of the example that all errors have the same magnitude, if 3 sources of errors are combined as uncorrelated or fully correlated affects the total uncertainty by a factor of 3/sqrt(3) = 1.7. For 200,000 glaciers, if errors are propagated spatially assuming they are uncorrelated or fully correlated, it affects the total uncertainty by a factor of 200000/sqrt(200000) = 447. This is why we focus on estimating spatial correlation to constrain errors more robustly in this study, while we give less attention to the correlation between sources that has little impact.

Nonetheless, we will edit the manuscript text to clarify the physical meaning of each term and the justification behind assumptions of error propagation as either uncorrelated, fully correlated, or correlated to a certain degree (for spatial correlations) at every instance. Below, our response and propose text to the five cases explicitly mentioned by reviewer #1 (L191, L255-257, L269, L276, L281)

**Case 1: referring to L191 in submitted manuscript.**
Proposed text:

We then estimate the uncertainty in the mean annual anomaly $\sigma_{\overline{\beta}_{g,Y}}$ by combining the uncertainty propagated from glaciological estimates $\overline{\sigma}_{B_{glac,Y}}$ and the uncertainty in the IDW spatial interpolation $\sigma_{IDW_{\beta_Y}}$. These two uncertainties capture independent sources (errors in interpolation and errors in glaciological measurement), and we thus propagate them as uncorrelated (Equation 4). We note that this assumption is conservative, because the variability of the glaciological estimates used to constrain the uncertainty in spatial interpolation is also affected by the uncertainties of glaciological estimates, which are therefore double counted. Uncertainties of glaciological estimates are largely independent spatially from one another, as they originate from differences in techniques, conditions, or locations in the field measurements. We thus combine them as fully uncorrelated (Equation 4a). The uncertainty in the IDW spatial interpolation is not directly provided by this method and thus delicate to assess, here we chose to estimate it using the variability of the sample (Equation 4b).

$$\sigma_{\overline{\beta}_{g,Y}} = \sqrt{\overline{\sigma}_{B_{glac,Y}}{}^2 + \sigma_{IDW_{\beta_Y}}{}^2} \qquad \text{(4 - edited)}$$

**Case 2: referring to L255-257 in submitted manuscript.**
We keep this paragraph as is, since it explains that spatial correlation is considered between three error sources: elevation change, density conversion and annual anomaly prediction.

**Case 3: referring to L276 in submitted manuscript.**

Proposed text:

We propagate the uncertainty in the specific regional mass change, the uncertainty in the regional area (Paul et al., 2015) and the uncertainty in the area change considering them uncorrelated. Errors in the area stem mostly from remote sensing delineation errors, while errors in area change stem from a lack of multi-temporal outlines to constrain area change. They are largely uncorrelated with error sources described above on elevation change, glaciological measurements and anomalies. However, elevation change estimates usually already consider errors in area at the scale of each glacier, so we might conservatively be double counting these.

$$\sigma_{\Delta M_{R,Y}} = \left|\Delta M_{R,Y}\right| \sqrt{\left(\frac{\sigma_{B_{R,Y}}}{B_{R,Y}}\right)^2 + \left(\frac{\sigma_{S_R}}{S_R}\right)^2 + \left(\frac{\sigma_{\Delta S_{R,Y}}}{\Delta S_{R,Y}}\right)^2} \qquad (19)$$

**Case 4: referring to L281 in submitted manuscript.**
Proposed text:

To simplify the combination of annual values into long term trends or cumulative annual values, we assume the yearly uncertainty to be independent of other years. This is true for glaciological measurement, having an independent uncertainty estimation for each individual year of the time series, but not for the elevation change measurements, where uncertainties are correlated over the years of the survey period. This approach was chosen to make the dataset user friendly.

Regarding **L269, eq. 16**, it is stated that the errors are assumed to be completely correlated at regional scales, but the equation suggests that complete independence is assumed (as indicated by summing the square roots). Which assumption was actually applied in the results? This was not clear from the code.

**Case 5: referring to L269 in submitted manuscript.**
As explained above, we follow the assumption that correlation between sources has a negligible impact compared to the spatial correlation of errors within the same source. For this reason, at L269 we express that, after applying spatial correlation within error sources, we combine all sources of error propagated at the regional-scale as independent.

**1.3 Leave-one-out cross validation**

Applying a leave-one-out cross-validation is crucial, and it is great that this validation is performed by using geodetic data available for all glaciers. However, given the nature of the reference glaciers, there are concerns about the validity of the conclusions drawn, such as the claim in line 452 that the "leave-one-out cross-validation results prove that our algorithm can capture the annual variability of individual glaciers."

We will clarify these claims in the revised manuscript after the leave-block-out cross-validation analysis.

As noted in lines 454-456, a major issue arises from the fact that the approach may work well for ref- erence glaciers, often located in regions with nearby glaciers with mass-balance time series. Therefore, evaluating the metrics for these glaciers may not be representative. For example, removing Hintereis- ferner still leaves the nearby Kesselwandferner, which could skew the results. To provide robust esti- mates of the method's performance, a "data-denial/blocking" cross-validation approach is necessary. This involves analyzing how well the algorithm performs when assuming that, for instance, Hintereis- ferner has only one or two randomly selected glacier anomalies located far away, such as in the French Alps. Repeating this analysis across many glaciers and examining how the performance metrics change, as illustrated in Fig. 6, would provide a clearer understanding of the method's robustness. Additionally, evaluating how performance metrics vary with the number of considered glaciers would be valuable.

Please evaluate the approach with a larger glacier sample and the data-denial experiment to better demonstrate the dataset's robustness or non-robustness.

We appreciate the reviewer's idea, and we agree that performing a so-called "data-denial/blocking" cross-validation approach will certainly add more insight into the robustness of our estimates. If the editor accepts to consider a revised manuscript, we will perform this analysis and add the results in the revised manuscript.

In more detail: For consistency with the literature on spatial statistics, we choose to use the term leave-block-out cross-validation analysis. The process will be similar to the leave-one-out cross validation, with the difference that, instead of removing only the reference glacier time series, we will remove all the spatially selected glacier anomalies surrounding the reference glacier at increasing distances ranges. The mean and standard deviation of the residuals will be calculated at every distance step, to assess potential systematic errors (with the mean), and the magnitude of random errors (with the standard deviation). Results will be plotted showing these errors as a function of the distance to the closer glacier anomaly considered. We will perform this analysis over our selected sample of reference and benchmark glaciers due to the reasons stated in the next answer.

Another consideration is the selection of glaciers for cross-validation. Why are e.g. Echaurren Norte and other WGMS reference or benchmark glaciers not chosen for the cross-validation? Including all glaciers with at least 10 years of observations could allow for a more comprehensive analysis, even if some glaciers have fewer years of data and are not validated. This inclusion would enable assessment in regions without reference glaciers and ensure that performance metrics are not skewed by a few well-sampled regions.

Glaciological time series are subject to biases inherent to the glaciological method. The WGMS highly recommends reference (+30 years) and benchmark (+10 years) glaciers glaciological time series to be reanalyzed every 10 years by calibrating them with long term trends derived from high resolution elevation change measurements (Zemp et al., 2013). We intentionally chose to perform the leave-one-out cross validation experiment with a selected list of reference and benchmark glaciers known to have been reanalyzed. These time series stand as the only ground truth available for validation of our global assessment. The decision of not using all glaciological time series in the experiment is justified by reducing the risk of validating over potentially erroneous "truths".

Regarding validation, if direct glaciological mass-balance observations were not included in the calibra- tion due to the lack of data over the baseline period 2010-2019, it would be beneficial to use these observations for additional validation if possible.

We disagree with the referee's comment for the same reasons stated above. These time series do not correspond to reference or benchmark glaciers and therefore might be biased due to the lack of reanalysis. To reduce the risk of validation against biased measurements, we intentionally exclude these glaciers and all non-reanalyzed time series from our cross-validation experiment.

Finally, the claim that cross-validation shows the uncertainty estimates are on the "conservative" side and that the dataset has realistic uncertainties needs clarification. The assessment of whether the cross-validation errors are sufficiently small is based on comparing them to the assumed uncertainties of the dataset. However, this approach may allow for "inflating" the uncertainties until they encompass the cross-validation errors.

The reviewer's comment is somewhat unclear, and we interpret that their statement "this approach may allow for inflating the uncertainties" refers to the practice of iterating (i.e. making changes) on the uncertainty calculation until they agree with the cross-validation results. If this is what is meant, we disagree with the reviewer that this is a potential issue. Iterating to improve theoretical uncertainty quantification until it matches empirical uncertainty estimates from the cross-validation is a good scientific practice, and the very purpose of cross-validation. It helps identify potential gross errors (mistakes in implementation) and ensures a realistic estimation of uncertainties. This is true as-long-as the cross-validation is representative of the conditions in which the methodology is applied for the whole dataset. In this case, as pointed out by the reviewer, we did not sufficiently discriminate estimates spatially during the leave-out process. The addition of the new leave-block-out cross-validation proposed by the reviewer should further

improve this.

We note however that the cross-validation cannot identify some sources of systematic errors already present in the estimates used (as they are validated against themselves), only the ones that might be introduced by our methodology. We will add sentences in the text to clarify these points.

In relation to Fig. 6d, there is confusion about the comparison presented. If the y-axis represents $\sigma var_{\beta Y}$ from line 193 (i.e., two times the standard error) and the x-axis shows the mean absolute error, there seems to be a comparison of two different types of errors. The metrics being compared are different in nature: the mean absolute error is calculated differently from the standard error. It is unclear whether these two metrics can be directly compared. Should the x-axis not display the RMSE (Root Mean Squared Error, i.e., typically larger than the MAE), as it involves estimating squared differences, which aligns more closely with the standard deviation? The standard deviation is typically used to measure the spread of errors around the mean, and RMSE would be more appropriate for comparing with it. Comparing RMSE on the x-axis with the standard deviation from the calibration on the y-axis would allow for a more consistent evaluation of prediction error (RMSE) relative to the inherent variability or spread of errors (standard deviation). Please verify this approach (if possible with a statistician) and provide a clear explanation for the chosen comparison, including its validity.

The confusion of the reviewer here is completely acceptable. Thanks to this comment we were able to detect that there was also confusion among coauthors in terms of the best metrics to analyze the cross-validation results. We have now agreed upon using only the mean of residuals and the standard deviation of residuals as metrics to quantify potential systematic errors and random errors within the cross-validation results, respectively. We will not use the mean absolute error or the RMSE, since they don't provide any additional information. We will clarify the meaning of this parameter in the revised manuscript Figures and text discussion and correct the panels from Fig.6 accordingly.

**1.4 Limited "glacier anomalies" for specific periods or regions**

The manuscript mentions a threshold of at least three glaciers with mass balance anomalies as necessary. However, it appears that in regions such as the Southern Andes or Subantarctic and Antarctic Islands, only Echaurren Norte is used as a source of MB anomalies before the year 2000, and after 2000, only two to three glaciers are included. Are these sources truly representative for all the RGI regions in these areas?

The Southern Andes is a special case because there is only one long-term and continued glaciological time series available for the Central Andes: Echaurren Norte (1976-2023, which is also the only reference glacier in the entire Southern Hemisphere) and only one sufficiently long glaciological time series for the Patagonia region: Martial Este (2001-2023). Both these regions are extremely different in climatology, and we decided to process them differently, considering the 2nd Order RGI regions for the Southern Andes, dividing Patagonia from the Central Andes at 46S. We intentionally tuned the Echaurren Norte anomaly as the mean annual glacier MB anomaly for the Central Andes, the Martial Este anomaly for the Patagonia Andes. The mean annual glacier MB anomaly uncertainty for both regions was calculated using the standard error of these two glaciological time series over their common period.

Past glacier annual mass change assessments (Zemp et al. 2019) used the full annual signal from Echaurren Norte "as is" to estimate glacier mass changes in the entire Andes, as well as all time-series in the Southern Hemisphere: New Zealand, Low Latitudes, Antarctic and subantarctic. In our study, we decided to include the Echaurren's full time series only for the Central Andes, where it belongs and where it is more likely to be representative of the climatology. For Patagonia, New Zealand, Low Latitudes, Antarctic and subantarctic we only include the Echaurren time series only for Gap filling of the past period (before the glaciological observational period of each independent regional sample). Furthermore, for each independent region, and to reduce the effect of possible climatic differences, the amplitude of the Echaurren glacier anomaly on these gap years is normalized to the amplitude of the mean glacier anomalies of the regional sample. The reduced robustness of the mean calibrated time series during these gap-filled years is apparent on the

larger uncertainties in the time series. We will make sure to show this more clearly in the revised manuscript figures using dashed lines over the gap-filled years of the over regional time series.

We acknowledge that considerable uncertainty remains, but we are confident that our present approach is better constrained than past studies, and it is the only possible way to go back in time for these regions, considering the lack of past period observations in the Southern Hemisphere.

Similarly, in the Alps, the MB time series are extracted only from Claridenfirn and Silvretta. To my knowledge, these observations are based on very few stakes during the first 40 years (only two stakes?), which likely introduces higher uncertainty compared to more recent MB time series (e.g., Huss et al., 2021, https://doi.org/10.3929/ethz-b-000474039; Huss et al., 2017, https://doi.org/10. 3189/2015JoG15J015). Was this increased uncertainty in the past data accounted for in your analysis? The dataset and the estimated individual glacier MB time series show relatively small uncertainties for Central Europe in the period when anomalies are sourced from only two glaciers. Please clarify how these factors were addressed.

As explained in the answer above regarding our decision to use the standard error as measure of uncertainty, this increased uncertainty in the past data has been accounted for in our analysis. The standard error allows years with few observations to get larger errors than years having a larger observation sample (i.e. larger errors in past years where only a few series are considered, or in regions with few time series). This effect is apparent and well represented in our resulting uncertainties for, e.g. Central Europe, where uncertainties are two times larger before 1952 compared to the better constrained period after 2000. This is also apparent in other regions during periods where only few, or neighboring region glacier time series are used. In general, our results achieve a consistently good representation of the uncertainties across all regions, with larger uncertainties in regions and periods with small glaciological samples or where neighboring glacier time series are used for filling gap years. And vice versa, lower uncertainties in regions and periods with large glaciological samples.

**1.5 Uncertainty analysis - signal to noise ratio**

The manuscript would benefit from a more comprehensive uncertainty analysis that examines how uncertainties vary between regions, glaciers, and time periods. This analysis should include a review of the number of glacier mass balance anomalies used, the covered years, their distances, and the amount of geodetic samples. Such information is crucial for potential data users to assess whether the data are suitable for their purposes.

In addition to this analysis, it would be valuable to include a metadata file for each glacier or grid point. This file should detail these statistics and clarify whether a glacier is "unobserved" and if the regional mean was used instead. Ideally, the metadata file would also list the glacier names used to extrapolate the MB anomaly for any given glacier.

While reviewing the paper and examining the data, several questions arise: Where is the annual time series valuable and usueful, and where should caution. A quantitative analysis with statistical tests would be useful for addressing these questions (more discussion on usage cases stated by the authors is in Sect. 1.6).

One potential approach could be a "signal-to-noise" ratio test, where the standard deviation of the mean interannual MB time series is divided by the mean uncertainties (also represented as a standard deviation). If this ratio exceeds one, it suggests that the data adds value; if below one, it implies that uncertainties might overshadow the signal. While this simple ratio is not a rigorous statistical test, it can provide initial insights into data usability. For most glaciers outside Central Europe, the estimated uncertainties are so large that the interannual variability appears smaller than the uncertainty, indicat- ing a signal-to-noise ratio below one (review Fig. 1 left), which raises concerns about data reliability. A more refined approach could involve detrending the time series and comparing the standard deviation of the residuals to the uncertainties (review Fig. 1 right). Repeating the analysis for different time peri- ods could further clarify the data's reliability. Please check with a statistician if this test or another test is suitable. This type of analysis should be included in the manuscript and referenced in the abstract and data documentation.

Figure 1: **Signal-to-Noise ratio analysis for the 20 regions of Dussaillant et al. (in review):** (left) Boxplots illustrating the signal-to-

[Figure]

noise ratio, calculated as the ratio of the standard deviation of the mean interannual time series to the mean of the estimated total uncertainties for each glacier individually. A ratio below one indicates that the signal (interannual variability) is smaller than the noise (uncertainties). (right) Untrended signal-to-noise ratio, where a linear trend was removed from the time series to isolate the residuals. The ratio compares the standard deviation of these residuals (signal) to the total uncertainties. Values below one suggest that the residual variability is less than the uncertainties. (right) Untrended signal-to-noise ratio where a linear fit was applied to compute a trend, and then the signal was defined as the "residual" only. In both plots, values below one potentially mean that the signal is smaller than the noise (here assumed to be the uncertainties). The signal-to-noise ratios were estimated from the entire provided time series of each region. The total uncertainties were estimated by assuming complete independence of the three given uncertainty sources.

Performing a signal-to-noise analysis is a good suggestion, it is one way to give a measure of trust in the use of the data. However, a signal to noise analysis can be done in many ways depending on the use that the dataset will be given, it can be performed over various spatial or temporal scales: a specific glacier, a specific region, only during specific years, full time period etc. Further, what is considered as signal and what is considered as noise must be arbitrarily determined depending on the analysis (e.g., 1-sigma or 2-sigma uncertainties?). One might want to observe the signal over a given year related to the entire period, or a specific period and compare it to the mean uncertainty over that specific period or the entire series, etc. Further, there is also the problem that depending on the analysis performed, if something is not statistically significant it doesn't necessarily mean that is not true. Possibilities are endless and will ultimately depend on the specific use of the data. We prefer to put our effort into giving all the necessary information that individual users might eventually require to perform this analysis according to their specific needs.

In addition to this analysis, it would be valuable to include a metadata file for each glacier or grid point. This file should detail these statistics and clarify whether a glacier is "unobserved" and if the regional mean was used instead. Ideally, the metadata file would also list the glacier names used to extrapolate the MB anomaly for any given glacier.

We fully agree that a clear metadata file would be a valuable addition to the dataset and would benefit potential data users to assess whether the data is suitable for their purposes. We will provide for every region and on a glacier-by-glacier basis, a file with all the additional information that might be useful for users:

- The number of glacier MB anomalies used to calculate the mean annual glacier MB anomaly
- List of IDS glaciers MB anomalies used to calculate the mean annual glacier MB anomaly
- The period where regional glacier anomalies are used to capture annual variability.
- The IDS of any additional neighboring region glacier time series used to fill gap in time series

- The period where neighboring region glacier anomalies are used to capture annual variability.
- The mean distance of the spatially selected glacier anomalies
- The distance to the closest glacier anomalies
- The number of elevation change observations available for calibration
- The period with elevation changes observations available
- Clearly identify unobserved glaciers

**1.6 Usage of the dataset as described by the authors**

Among others, the following usages of the dataset are mentioned by the authors:

- L20: "new baseline for future glacier change modelling assessments and their impact on the world's energy, water, and sea-level budget."

- L376: "This versatility enables identification of individual years marked by significant glacier changes and the detection of zones with varying impacts. For instance, it allows to pinpoint glaciers within a region that were affected by specific annual climate variations (e.g. droughts, floods, heat waves, etc.), as well as those with a larger or smaller influence on the yearly contri- bution to hydrology and annual sea level rise."

- L391: "spatial and temporal impact of known glaciological trends and anomalies like, for example, the Andes Megadrought (Gillett et al., 2006; Garreaud et al., 2017, 2020; Dussaillant et al., 2019) or the Karakoram anomaly (Farinotti et al., 2020; Gao et al., 2020; Ougahi et al., 2022) at an unprecedented yearly temporal resolution.

- L644: "... vast potential for applications in various fields within and beyond 645 glaciology. These include international cryosphere observation intercomparison exercises; multi-Essential Climate Variable (ECV) products; serving as invaluable resources for calibrating and validating climate models; and advancing our understanding of the broader implications of glacier melt on sea levels, freshwater resources, global energy budgets, and nutrient cycling. This work opens new oppor- tunities for future assessments of global glacier mass changes at increased temporal resolutions, fostering a more detailed examination of their climate and hydrological impacts worldwide."

The manuscript suggests that the dataset can be used for a variety of applications; however, there are concerns about the practicality and reliability of these uses, especially considering the uncertainties involved. Also, some of the examples provided are not sufficiently concrete, and it is unclear how uncertainties are integrated into these applications.

Fig. 5 presents an example from Iceland, but uncertainties are not shown. It raises questions about the reliability of pinpointing individual years when uncertainties are accounted for. Iceland benefits from relatively good coverage of mass balance time series and has unique conditions due to the presence of volcaninc eruptions, and is thus not very representative of other regions.

For regions such as the Southern Andes, Subantarctic, and Antarctic Islands, where annual data before 2000 are derived from a single glacier, the added value of the dataset compared to using data from that single glacier (or the few glaciers available) needs clarification. The dataset's ability to represent these regions accurately, considering the associated uncertainties, requires a more detailed discussion.

In lines **357-366**, the manuscript discusses mass changes for regions like the Subantarctic Islands and Periphery. Since these estimates are based on extrapolated data from Echaurren Norte and a few other glaciers post-2000, the confidence in these annual estimates may be limited. A more thorough discussion on how uncertainties impact the interpretation of mass changes should be included if these estimates are to be retained in the manuscript.

In the abstract, line 20 states: "...new baseline for future glacier change modelling assessments". Do the

authors believe that glacier models should now calibrate their models to match the per-glacier annual anomalies? In my opinion, glacier models should not, because the uncertainties are way too large. Most calibration procedures just completely neglect uncertainties, and in that case, just calibrating to highly uncertain per-glacier annual MB time series would give a false estimate of confidence. While glacier modelers may benefit from having a more detailed MB time series to better constrain model parameters (such as the precipitation factor), the current dataset may not yet provide the level of precision required for direct application in glacier modeling due to its significant uncertainties. Some modeling approaches do incorporate uncertainties, such as the Bayesian calibration framework utilized by Rounce et al. (2023), which includes uncertainties from the 2000-2019 geodetic observations of Hugonnet et al. (2021). Once the uncertainty estimation approach is clarified and cross-validation is repeated with a data-denial approach, the MB time series and associated uncertainties may become valuable for such calibration methods. However, it is noteworthy that Rounce et al. (2023) did not incorporate the 5-year averaged per-glacier mass change observations from Hugonnet et al. (2021) due to the excessive uncertainties associated with these observations. A similar issue may arise with the current dataset.

Regarding the reviewer's concerns about the practicality and reliability of our dataset uses, especially considering the uncertainties involved, we argue as follows. We agree that the submitted manuscript and dataset was not clear enough to allow users to address this concern. However, we think that the changes suggested by reviewer #1, that have been addressed in the previous answers and will be considered on an updated version of the dataset and revised manuscript, will provide users with transparent information to allow them to define the practicality and reliability of their individual data usage.

To address the specific comments in this section:

**Fig 5:** The aim of this figure is to provide a visualization of the spatio-temporal resolution of the dataset (i.e. available for individual glaciers, gridded tiles, regions). Iceland was selected as an example for aesthetic reasons: it's the smallest region and easy to visualize fully in one figure. There is no intention of showing or analyzing the specific results or the uncertainties here. This is clearly shown on Fig. 3 and perfectly analyzable from the individual glacier time series and gridded product.

**Regions like the Southern Andes and Antarctic and subantarctic islands:** The issues regarding these regions have been discussed above. They will be properly addressed in the revised manuscript and figures and in the metadata of the dataset, so that users are aware of the periods where the time series are less robust.

**Baseline for future modeling:** We agree that the usefulness for modeling is unclear, as a large part of our estimates are extrapolated, rather than interpolated, due to the limited amount of glaciological time series available. We will modify our statements accordingly.

We agree that describing the dataset as a 'new baseline' is beyond our judgment. Data users are in a better position to make such a statement after testing the dataset. We will modify these statements everywhere in the revised manuscript giving them a more cautious tone as potential uses and advantages of the dataset for the modeling community.

**1.7 Data and code documentation and availability**

Firstly, it is great that the code and data are made fully available.

I have a few comments first on the provided data:

- Hosting the extrapolated / modeled per-glacier annual data on the WGMS website could po- tentially lead to misunderstandings. Given that this dataset is not purely observation-based, its direct availability at the WGMS website could result in misleading conclusions. If the decision is made to include the data directly on the WGMS website, it is essential to include a comprehen- sive "meta"- dataset and a flagging system to highlight glaciers/areas where the uncertainties are too large to extract a signal (as discussed in Section 1.5).

As mentioned above in the answer to comment 1.5, we will provide for every region and on a glacier-by-glacier basis, a .csv file with all the additional information that might be useful for users.

- The type of uncertainty documented in the dataset requires clarification. The term "uncertainty" is used generically, but it is unclear whether this refers to two times the standard error as de- scribed in Line 187, or one or two times the standard deviation (related to Sect. 1.1).

All equations represent uncertainties at $1\sigma$. Reported uncertainties in the text correspond to $2\sigma$ = 95% confidence. Therefore, the term "uncertainty" corresponds to $1\sigma$ when describing equations and $2\sigma$ for reported values.

- Currently, only individual uncertainties are provided, requiring data users to perform their own aggregation. It is strongly recommended to include a dataset with total uncertainties, as this will likely be the most utilized. Additionally, understanding the different sources of uncertainty and their origins took considerable effort. Enhanced documentation explaining these aspects would be beneficial for users.

We will add to the dataset a 4$^{th}$ file for total uncertainties for each glacier combining the individual errors from elevation change, annual anomaly and density conversion factor, as in EQ12:

$$\sigma^2_{\bar{B}_{cal,g,Y}} = \bar{\sigma}^2_{dh,\bar{B}_{cal,g,Y}} + \bar{\sigma}^2_{f_\rho,\bar{B}_{cal,g,Y}} + \bar{\sigma}^2_{\beta,\bar{B}_{cal,g,Y}} \quad (12)$$

**Total uncertainty file name:**
RRR_ gla_mean-cal-mass-change_uncertainty_total.csv

One file per RGI 1$^{st}$ order region, where RRR corresponds to the RGI-region code

- To enforce people, to look into the uncertainties, consider creating a netcdf file that has the mean time series, the total uncertainties, and a "flagging" system

The gridded netcdf files already contain the mean time series and the total uncertainties per grid point and per year. As suggested by the reviewer, we will add to this file the additional metadata (as mentioned above in the answer to comment 1.5) as attributes per grid point. This will allow users to easily flag out the dataset to consider only the values that support their specific requirements. In this case, because the netcdf format allows us to have specific metadata for every grid point and every year, a metadata index can be applied to allow users to "flag out" fields depending on index value over specific periods. This will allow, for example, to flag out only specific periods within gridded time series that are not robust, but keep the years where estimates are more robust.

- **Issues found in the per-glacier annual time series**
    - no glacier ID for Greenland, everywhere NaN values as IDs. Please update the glacier IDs for Greenland!

True and well spotted. This bug in the code has been corrected now. Thank you.

    - a bit confusing to have sometimes GLIMS_ids and sometimes RGI_ids

The Hugonnet et al. 2021 dataset used the Tielidze and Wheate (2018) inventory available from GLIMS to calculate elevation changes for in region RGI-12 Caucasus and Middle East. This decision was made because the glacier outlines from the RGI06 inventory are to a great extent erroneous in this region. The Hugonnet et al. 2021 observations were ingested to the FoG database related to the GLIM-Id, to make sure that the calculations correspond to the GLIMS glacier extents. For consistency, in order to use the elevation changes from Hugonnet et al. 2021 for the Caucasus glaciers for this assessment's calibration step, we had to consider the Tielidze and Wheate (2018) inventory as well. We think this decision makes sense. We will make sure this is clearly explained in the revised manuscript.

Comments on the github/code:

- It would be beneficial to include a README document in the GitHub repository that provides a brief overview of the functionality of each script. Such a document would guide interested users on where to find specific processes or analyses within the codebase. While the code does not need to be meticulously documented, a general overview in the README would greatly enhance the accessibility and usability of the repository.

Agreed we will add a README document in the GitHub repository providing a brief overview of the functionality of each script.

**1.8 Terminology**

- The terms "(mean) glacier (annual) anomaly" appear to be unclear and could benefit from clarification. It is recommended to use more specific terminology, such as "(mean) glacier (annual) MB anomaly" or "glaciers with glaciological MB time series". This issue is particularly evident in Figure 1, where the term is not yet explained. The phrase "glacier anomaly" may imply that the glacier itself is unusual or deviates from expected behavior, rather than referring to mass- balance measurements. Including the term "mass-balance" would help clarify the meaning and ensure consistency throughout the manuscript (e.g., line 169 and other mentions).

Agreed we will replace "(mean) glacier (annual) anomaly" with "(mean) glacier (annual) MB anomaly" everywhere in the revised manuscript text.

- What is the difference between GTN-G regions and RGI6? For instance, in Line 102, GTN-G regions are mentioned, yet later references seem to align more closely with the "usual" RGI6 regions, with the exception of the Southern Andes, which is split differently. It would be beneficial to review the references to GTN-G and RGI6 throughout the manuscript to ensure consistency. If possible, it is recommended to use only one of these terms to avoid confusion.

Agreed we will refer only to RGI regions in the revised manuscript text.

We will also change the generic term "geodetic" to more specific "DEM differencing" or "elevation change" to avoid confusion of dataset users outside of glaciology, since geodetic is a generic term with signification beyond glaciology.

**Specific comments**

As pointed out before, at this stage we will provide answers to the general and major comments by reviewers. The resulting improvements will then be further described in a complete and detailed response to this review, with individual answers to the following specific comments and a fully revised manuscript. Most of these specific comments can only be properly answered after the dataset has been reprocessed, figures updated, and posterior analysis completed. This is why they are not listed in the present initial response.

---

## Author Comment (AC2)

**Reply to reviewer #2**

We are very thankful for the anonymous reviewer's constructive review of our manuscript readability and presentation. Please find here our preliminary answers to their main comments. The resulting improvements and the response to the specific/minor comments will be detailed in our final response letter and a fully revised manuscript if the editor considers that our manuscript is appropriate for Earth System Science Data.

The document is color coded as follows:

**Black: reviewer general comment**

**Green: answers to reviewer**

In this study the authors provide mass changes for a large number of world glaciers from 1976 to 2023 by combining data sets based on in situ and space-based measurements. This new product, to be distributed by Copernicus services, will be of invaluable value for climate applications, including sea level rise and land hydrology. It will be definitely of high interest for the scientific community.

While I greatly appreciate the efforts made by the authors in combining different datasets, in performing appropriate calibration and in providing product uncertainties, I find that the manuscript requires major improvements in terms of presentation. The paper is very difficult to read. First of all, it lacks a number of general information about the data used to be understandable to non-experts. Some sections are quite technical and poorly explained. A large number of variables are not defined, and some important information is just provided in tables without explanation in the main text.

I recommend to the authors to consider my comments below and provide in a revised version a text clear enough to be appreciated by both experts and non-experts. As it stands, it is not the case.

We will provide a revised manuscript where important variables are clearly defined and explained in the main text (already in the abstract and introduction), technical sections are better explained and provide a text clear enough to be appreciated by both experts and non-experts.

**General comments:**

- The abstract and introduction are too vague and lack useful information. It is unclear in both the abstract and the introduction which data sets are considered and combined. How have they been obtained? The data section refers to the data sources given by their acronyms, but no information is provided on the methods to obtain the data. What is the proportion of in situ data and remote sensing data? Are the latter only based on ASTER DEMs as described in Hugonnet et al. (2021)? What is the spatio-temporal coverage of each data set? It is insufficient to say (as written in the abstract and introduction) that geodetic and glaciological data are used. It is unclear whether in addition to the Hugonnet et al's data, other remote sensing data are considered.

We agree the text was not fully clear in showing this specific information already in the abstract and introduction sections. Most of this is communicated later in the input datasets section and Fig. 1 in the data and methods section. We will make sure the general information about the data used is also available in the abstract and introduction and sufficiently clear to be understandable by non-experts, more specifically:

    (i)      The input data sets used with their proper references, where users can find all the information on the methods to obtain the data.

    (ii)     The proportion of in situ and remote sensing data used, their spatio-temporal coverage and a summary of their technical details (sensor sources) and references.

- While the paper mentions the percentage of glaciers considered by Hugonnet et al (96%) and the total number of glaciers is never mentioned, the percentage of glaciers considered from in situ measurements.

This information is available from Fig. 1 but as the reviewer correctly expresses, not yet in the abstract, introduction or clearly expressed in the manuscript text. We will make sure this information is already available in the introductory sections.

I suggest to rewrite the abstract and introduction to clarify these issues and provide the reader the missing information. The figures are in general too busy. The figure captions need to be extended and provide the definition of the parameters appearing in the figure.

The abstract and introduction will be re-written accordingly. We will make sure all useful information about the dataset is well explained from the beginning and the proper references are available when needed. We will re-think the figures so that they provide useful and intuitive information properly defined in the captions.

**Line-by-line comments**

As pointed out before, at this stage we will provide answers to the general and major comments by reviewers. The resulting improvements will then be further described in a complete and detailed response to this review, with individual answers to the following specific line by line comments and a fully revised manuscript. Most of these specific comments can only be properly answered after the dataset has been reprocessed, figures updated, and posterior analysis completed. This is why they are not listed in the present initial response.

---

## Author Comment (AC3)

**Reply to reviewer #1**

We are very thankful for the anonymous reviewer's detailed and thorough analysis of our dataset and constructive review of our manuscript. Most of their suggestions have been considered in our revised manuscript. We acknowledge that the proposed changes have made the dataset more robust on its algorithm and uncertainty assessment and the revised manuscript clearer and better expressing the strengths and limitations of the dataset to potential users. We here respond point by point to all major and specific concerns raised by reviewer #1.

The document is color coded as follows:
**Black: reviewer comment**
Green: answers to reviewer
Blue: extracts from the revised manuscript

This study combines in-situ glaciological annual mass-balance observations with remotely sensed sur- face elevation data to provide an annual time series of individual glacier mass changes over the last decades to a century. Based on the assumption that glacier mass-balance anomalies are similar be- tween neighbouring glaciers, annual observed glaciological mass-balance anomalies are extrapolated to all glaciers globally. These anomalies are combined with several geodetic samples to calibrate a mean annual mass change time series and its respective uncertainties.

This extensive data merging and extrapolation study presents an interesting analysis. However, major revisions are necessary before the publication of this dataset to avoid potential misuse by future users.

First of all, there are several major issues, questions and obscurities in the approach of how the un- certainties were estimated (see Sect. 1.1, 1.2) and how the cross-validation was done and analysed (see Sect. 1.3). Another critical aspect is the missing discussion about uncertainties at the glacier scale in the paper itself. The large per-glacier uncertainties become apparent only by checking the dataset itself. Many data users might not be able to use the per-glacier dataset as the uncertainties surpass the signal in many cases (see Sect. 1.4, Sect. 1.5).

Another essential aspect is to communicate clearly that this dataset is not a purely observed dataset since it was created by extrapolation. When extrapolating, predictions are made about unobserved glaciers/years based on an underlying assumption or rule. This dataset is based upon a model of belief of how the system behaves. Upon publication on the WGMS website, the dataset might be misused and falsely interpreted as observations. Most data users neglect or do not include the uncertainties in their frameworks. Therefore, it is important that data providers clearly state the limitations of their dataset. It may imply, for example, adapting the title and the analysis (see Sect. 1.6), and also "flagging" respective regions or glaciers by adding some "metadata" to the data (see Sect. 1.7). In general, the manuscript should focus much more on the uncertainties that vary regionally and temporally and showcase for what use case the data can be used and for what the data might not be useful.

There are several steps in the manuscript that I find unclear, particularly concerning the statistical analysis. I believe these issues need to be addressed by the author team and eventually reviewed by a statistical expert (if not already done). Additionally, the paper and data require a substantial rewrite before they can be reviewed properly. Consequently, I am only able to partially evaluate the manuscript and the dataset at this time. Only a revised version that incorporates or addresses my comments will enable me to fully assess the study and the dataset's added value.

My major comments are summarised in the 'General Comments' (Sect. 1). Line-by-line comments are in the 'Specific comments' (Sect. 2). After each manuscript section, the respective figures and tables and their captions are commented there as well.

**General comments**

**1.1 Standard error**

The manuscript uses two times the standard error as the uncertainty measure for the glacier mass balance anomalies. The standard error of the mean describes the uncertainty in estimating the mean, essentially providing the precision of the mean. In contrast, the standard deviation describes the variability of individual points around the mean, indicating the spread of the mass-balance anomalies. This distinction is visually explained on the following website: https://seaborn.pydata.org/tutorial/error_bars.html. In this context, the standard deviation would indicate how much the glacier anomalies deviate from their mean across different locations and years, which is likely of primary interest to data users. Using the standard error because it "allows" years with more observations to have smaller uncertainties is, to my knowledge, an uncommon approach.

As the reviewer correctly defines, the standard error of the mean describes the uncertainty in estimating the mean, while the standard deviation can refer to any measure of sample variability or spread. However, it seems unclear to the reviewer that the two terminologies are not mutually exclusive. By definition, the standard error is also a standard deviation, namely the standard deviation of all possible means. During uncertainty analysis, a standard deviation implicitly refers to the associated estimate (which can be a mean, or other), and is thus always a measure of uncertainty. We confirm that this is also the case everywhere in our study.

We agree with the reviewer that the justification of our uncertainty methods was sometimes convoluted, and that the language was sometimes inconsistent. In the revised document, we now explain the methods more clearly keeping a consistent language. We relate the uncertainties to their physical meaning for a new source, and justify how they propagate from other sources.

Additionally, the standard deviation ap- pears to be used later for other uncertainties (e.g., $\sigma B_{glac}$, density conversion factor uncertainty). The uncertainties are combined by adding the standard error and standard deviation together, which is probably also not a standard practice. Are there references that justify the approaches described in this paragraph?

We refer to the answer above on the fact that standard error and standard deviation are not mutually exclusive, and instead refer to the same concept in uncertainty analysis. Standard deviations always refer to the variability of the estimate (whether it is a mean of samples, a modelling estimate, a temporal or spatial sum for aggregating to larger regions).

For instance, the uncertainty in density conversion of Huss (2013) mentioned by the reviewer is a standard deviation of estimates of density conversions, derived from a modeling exercise. As we do not have any other prior knowledge and use a constant mean density conversion globally, also corresponding to Huss (2013), this standard deviation corresponds to our uncertainty in density conversion at the glacier scale.

The equation in line 193 is unclear and raises further confusion. The equation states that the uncertainty is two times the sum of the different standard deviations of the individual N selected annual glacier mass balance anomalies, divided by the square root of the number of observations. It appears these uncertainties correspond to the individual lines in Fig. 2b. However, it is unclear why a standard deviation is calculated for each glacier anomaly (*i*), which is then summed. Additionally, the explanation provided in the text (line 188) does not seem to align with the equation on line 193. To clarify, the code was briefly reviewed at https://github.com/idussa/mb_data_crunching/blob/c9ab8e10198583d0cb2fc1de809e01e4bd5fbca3/2.1_spatial_anomalies/calc_global_gla_spatial_anom.py#L505. Based on this, it seems the standard deviation is computed over the observations, not summed, which conflicts with the equation in line 193. The code then appears to calculate a mean over another variable. It also seems that the error is first calculated for every "line" shown in Fig. 2c, leading to an average in the script. However, there does

not appear to be any summation applied, suggesting a possible discrepancy in the equation on line 193 or a misunderstanding of the correct line of code. Please clarify this process. Moreover, the rationale behind using a factor of two for the standard error is unclear. Please clarify the reasoning behind this choice (described further in Sect. 1.3).

The issue in Equation 4 raised here by the reviewer is no longer relevant since we decided to replace the inverse-distance weighting (IDW) spatial interpolation (for which there is no integrated error propagation) by a kriging spatial interpolation (that includes error propagation natively). This is explained further below.

Another point of concern is that the current approach results in glacier mass-balance (MB) anomaly un-certainties that only depend on the amount of included glaciers and their differences in the anomaly. I suggest that a mass balance anomaly from a glacier located further away results in larger uncertainties compared to one that is nearer. Is this accounted for in the uncertainty estimates? Do the uncer-tainties increase if only distant glaciers are available? In some cases, this might occur naturally if the distant glaciers are not clustered, leading to significant differences in MB anomalies and, consequently larger uncertainties. However, if the available glaciers with MB time series are far away but clustered closely together, could the assessed uncertainties be underestimated? Is there any algorithm in place to prevent this potential underestimation?

We agree with the referee that our methodology lacks a way to capture the varying errors with distance to the measurement used for interpolation. The uncertainty in the mean glacier mb-anomaly described in the submitted manuscript depended only on the number of glaciers included, their individual uncertainties, the distance of the selected anomalies used (inverse-distance weighting spatial interpolation, for which there is no integrated error propagation) and the differences of the individual glacier anomalies as standard error or a measure of the uncertainty in estimating the yearly means.

To solve this issue, we replaced the inverse-distance weighting (IDW) spatial interpolation with a kriging spatial interpolation. IDW was not the best suited method for an error-prone statistical analysis like ours because there is no integrated error propagation. Kriging instead, replaces the weight function by an empirically determined one based on the data covariance, and therefore natively supports error propagation. As stated in our preliminary answers (ESSD online discussion), we worried that the kriging implementation would not be sufficiently efficient computationally to run on our dataset in a reasonable time. However, not only did we successfully implement it but also, it provides a much robust estimation of both the mean annual mb-anomaly and its uncertainties. We refer to L182-L205, EQ 3 and 4 and Fig. 3 in the manuscript where the kriging method is explained.

Not only is kriging empirically-based (uses distance-weights that are not arbitrary but estimated from the data itself), it also provides robust errors that vary with the distance to measurements. For instance, a point interpolated right next to a measurement will have the same error as that of the measurement. In contrast, a point located further away than the maximum correlation length of the data from a measurement will have an error equal to the variability of the samples (maximum possible error). Additionally, kriging works best on second-order stationary data (stationary mean and variance), which is well verified in the case of our anomalies that are centered on zero, with similar variance within each region. For this reason, on this type of data, kriging is coined as the "best linear unbiased estimator".

In summary:
- Kriging is known to provide robust theoretical errors under assumptions of stationarity solving the arbitrary uncertainties issue raised by the referee.
- Kriging depends on the distance to the observations, with larger errors further away from observations solving the distance-dependency of uncertainties raised by the referee.
- The Kriging distance-weight function is robust as it is empirical (estimated from the data), and based on covariance to minimize variance (thus coined "best unbiased interpolator"). Solving the issue of an arbitrary weight function.

As the reviewer suggests, in our revised dataset and manuscript we now make sure users are aware of where our dataset is better or less constrained, not only through the uncertainties but also through providing the necessary metadata. We additionally provide a clearer illustration of the results in the manuscript figures and discussion to flag the periods and regions where the dataset is less constrained. As an example: the reduced robustness of the mean calibrated time series during filled-up past years where neighboring glaciers have been used, are now evident not only because of their larger uncertainties, but also clearly delineated in the manuscript main Fig. 4 with grey lines and shading.

**1.2 Uncertainties/Error propagation**

For the analysis of per-glacier mass balance uncertainties, the law of random error propagation is frequently used. It would be beneficial to explain, in each instance, why it is considered valid to assume that the errors are completely uncorrelated. Specific examples where random error propagation might not be valid or should at least be discussed are noted in the specific comments (e.g., **L191, L255-257, L269, L276, L281**). It may also be necessary to mention that assuming complete independence could lead to underestimating the actual uncertainties.

We agree the text was not fully clear in showing where assumptions of correlation or no correlation are applied between sources, as we instead primarily focused on spatial correlations. The reason behind this difference in focus is that the assumption of correlation between sources has negligible impact compared to the spatial correlation of errors within the same source. This is because we only have 3 main sources of uncertainties, while we have 200,000 glaciers distributed spatially. For instance, for a sum (of volume changes), assuming for the sake of the example that all errors have the same magnitude, if 3 sources of errors are combined as uncorrelated or fully correlated affects the total uncertainty by a factor of 3/sqrt(3) = 1.7. For 200,000 glaciers, if errors are propagated spatially assuming they are uncorrelated or fully correlated, it affects the total uncertainty by a factor of 200000/sqrt(200000) = 447. This is why we focus on estimating spatial correlation to constrain errors more robustly in this study, while we put less energy in estimating the correlation between sources that has little impact.

Nonetheless, we edited the manuscript to clarify the physical meaning of each term and the justification behind assumptions of error propagation as either uncorrelated, fully correlated, or correlated to a certain degree (for spatial correlations) at every instance. Below, our response and/or new text to the five cases explicitly mentioned by reviewer #1 (L191, L255-257, L269, L276, L281)

**Case 1: referring to L191 in the previously submitted manuscript.**
No longer relevant due to change to kriging

**Case 2: referring to L255-257 in the previously submitted manuscript.**
We decided to keep this paragraph as is, since it explains that spatial correlation is considered between three error sources: elevation change, density conversion and annual anomaly prediction.

**Case 3: referring to L276 in the previously submitted manuscript.**
New text, L297-L302 in the revised manuscript:

We propagate the uncertainty in the specific regional mass change, the uncertainty in the regional area (Paul et al., 2015) and the uncertainty in the area change considering them uncorrelated. Errors in the area stem mostly from remote sensing delineation errors, while errors in area change stem from a lack of multi-temporal outlines to constrain area change. They are largely uncorrelated with error sources described above for elevation change, glaciological measurements and anomalies. However, elevation change estimates usually already consider errors in area at the scale of each glacier, so we might conservatively be double counting these.

$$\sigma_{\Delta M_{R,Y}} = |\Delta M_{R,Y}| \sqrt{\left(\frac{\sigma_{B_{R,Y}}}{B_{R,Y}}\right)^2 + \left(\frac{\sigma_{S_R}}{S_R}\right)^2 + \left(\frac{\sigma_{\Delta S_{R,Y}}}{\Delta S_{R,Y}}\right)^2} \qquad (19)$$

**Case 4: referring to L281 in the previously submitted manuscript.**
New text in L320-323 in the revised manuscript:

To simplify the combination of annual values into long term trends or cumulative annual values, we assume the yearly uncertainty to be independent of other years. This is true for glaciological measurement, having an independent uncertainty estimation for each individual year of the time series, but not for the elevation change measurements, where uncertainties are correlated over the years of the survey period.

Regarding **L269, eq. 16**, it is stated that the errors are assumed to be completely correlated at regional scales, but the equation suggests that complete independence is assumed (as indicated by summing the square roots). Which assumption was actually applied in the results? This was not clear from the code.

**Case 5: referring to L269 in the previously submitted manuscript.**
As explained above, we follow the assumption that correlation between sources has a negligible impact compared to the spatial correlation of errors within the same source. For this reason, we express that, after applying spatial correlation within error sources, we combine all sources of error propagated at the regional-scale as independent.

**1.3 Leave-one-out cross validation**

Applying a leave-one-out cross-validation is crucial, and it is great that this validation is performed by using geodetic data available for all glaciers. However, given the nature of the reference glaciers, there are concerns about the validity of the conclusions drawn, such as the claim in line 452 that the "leave-one-out cross-validation results prove that our algorithm can capture the annual variability of individual glaciers."

We clarified these claims in the revised manuscript making them less absolute.

As noted in lines 454-456, a major issue arises from the fact that the approach may work well for ref- erence glaciers, often located in regions with nearby glaciers with mass-balance time series. Therefore, evaluating the metrics for these glaciers may not be representative. For example, removing Hintereis- ferner still leaves the nearby Kesselwandferner, which could skew the results. To provide robust esti- mates of the method's performance, a "data-denial/blocking" cross-validation approach is necessary. This involves analyzing how well the algorithm performs when assuming that, for instance, Hintereis- ferner has only one or two randomly selected glacier anomalies located far away, such as in the French Alps. Repeating this analysis across many glaciers and examining how the performance metrics change, as illustrated in Fig. 6, would provide a clearer understanding of the method's robustness. Additionally, evaluating how performance metrics vary with the number of considered glaciers would be valuable.

Please evaluate the approach with a larger glacier sample and the data-denial experiment to better demonstrate the dataset's robustness or non-robustness.

We appreciate the reviewer's idea, and we agree that performing a so-called "data-denial/blocking" cross-validation approach would add more insight into the robustness of our estimates. We performed this analysis and added the results in a new section in the revised manuscript: 5.3 Leave-block-out cross validation. For consistency with the literature on spatial statistics, we chose to use the term leave-block-out cross-validation analysis. The process is fully explained L514-L520, results are displayed in Fig. 9 and discussed in respective section in the revised manuscript.

The exercise is similar to the leave-one-out cross validation, with the difference that, instead of removing only the reference/benchmark glacier mass-balance time series, we remove all the glacier mass-balance time series surrounding the reference/benchmark glacier for increasing distances ranges. The mean and standard deviation of the residuals are calculated at every distance step, to assess potential systematic errors (with the

mean), and the magnitude of random errors (with the standard deviation). We performed this analysis over a sample of 74 reference and benchmark glaciers.

The revised manuscript states (L535-L544):

There is no apparent influence of the distance on systematic errors in the calculated glacier-wide leave-block-out mb justified by absence of trends in Fig. 9a. In these cases, the slight systematic errors will mostly depend on whether the reference series are reanalysed or not, and the quality of the elevation change used for calibration. As expected, random errors (residual S) increases as the mean glacier anomaly is calculated from a more distant sample (Fig. 9b), from 0.5 m w.e. for nearby time series up to 1 m w.e. for series located farther than 2000 km. Importantly, in most cases both systematic and random errors are captured by the mean calibrated annual mass-change uncertainty at σ independent of the distance of the sample (Fig. 9c and 9d). This means that our predicted uncertainties reflect the true variability in the residuals, and that our model is providing realistic confidence intervals for the mean annual mass-balance anomaly predictions. S is larger than σ only in some few cases with distances to the closest glacier > 500km, but the large spread suggests this is coming from the randomness of the predictions.

Another consideration is the selection of glaciers for cross-validation. Why are e.g. Echaurren Norte and other WGMS reference or benchmark glaciers not chosen for the cross-validation? Including all glaciers with at least 10 years of observations could allow for a more comprehensive analysis, even if some glaciers have fewer years of data and are not validated. This inclusion would enable assessment in regions without reference glaciers and ensure that performance metrics are not skewed by a few well-sampled regions.

Glaciological time series are subject to biases inherent to the glaciological method. The WGMS highly recommends reference (+30 years) and benchmark (+10 years) glaciers glaciological time series to be reanalyzed every 10 years by calibrating them with long term trends derived from high resolution elevation change measurements (Zemp et al., 2013). We intentionally chose to perform the leave-one-out cross validation experiment with a selected list of reference and benchmark glaciers known to have been reanalyzed. These series stand as the only ground truth available for validation of our global assessment. The decision of not using all glaciological time series in the experiment was justified by reducing the risk of validating over potentially erroneous "truths".

There exist 74 reference and benchmark glaciers in the WGMS database, 32 of them are proven to be reanalyzed (those used in the previously submitted manuscript leave-one-out exercise). We now increased the sample of the leave-one-out and the leave-block-out cross validation to the 74 reference and benchmark glaciers.

Results for the leave-one-out cross validation experiment are summarized in Fig.7 and discussed in section 5.2 of the revised manuscript. The revised figure 8 is now divided into two, Fig. 8.1 comparing the leave-one-out cross validation results on selected reference glaciers (>30 years of data), and Fig. 8.2 comparing the leave-one-out cross validation results on selected benchmark glaciers (>10 years of data).

We note that we changed the measured statistics in all figures to the Mean error (ME) and standard deviation of the residuals (S) between the calculated leave-one-out cross validation time series, as response to some confusion in the parameters chosen and other concerns raised by referee #1 that will be discussed later in this document.

We add here the same as Fig.6 but considering only the reanalyzed glaciers (n=32) for the referee and editor to confirm how little statistics differ.

[Figure]

**Figure 6: Leave-one-out cross-validation results and statistics over 32 reanalyzed glaciological time series from reference and benchmark glaciers.** Mean error (ME) and standard deviation of the residuals (S) between the calculated leave-one-out cross validation time series and the 32 reference and benchmark glacier time series. **(a)** Yearly results from the leave-one-out calculated annual mass changes against the reference and benchmark glaciers annual mass changes. **(b)** Long term trends (period 1976-2023) from the calculated leave-one-out time series against long term trends for reference and benchmark glaciers. **(c)** Amplitude of the annual variability measured as the time series variability STD (not to be confused with the standard deviation of the residuals noted S) for the period 1976-2023 from the calculated leave-one-out time series against the reference and benchmark glaciers time series. **(d)** ME and **(e)** S of residuals for each reference and benchmark glacier against the estimated uncertainty of the mean calibrated estimate for the same glacier at σ. In a, b, c and d, each value corresponds to one of the 32 reference and benchmark glaciers used for cross-validation, symbols correspond to the glacier regions to which they belong. The size of the symbol is related to the area of the glacier.

Regarding validation, if direct glaciological mass-balance observations were not included in the calibration due to the lack of data over the baseline period 2010-2019, it would be beneficial to use these observations for additional validation if possible.

We disagree with the referee's comment for the same reasons stated above. These time series do not correspond to reference or benchmark glaciers and therefore might be biased due to the lack of reanalysis. To reduce the risk of validation against biased measurements, we intentionally exclude these glaciers and all non-reanalyzed time series from our cross-validation experiment.

Finally, the claim that cross-validation shows the uncertainty estimates are on the "conservative" side and that the dataset has realistic uncertainties needs clarification. The assessment of whether the cross-validation errors are sufficiently small is based on comparing them to the assumed uncertainties of the dataset. However, this approach may allow for "inflating" the uncertainties until they encompass the cross-validation errors.

The reviewer's comment is somewhat unclear, and we interpret that their statement "this approach may allow for inflating the uncertainties" refers to the practice of iterating (i.e. making changes) on the

uncertainty calculation until they agree with the cross-validation results. If this is what is meant, we disagree with the reviewer that this is a potential issue. Iterating to improve theoretical uncertainty quantification until it matches empirical uncertainty estimates from the cross-validation is a good scientific practice, and the very purpose of cross-validation. It helps identify potential gross errors (mistakes in implementation) and ensures a realistic estimation of uncertainties. This is true as-long-as the cross-validation is representative of the conditions in which the methodology is applied for the whole dataset. In this case, as pointed out by the reviewer, we did not sufficiently discriminate estimates spatially during the leave-out process. The addition of the new leave-block-out cross-validation proposed by the reviewer should further improve this.

We note however that the cross-validation cannot identify some sources of systematic errors already present in the estimates used (as they are validated against themselves), only the ones that might be introduced by our methodology.

In relation to Fig. 6d, there is confusion about the comparison presented. If the y-axis represents $\sigma var_{\beta Y}$ from line 193 (i.e., two times the standard error) and the x-axis shows the mean absolute error, there seems to be a comparison of two different types of errors. The metrics being compared are different in nature: the mean absolute error is calculated differently from the standard error. It is unclear whether these two metrics can be directly compared. Should the x-axis not display the RMSE (Root Mean Squared Error, i.e., typically larger than the MAE), as it involves estimating squared differences, which aligns more closely with the standard deviation? The standard deviation is typically used to measure the spread of errors around the mean, and RMSE would be more appropriate for comparing with it. Comparing RMSE on the x-axis with the standard deviation from the calibration on the y-axis would allow for a more consistent evaluation of prediction error (RMSE) relative to the inherent variability or spread of errors (standard deviation). Please verify this approach (if possible with a statistician) and provide a clear explanation for the chosen comparison, including its validity.

The confusion of the reviewer here is completely understandable. Thanks to this comment we were able to detect that there was also confusion among coauthors in terms of the best metrics to analyze the cross-validation results. We have now agreed upon using only the mean of residuals and the standard deviation of residuals as metrics to quantify potential systematic errors and random errors within the cross-validation results, respectively. We do not use the mean absolute error or the RMSE anymore, since they don't provide any additional information. We clarified the meaning of this parameter in the revised manuscript text discussion and corrected the panels from Fig. 7, Fig. 8.1, Fig. 8.2, Fig.9 accordingly.

**1.4 Limited "glacier anomalies" for specific periods or regions**

The manuscript mentions a threshold of at least three glaciers with mass balance anomalies as necessary. However, it appears that in regions such as the Southern Andes or Subantarctic and Antarctic Islands, only Echaurren Norte is used as a source of MB anomalies before the year 2000, and after 2000, only two to three glaciers are included. Are these sources truly representative for all the RGI regions in these areas?

The Southern Andes is a special case because there is only one long-term and continued glaciological time series available for the Central Andes: Echaurren Norte (1976-2023, the only reference glacier in the entire Southern Hemisphere) and only one sufficiently long glaciological time series for the Patagonia region: Martial Este (2001-2023). Both these regions are extremely different in climatology, and we decided to process them differently, considering the 2nd Order RGI regions for the Southern Andes, dividing Patagonia from the Central Andes at 46S. We intentionally tuned the Echaurren Norte anomaly as the glacier mean annual mb-anomaly for the Central Andes, the Martial Este anomaly for the Patagonia Andes. The mean annual mb-anomaly uncertainty for both regions is calculated by kriging accounting for the distance.

Past assessments of glacier annual mass change (Zemp et al. 2019) used the full annual signal from Echaurren Norte "as is" to estimate glacier mass changes in the entire Andes, as well as all time-series in the Southern Hemisphere: New Zealand, Low Latitudes, Antarctic and subantarctic. In our study, we decided to include

the Echaurren's full time series only for the Central Andes, where it belongs and where it is more likely to be representative of the climatology. For Patagonia, New Zealand, Low Latitudes, Antarctic and subantarctic we only include the Echaurren time series to "fill up" past period years (e.g. from 1976-2000 for ANT, 1976-2001 for NZL and SAN (Patagonia) and 1976-1992 for TRP, before the glaciological observational period of each independent regional sample). To reduce the effect of possible climatic differences between each neighboring regions, the amplitude of the Echaurren glacier anomaly over these "fill up" past period years is normalized to the amplitude of the mean glacier anomalies of the regional sample. The reduced robustness of the mean calibrated time series during these past filled-up years is apparent on the larger uncertainties in the time series. We make sure to clearly show these "fill up" past period years in the revised manuscript main Fig. 4 with grey lines and shading. Further, we added to the metadata files in every region both the number of anomalies used in each case, their ids and the "filled up" period they cover.

We acknowledge that considerable uncertainty remains in these undersampled regions, but we are confident that our present approach is better constrained than past studies, and it is the only possible way to go back in time with annual temporal variability for these regions, considering the lack of past period observations in the Southern Hemisphere.

Similarly, in the Alps, the MB time series are extracted only from Claridenfirn and Silvretta. To my knowledge, these observations are based on very few stakes during the first 40 years (only two stakes?), which likely introduces higher uncertainty compared to more recent MB time series (e.g., Huss et al., 2021, https://doi.org/10.3929/ethz-b-000474039; Huss et al., 2017, https://doi.org/10. 3189/2015JoG15J015). Was this increased uncertainty in the past data accounted for in your analysis? The dataset and the estimated individual glacier MB time series show relatively small uncertainties for Central Europe in the period when anomalies are sourced from only two glaciers. Please clarify how these factors were addressed.

The increased uncertainty in the past is well accounted for in our analysis. This effect is apparent and well represented in our resulting uncertainties for, e.g. Central Europe, where uncertainties are two times larger before 1952 compared to the better constrained period after 2000. This is also apparent in other regions during periods where only few, or neighboring region glacier time series are used. In general, our results achieve a consistently good representation of the uncertainties across all regions, with larger uncertainties in regions and periods with small glaciological samples or where neighboring glacier time series are used for filling up past years. And vice versa, lower uncertainties in regions and periods with large glaciological samples.

**1.5 Uncertainty analysis - signal to noise ratio**

The manuscript would benefit from a more comprehensive uncertainty analysis that examines how uncertainties vary between regions, glaciers, and time periods. This analysis should include a review of the number of glacier mass balance anomalies used, the covered years, their distances, and the amount of geodetic samples. Such information is crucial for potential data users to assess whether the data are suitable for their purposes.

In addition to this analysis, it would be valuable to include a metadata file for each glacier or grid point. This file should detail these statistics and clarify whether a glacier is "unobserved" and if the regional mean was used instead. Ideally, the metadata file would also list the glacier names used to extrapolate the MB anomaly for any given glacier.

While reviewing the paper and examining the data, several questions arise: Where is the annual time series valuable and usueful, and where should caution. A quantitative analysis with statistical tests would be useful for addressing these questions (more discussion on usage cases stated by the authors is in Sect. 1.6).

One potential approach could be a "signal-to-noise" ratio test, where the standard deviation of the mean interannual MB time series is divided by the mean uncertainties (also represented as a standard deviation). If this ratio exceeds one, it suggests that the data adds value; if below one, it implies that uncertainties might overshadow the signal. While this simple ratio is not a rigorous statistical test, it can provide initial

insights into data usability. For most glaciers outside Central Europe, the estimated uncertainties are so large that the interannual variability appears smaller than the uncertainty, indicat- ing a signal-to-noise ratio below one (review Fig. 1 left), which raises concerns about data reliability. A more refined approach could involve detrending the time series and comparing the standard deviation of the residuals to the uncertainties (review Fig. 1 right). Repeating the analysis for different time peri- ods could further clarify the data's reliability. Please check with a statistician if this test or another test is suitable. This type of analysis should be included in the manuscript and referenced in the abstract and data documentation.

Figure 1: **Signal-to-Noise ratio analysis for the 20 regions of Dussaillant et al. (in review):** (left) Boxplots

[Figure]

illustrating the signal-to-noise ratio, calculated as the ratio of the standard deviation of the mean interannual time series to the mean of the estimated total uncertainties for each glacier individually. A ratio below one indicates that the signal (interannual variability) is smaller than the noise (uncertainties). (right) Untrended signal-to-noise ratio, where a linear trend was removed from the time series to isolate the residuals. The ratio compares the standard deviation of these residuals (signal) to the total uncertainties. Values below one suggest that the residual variability is less than the uncertainties. (right) Untrended signal-to-noise ratio where a linear fit was applied to compute a trend, and then the signal was defined as the "residual" only. In both plots, values below one potentially mean that the signal is smaller than the noise (here assumed to be the uncertainties). The signal-to-noise ratios were estimated from the entire provided time series of each region. The total uncertainties were estimated by assuming complete independence of the three given uncertainty sources.

Performing a signal-to-noise analysis is a good suggestion, it is one way to give a measure of trust in the use of the data. However, a signal to noise analysis can be done in many ways depending on the use that the dataset will be given, it can be performed over various spatial or temporal scales: a specific glacier, a specific region, only during specific years, full time period etc. Further, what is considered as signal and what is considered as noise must be arbitrarily determined depending on the analysis (e.g., 1-sigma or 2-sigma uncertainties?). One might want to observe the signal over a given year related to the entire period, or a specific period and compare it to the mean uncertainty over that specific period or the entire series, etc. Further, there is also the problem that depending on the analysis performed, if something is not statistically significant it doesn't necessarily mean that is not true. Possibilities are endless and will ultimately depend on the specific use of the data. We prefer to put our effort into giving all the necessary information that individual users might eventually require to perform this analysis according to their specific needs.

In addition to this analysis, it would be valuable to include a metadata file for each glacier or grid point. This file should detail these statistics and clarify whether a glacier is "unobserved" and if the regional mean was used instead. Ideally, the metadata file would also list the glacier names used to extrapolate the MB anomaly for any given glacier.

We fully agree that a clear metadata file is a valuable addition to the dataset and will benefit potential data users to assess whether the data is suitable for their purposes.

We provide now for every region and on a glacier-by-glacier basis (i.e. for every single glacier in the RGI inventory), a .csv file with the additional metadata information as individual columns described in the following table and a metadata_README file:

| Col_id | column_name | description |
|---|---|---|
| 0 | RGIId | Glacier identifier from RGI60 - every file includes all RGI glaciers in the region (GLIMS_ID identifier for Caucasus region 12) |
| 1 | REGION | RGI 1st order region code where the glacier belongs |
| 2 | CenLon | Glacier centroid Longitude (WGS 84 – EPSG:4326) extracted from the RGI60 glacier outline geometry (GLIMS outlines for Caucasus region 12) |
| 3 | CenLat | Glacier centroid Latitude (WGS 84 – EPSG:4326) extracted from the RGI60 glacier outline geometry (GLIMS outlines for Caucasus region 12) |
| 4 | Area | RGI60 glacier geometry area in km2 (GLIMS area for Caucasus region 12) |
| 5 | WGMS_ID | WGMS-FOG identifier for the given RGI glacier - glaciers with no WGMS_ID = no_WGMS_ID |
| 6 | N_gla_anom_used | The number of glacier mb anomalies used to calculate the mean annual glacier mb anomaly |
| 7 | ID_gla_anom_used | List of IDs of the spatially selected glaciers mb anomalies used to calculate the mean annual glacier mb anomaly (in WGMS_ID) |
| 8 | min_dist_gla_anom | Minimum distance of the spatially selected glaciers mb anomalies sample (i.e. distance to the closest glacier mb anomaly used to calculate the mean annual glacier mb anomaly) |
| 9 | max_dist_gla_anom | Maximum distance of the spatially selected glaciers mb anomalies sample (i.e. distance to the farthest glacier mb anomaly used to calculate the mean annual glacier mb anomaly) |
| 10 | mean_dist_gla_anom | Mean distance of the spatially selected glaciers mb anomalies sample (i.e. the mean of the distance of all glacier MB anomaly used to calculate the mean annual glacier MB anomaly) |
| 11 | std_dist_gla_anom | Standard deviation of the distances of the spatially selected glaciers MB anomalies sample (i.e. the std of the distance of all glacier MB anomaly used to calculate the mean annual glacier MB anomaly) |
| 12 | period_reg_obs | Period of the glacier time series where the sampled regional glacier anomalies are used to capture annual variability |
| 13 | N_gla_anom_neighbour_reg | Number of additional neighboring region glacier anomalies used to fill the annual variability over past years in time series |
| 14 | ID_gla_anom_neighbour_reg | List of IDs of the additional neighboring glacier anomalies used to fill the annual variability over past in time series (in WGMS_ID) |
| 15 | period_neighbour_reg_obs | Period of the glacier time series where neighboring region glacier anomalies are used to capture the annual variability |

| | | (i.e. period of filled-up past years) |
|---|---|---|
| 16 | **N_geo_obs** | Number of elevation change (geodetic) observations available for the glacier for calibration |
| 17 | **period_geo_obs** | Periods where elevation change (geodetic) observations are available for the glacier |

*Columns 6 to 17, unobserved glaciers = no_obs
*Columns 13-15, glaciers where no additional neighboring region glacier time series are used = N/A

We revised Table 4, adding the following details to the data filename sections for **Dataset 1: Individual glacier annual mass change time series.**

**Metadata file name:**
>    RRR_rgi-region-longname_metadata.csv
>    README_metadata.txt
>
>    One file per RGI 1$^{st}$ order region, where RRR corresponds to the RGI region code, and rgi-region-longname to the RGI region long name.

We note that we have not yet added the metadata to the netcdf files of **Dataset 2: Global gridded annual glacier mass changes.** This can be done later under request of the editor.

**1.6 Usage of the dataset as described by the authors**

Among others, the following usages of the dataset are mentioned by the authors:

- L20: "new baseline for future glacier change modelling assessments and their impact on the world's energy, water, and sea-level budget."

- L376: "This versatility enables identification of individual years marked by significant glacier changes and the detection of zones with varying impacts. For instance, it allows to pinpoint glaciers within a region that were affected by specific annual climate variations (e.g. droughts, floods, heat waves, etc.), as well as those with a larger or smaller influence on the yearly contri- bution to hydrology and annual sea level rise."

- L391: "spatial and temporal impact of known glaciological trends and anomalies like, for example, the Andes Megadrought (Gillett et al., 2006; Garreaud et al., 2017, 2020; Dussaillant et al., 2019) or the Karakoram anomaly (Farinotti et al., 2020; Gao et al., 2020; Ougahi et al., 2022) at an unprecedented yearly temporal resolution."

- L644: "... vast potential for applications in various fields within and beyond 645 glaciology. These include international cryosphere observation intercomparison exercises; multi-Essential Climate Variable (ECV) products; serving as invaluable resources for calibrating and validating climate models; and advancing our understanding of the broader implications of glacier melt on sea levels, freshwater resources, global energy budgets, and nutrient cycling. This work opens new oppor- tunities for future assessments of global glacier mass changes at increased temporal resolutions, fostering a more detailed examination of their climate and hydrological impacts worldwide."

The manuscript suggests that the dataset can be used for a variety of applications; however, there are concerns about the practicality and reliability of these uses, especially considering the uncertainties involved. Also, some of the examples provided are not sufficiently concrete, and it is unclear how uncertainties are integrated into these applications.

Fig. 5 presents an example from Iceland, but uncertainties are not shown. It raises questions about the reliability of pinpointing individual years when uncertainties are accounted for. Iceland benefits from

relatively good coverage of mass balance time series and has unique conditions due to the presence of volcaninc eruptions, and is thus not very representative of other regions.

For regions such as the Southern Andes, Subantarctic, and Antarctic Islands, where annual data before 2000 are derived from a single glacier, the added value of the dataset compared to using data from that single glacier (or the few glaciers available) needs clarification. The dataset's ability to represent these regions accurately, considering the associated uncertainties, requires a more detailed discussion.

In lines **357-366**, the manuscript discusses mass changes for regions like the Subantarctic Islands and Periphery. Since these estimates are based on extrapolated data from Echaurren Norte and a few other glaciers post-2000, the confidence in these annual estimates may be limited. A more thorough discussion on how uncertainties impact the interpretation of mass changes should be included if these estimates are to be retained in the manuscript.

In the abstract, line 20 states: "...new baseline for future glacier change modelling assessments". Do the authors believe that glacier models should now calibrate their models to match the per-glacier annual anomalies? In my opinion, glacier models should not, because the uncertainties are way too large. Most calibration procedures just completely neglect uncertainties, and in that case, just calibrating to highly uncertain per-glacier annual MB time series would give a false estimate of confidence. While glacier modelers may benefit from having a more detailed MB time series to better constrain model parameters (such as the precipitation factor), the current dataset may not yet provide the level of precision required for direct application in glacier modeling due to its significant uncertainties. Some modeling approaches do incorporate uncertainties, such as the Bayesian calibration framework utilized by Rounce et al. (2023), which includes uncertainties from the 2000-2019 geodetic observations of Hugonnet et al. (2021). Once the uncertainty estimation approach is clarified and cross-validation is repeated with a data-denial approach, the MB time series and associated uncertainties may become valuable for such calibration methods. However, it is noteworthy that Rounce et al. (2023) did not incorporate the 5-year averaged per-glacier mass change observations from Hugonnet et al. (2021) due to the excessive uncertainties associated with these observations. A similar issue may arise with the current dataset.

Regarding practicality and reliability of our dataset, especially considering the uncertainties, we agree that the submitted manuscript and dataset was not clear enough to allow users to address this concern. However, we think that the new changes suggested by reviewer #1, addressed in the previous answers and considered on the revised version of the datasets and manuscript, now provide users with full transparency, allowing to define the practicality and reliability of their individual data usages.

To address the more specific comments:

**Fig 5 in the previously submitted manuscript (now Fig 6):** The aim of this figure is to provide a visualization of the spatio-temporal resolution of the dataset (i.e. available for individual glaciers, gridded tiles, regions). Iceland was selected as an example purely for aesthetic reasons: it's the smallest region and easy to visualize fully in one figure. There is no intention of showing or analyzing the specific results or the uncertainties of this region here. These aspects are clearly shown on Fig. 4, discussed thoroughly in the text and perfectly analyzable from the individual glacier time series and gridded product.

**Regions like the Southern Andes and Antarctic and subantarctic islands:** The issues regarding these regions have been discussed above. They are properly addressed now in the revised manuscript and figures and in the metadata of the dataset so that users are aware of the periods where the time series are less robust.

**Baseline for future modeling:** We agree that the usefulness for modeling was overstated, as a large part of our estimates are extrapolated, rather than interpolated, due to the limited amount of glaciological time series available. We removed these statements from the revised manuscript.

We agree that describing the dataset as a 'new baseline' is beyond our judgment. Data users are in a better position to make such a statement after testing the dataset. We removed these statements everywhere in

the revised manuscript.

**1.7 Data and code documentation and availability**

Firstly, it is great that the code and data are made fully available.

I have a few comments first on the provided data:

- Hosting the extrapolated / modeled per-glacier annual data on the WGMS website could po- tentially lead to misunderstandings. Given that this dataset is not purely observation-based, its direct availability at the WGMS website could result in misleading conclusions. If the decision is made to include the data directly on the WGMS website, it is essential to include a comprehen- sive "meta"- dataset and a flagging system to highlight glaciers/areas where the uncertainties are too large to extract a signal (as discussed in Section 1.5).

As mentioned above in the answer to comment 1.5, we now provide for every region and on a glacier-by-glacier basis, a .csv file with additional information that might be useful for users.

- The type of uncertainty documented in the dataset requires clarification. The term "uncertainty" is used generically, but it is unclear whether this refers to two times the standard error as de- scribed in Line 187, or one or two times the standard deviation (related to Sect. 1.1).

All equations represent uncertainties at $1\sigma$. Reported uncertainties in the text correspond to $2\sigma$ = 95% confidence. Therefore, the term "uncertainty" corresponds to $1\sigma$ when describing equations and $2\sigma$ for reported values.

- Currently, only individual uncertainties are provided, requiring data users to perform their own aggregation. It is strongly recommended to include a dataset with total uncertainties, as this will likely be the most utilized. Additionally, understanding the different sources of uncertainty and their origins took considerable effort. Enhanced documentation explaining these aspects would be beneficial for users.

We added to the dataset a 4$^{th}$ file for total uncertainties for each glacier combining the individual errors from elevation change, annual anomaly and density conversion factor, as in EQ12:

$$\sigma^2{}_{\bar{B}_{cal,g,Y}} = \bar{\sigma}^2{}_{dh,\bar{B}_{cal,g,Y}} + \bar{\sigma}^2{}_{f_\rho,\bar{B}_{cal,g,Y}} + \bar{\sigma}^2{}_{\beta,\bar{B}_{cal,g,Y}} \qquad (12)$$

We revised Table 4, adding the following details to the data filename sections for **Dataset 1: Individual glacier annual mass change time series.**

**Total uncertainty file name:**
   RRR_ gla_mean-cal-mass-change_uncertainty_total.csv

   One file per RGI 1$^{st}$ order region, where RRR corresponds to the RGI-region code

- To enforce people, to look into the uncertainties, consider creating a netcdf file that has the mean time series, the total uncertainties, and a "flagging" system

The gridded netcdf files already contain the mean time series and the total uncertainties per grid point and per year. We could also apply a metadata index as an attribute field to allow users to "flag out" fields depending on index value over specific periods. This would make the dataset more user friendly, allowing users to easily flag out the gridded dataset to consider only the values that support their specific requirements. Still, we think it is more important to have this metadata information for individual glaciers, and then users can decide how to integrate them according to their own requirements. We will generate this additional metadata attribute per grid point if requested by the editor.

- **Issues found in the per-glacier annual time series**
  - no glacier ID for Greenland, everywhere NaN values as IDs. Please update the glacier IDs for Greenland!

True and well spotted. This bug in the code has been corrected now. Thank you.

  - a bit confusing to have sometimes GLIMS_ids and sometimes RGI_ids

The Hugonnet et al. 2021 dataset used the Tielidze and Wheate (2018) inventory available from GLIMS to calculate elevation changes in region RGI-12 Caucasus and Middle East. This decision was made because the glacier outlines from the RGI06 inventory are to a great extent erroneous in this region. The Hugonnet et al. 2021 observations were ingested to the FoG database related to the GLIMS-Id, to make sure that the calculations correspond to the GLIMS glacier extents. For consistency, in order to use the elevation changes from Hugonnet et al. 2021 for the Caucasus glaciers for this assessment's calibration step, we had to consider the Tielidze and Wheate (2018) inventory as well. We think this decision makes sense and is now more clearly explained in the revised manuscript.

Comments on the github/code:

We note to reviewer #1 that the updated code (after the changes done in this first round of review) is available from the new public repository: https://github.com/idussa/global_mb_fusion

- It would be beneficial to include a README document in the GitHub repository that provides a brief overview of the functionality of each script. Such a document would guide interested users on where to find specific processes or analyses within the codebase. While the code does not need to be meticulously documented, a general overview in the README would greatly enhance the accessibility and usability of the repository.

We will add a summary README document in the GitHub repository providing a brief overview of the functionality of each script. This will be done at the end of the review process when the final code has been updated.

**1.8 Terminology**

- The terms "(mean) glacier (annual) anomaly" appear to be unclear and could benefit from clarification. It is recommended to use more specific terminology, such as "(mean) glacier (annual) MB anomaly" or "glaciers with glaciological MB time series". This issue is particularly evident in Figure 1, where the term is not yet explained. The phrase "glacier anomaly" may imply that the glacier itself is unusual or deviates from expected behavior, rather than referring to mass- balance measurements. Including the term "mass-balance" would help clarify the meaning and ensure consistency throughout the manuscript (e.g., line 169 and other mentions).

Agreed we replaced "(mean) glacier (annual) anomaly" with "(mean) glacier annual mass-balance anomaly" everywhere in the revised manuscript text.

- What is the difference between GTN-G regions and RGI6? For instance, in Line 102, GTN-G regions are mentioned, yet later references seem to align more closely with the "usual" RGI6 regions, with the exception of the Southern Andes, which is split differently. It would be beneficial to review the references to GTN-G and RGI6 throughout the manuscript to ensure consistency. If possible, it is recommended to use only one of these terms to avoid confusion.

Agreed we now refer only to RGI regions in the revised manuscript text.

**Specific comments**

To maintain relative conciseness in the review, specific comments have been provided without consistently using phrases such as "please reconsider" or "please change." However, many of these specific comments are intended as suggestions to guide improvements and offer constructive feedback, rather than as strict directives. Some comments made in the general comments section may be repeated in the specific comments section, potentially with additional elaboration or different descriptions. In the response to this review, you may disregard specific comments that you have already addressed in the general comments.

**2.1 Title, Abstract & Introduction**

- **L1** The title overstates the precision of the data and does not acknowledge the uncertainties sufficiently. Maybe change to something like "Uncertainties in extrapolating annual glacier mass changes from 1976 to 2023: Estimates for every glacier world-wide based on in-situ and geodetic data". At least the methods of your approach should somehow be incorporated in the title. For example, the phrase "from in-situ extrapolation" could be added to the title.

**Regarding the Tittle:**
We propose two new titles as a better fit regarding referee #1 (one more technical and the other probably for non glaciologist). The take of the editor and reviewers on these title is welcome.

1. Annual mass change of the world's glaciers from 1976 to 2023 by temporal downscaling of geodetic estimates with glaciological observations

   Annual mass change for the world's glaciers from 1976 to 2023 by temporal downscaling of satellite data with in-situ observations

**Regarding the Abstract and Introduction:**
The Abstract and Introduction have been fully rewritten accounting for reviewer #1 and #2 comments. Some comments remain unanswered because they are no longer relevant for the revised manuscript (e.g. typos of non-existent text, comments on inexistent figures, etc). We state in every case if this is no longer relevant.

- **L19-20** In my opinion, you can not yet conclude that from the current leave-one-out cross validation (see Sect. 1.3). Also change: "in the conservative side" to "on the conservative side

No longer relevant in the revised manuscript

- **L32-34** From these 500 glaciers, much less are actually usable time series. I think it would be valuable to rather mention how many you use (the glaciers with "glacier anomalies"). You also say, that nearly all glacier regions are represented. However, from these 500 glaciers, a lot are in Central Europe... related to Sect. 1 and idea of giving a clear overview of amount of used glacier MB anomalies and covered years-

We state in the introduction the number of glaciers with glaciological observations available in the WGMS-FoG database, which is about 500 glaciers. Their number, coverage, and hypsometry are well represented later in Fig.1. As for the number of annual mass-balance anomalies available in every region, the information is available in Fig. 1. Information on the annual mass-balance anomalies used by every glacier to calculate the mean annual mass-balance anomaly is available in the metadata.

- **L54** define FoG here (it is defined only later)
- **L54-58** long sentence, I don't understand the meaning of the sentence, specifically of "and evident..."
- **L67** "global glacier ice changes": mass or volume?

- **L68-71** Why are the geodetic observations of Hugonnet et al. 2021 giving you information on annual mass changes. For example from where do you get the density conversion information here? Rephrase the paragraph to clarify that you use the glaciological mass-balance observations.
- **L74** define "Fluctuations of Glaciers" at the first usage (L54)
- **L77-87** I am not sure if this paragraph is really necessary. You write here what you did, but not really the results. E.g. L78-80: strange to mention that here, as it does not tell the reader in which regions it works and in which it does not work so well...
  L85-87: the two sentences seem to be very similar, maybe combine in one sentence
  The last 6 points are no longer relevant in the revised manuscript

**2.2 Data and Methods**

- **L93** do you apply any correction as the RGI dates are often different to year 2000?

Yes, when transforming from m w.e. to Gt (individual glaciers, regional or global estimates), we use glacier areas that have been corrected to the year 2000 using area change rates from literature and updated from Zemp et al. (2019). This is stated in Method section L297 and further discussed in section 5.5.5

- **L109** to which year do these outlines correspond?

They are also corrected to 2000

- **L104** this should be in the caption, but actually the grey bars are almost not visible
  We state this in the caption "Glacier hypsometry from RGI 6.0 (grey) is overlaid (and almost hidden) with glacier hypsometry of the geodetic elevation changes (Geo, blue)"

- **L108** FoG already defined in database

No longer relevant in the revised manuscript

- **L114-L115** "throughout their full hypsometry" : what is meant by that?

That means, throughout their entire elevation bands

- **L136** Are these short-term geodetic records also excluded in the statistics of Fig. 1. Please clarify! Similarly, in Fig. 2, you show the short-term geodetic records, I would recommend removing or labelling them as you don't use them for the calibration.

The reviewer must be confused here. Figure 1 is showing the number of glaciers with geodetic observations, and the surface covered by the glaciers with geodetic observations. It is not showing the number or period of the total geodetic observations. We removed the geodetic observations shorter than 5 years from Fig. 2.

- **L142-148** It is difficult to understand these steps if you haven't read the individual subsections. I would suggest to either put that at the end of Sect. 2.2.3, or instead move the important stuff into the captions, or somehow clarify that the individual steps will be explained later ...

The methods text refers to the figure at every step. The individual steps are explained briefly at L150-L155, and then later in more detail.

- **L150** Fig. 2 caption: (see Fig. 2 comments below)

The figure has been remade completely

- **L160** What is $\sigma B_{glac}$ ? I assume it corresponds to one standard deviation, but I think it is important to mention that. You often have the σ term in your equations, maybe clarify here directly if those represent always one standard deviation.

Everytime there is σ it means one standard deviation, otherwise it will say 2σ

- **L169** "glacier annual anomaly" → see Sect. 1.8

Changed

- **L168-170** You choose a threshold for the amount of years within 2011-2020 for a glacier to be chosen to be used for your calibration approach. However, how many years outside of the 2011-2020 years are necessary that a glacier is chosen to be used? I think this is important to mention / analyse to understand the "number of glacier anomalies" of Fig. 1

If a glacier has glaciological measurements during at least 8 years and during the reference period, the anomaly is calculated and used.

- **L174** "more than two...": replace with "at least three glaciers with mass-balance anomalies". This threshold is only valid for the search radius, as there are many periods and regions where the anomalies come from less than three glaciers? (related to Sect. 1.4)

This comment is no longer relevant with the change to the kriging method

- **L180** the uncertainties of the anomaly are computed without any inverse weighting correctly? Maybe clarify that!

This comment is no longer relevant with the change to the kriging method

- **L182** Do you mean here $\sigma B_{glac}$ ? If yes, please clarify and maybe refer to eq. 1?

This comment is no longer relevant with the change to the kriging method

- **L183-186** These are almost the same sentences as in line 162-164! Remove one of them. Also typo: replace "ire" with "are"

This comment is no longer relevant with the change to the kriging method

- **L186-189** Why do you use two times the standard error here, why not the standard deviation. More about that in Sect. 1.1

This comment is no longer relevant with the change to the kriging method

- **L191** Why can you assume here random error propagation? Couldn't it be that on a specific year, glacier mass balance is under/over estimated for several glaciers because of a specific "climatic" phenomenon? (see also Sect. 1.2)

This comment is no longer relevant with the change to the kriging method

- **L194** "low confidence glaciological series" → how do you define "low confidence"? Are only high confidence glaciological series included in the statistics of "glacier anomalies" in Figure 1?

L173-174 comment changed to this line and is explained better. For example: Urumqi East and west branches (ASC) correspond to the separation of the Urumqi glacier into two different branches, we use only the full glacier time series because it is longer, and because it is likely more representative of the climatology of the region, since the branches are very small patches of ice.

**L196-197** What do you mean by that? I guess you mean by that that you want them to have anomalies until then? This sentence does not say something about how much glaciers with
"glacier mass-balance anomalies" are used, but when I first read over the sentence I thought that you want to say that you somehow only select glacier mass-balance anomalies that cover the entire period 1976 to 2023. Please rephrase.

This has been rephrased

- **L197-200** How do you define "climatically similar" (see comment to Table 2)? Here you describe gaps in mean calibrated glacier MB annual-anomalies that result from the aspect that no glaciers with observations for these gap years were found within the search radius? But do you also explain how the uncertainties are estimated for these gaps?

"climatically similar" = L198 the best correlated glaciological series from neighboring regions

- **L212** add the unit of the density conversion factor (maybe best to make it in kg m-3 and then divide by 1000 in eq.5) it should also be 0.85+/-0.06 (not 0.60)

Done

- **L215** here the σρ is in units of kg m-3, please be consistent. The acronym is also different to line 212.

Corrected

- **L213** eq.6 is a bit confusing, as it is unclear for what this is done. Maybe add that this uncertainty is later used for the weighting algorithm of the geodetic samples in eq. 8 (if I got it correctly).

Clarified in L236: This corresponds to the uncertainty on the geodetic mass balance rate in m w.e.

- **L218-222** I had to read this several times to hopefully understand it. You create for every geodetic sample of >= 5 years, an individual time series which goes over the entire considered period (not only the period of record of the geodetic sample), correct?

Correct

- **L222** here you write "only geodetic observations larger than 5 years" are used (also in L230), but in L155, you wrote longer or equal to 5 years. Which option did you choose?

Corrected in the text, only larger than 5 years are used.

- **L224-226** You explain the uncertainty estimate, but there is no equation to it. I have the feeling that exactly this is already done in eq. 9-12, maybe? If yes, please merge this sentence with the explanations there? Or is this where you basically convert $\sigma b\ e\bar{t}ag,y$ to $\sigma b\ e\bar{t}ak,y$ ?

This has been rephrased

- L231-234 Maybe add the respective acronyms (such as $W_t$ for the second uncertainty part) at the end of the descriptive sentences. Like that it is easier to understand eq. 8, and otherwise the acronyms of eq. 8 are not explained.

Done

- **L239** replace considering with "assuming", because you do not show that,or?

This has been rephrased

- **L241** I don't really understand why you call it "error" separation. You aggregated the errors be-forehand for the weighting, but you have the errors already indivudally, so you don't need to separate them again, or? What eq. 9-11 do is basically averaging the uncertainties from the in- dividual k mass change time series of the individual geodetic samples. If yes, maybe instead add something like that instead of saying error separation. You could also maybe explain everything from line from 239 to 244 before explaining the weighting, and then at the end explain how you get to the total uncertainties (L245). Then you don't need to say you look again at the errors separately...

This section has been rephrased

- **L245** Why can you assume independence and add up the square roots of the different squared standard deviation uncertainty sources to get to the total uncertainty? If I get it right these three uncertainties are the ones available in the per-glacier files. In the data files the total uncertainty is not available, maybe consider adding the total uncertainty as dataset, or somehow clarify that in the manuscript

Total uncertainties have been added

- **L240-246** What kind of errors do these uncertainties represent? Standard errors or standard deviation? I am just wondering because at least one of them apparently represents the standard error (i.e. the one calculated via line 192?)

This comment is no longer relevant due to the change to the kriging method

- **L248-249** I find this sentence a bit confusing. Maybe clarify that this is what you did, i.e. rephrase by saying.... We calibrated a mean annual mass change for …

- **L249** Explain what you mean with unobserved. If I understood correctly, it is a glacier that is not

available in Hugonnet et al. 2021 (and also does not have any other geodetic observations or in-situ observations). Correctly

This has been clarified in the revised manuscript. Hugonnet did not have full coverage, only 97.4% of the global glacier surface. In the Fog, only estimates for 205.120 glaciers have Hugonnet observations, which is 95% of the global glacier by number. The full coverage of the FoG is 206554 glaciers with Geodetic observations, which is 96% of the global glaciers by number.

- **L250** replace "Individual" with "individual"

Done

- **L254** why is it only of the observed glaciers? Don't you use all glaciers where you created calibrated data and then in addition the "unobserved" glaciers?

It refers to all glaciers, unobserved glaciers are considered to behave as the regional mean, so then this regional mean is multiplied by the unobserved area.

- **L255-257** It is great that you account here for spatial correlations, and have identified that some error sources are significantly correlated spatially, such as elevation change, density conversion and annual anomaly prediction. Are there any figures/analysis for that? Iam not an expert here, but I am wondering if you need to account for a similar spatial correlation when estimating the total uncertainty of an individual glacier mass balance time series? The reason is that the glacier anomalies are estimated from time series that are coming from different glaciers.

Spatial correlations are considered wherever it was possible. This is now much more clearly explained in the revised manuscript.

- **L263** Huss et al., in preparation: It would be good to add some details here. When looking into the density uncertainties of individual glaciers, they seem to be quite small.
This has been rephrased

- **L268** "errors to the real values": what do you mean by that
This has been rephrased

- **L269, eq.16** see Sect. 1.2

Answered in 1.2

- **L270** "as independent". Why can you assume that?

Se response to comment 1.2: The reason behind this difference in focus is that the assumption of correlation between sources has negligible impact compared to the spatial correlation of errors within the same source. This is because we only have 3 main sources of uncertainties, while we have 200,000 glaciers distributed spatially.

- **L273** Fig. 2f does not exist

True, deleted

- **L276** Same as L268. Why can you assume that? you use different expression for the same aspect, i.e. assuming that the errors are independent, then write law of random error propagation... maybe stick with one thing

This has been rephrased
- **L279** Zemp et al. 2019: "et" is written in different text style

Checked

- **L281** regional mass loss uncertainties independent and uncorrelated : from where do you know that you can assume that...

Answered in 1.2

- **L285** do you assume that all mass loss is above sea level? eq. 21-23 are not explained

Yes we assume all loss above sea level. They are explained in L280.

- **L315** starts at the first year of mass change records, is a single mass change record sufficient to start from that year, or is is three glacier MB anomalies?

We do not fully understand what the reviewer means here

**Table 1**

- **L5** maybe add another row with the differences in the Uncertainty?
Done, as in the GCOS 2020 requirements

**Fig 1**

- It it strange that you show the location of all glaciological samples, although in this study you only use a fraction of these (i.e., just those with glacier anomalies). I think it makes much more sense to visualise those glaciers with a MB time series. I would prefer to see the hypsometry of those glaciers with "glac anomalies" instead of those from all glaciers with measurements.
- For me, the focus of this figure at the moment is to show the hypsometry of the glaciers of that region. For me it would be more interesting to see the statistics of the glaciers that are used for the "glacier mass-balance anomalies" visually. For example, how many glacier MB anomalies are actually used for the individual years of the time series

Answering the two previous points, Figure 1 is meant to show the number, spatial coverage and hypsometric coverage of the input geodetic and glaciological observations existing in the FoG, our input dataset. The information on the number of glacier annual mass-balance anomalies calculated for every region considering our reference period (2011-2020) is stated in the figure. Further information on statistics requested by the reviewer is more relevant at glacier level and is available in the metadata. It is not possible to add all this information for each glacier in Fig. 1.

- maybe remove duplicate labels to make the figure less busy
We disagree, having the labels for every region is reader friendly.

- it is very difficult to see the hypsometry of the "glaciological" sample as the red is difficult to see
This is exactly the point, to show that the coverage of glaciological observations is very little both in number, coverage and hypsometry. Whereas the coverage of geodetic observations is nearly complete, as it hides the grey (full) hypsometry of the RGI glaciers.

- you give the % of observed glaciers in terms of glaciological or geodetic observations. Is a single observation in one year that the glacier is here "observed"?
The figure is not showing the number of geodetic observations, but the number of glaciers having geodetic observations, and the area and hypsometry they cover. As reminder, 205.120 glaciers (95% of the world's glaciers) are covered by the 20-year estimates from Hugonnet et al. (2021). The geodetic observations coming from other studies cover 14% of the worlds glaciers. Considering all this observations together, there exist geodetic measurements longer than 5 years for the number of glaciers shown in Figure 1, the area and hypsometry they represent. Geodetic estimates shorter than 5 years are very rare, and usually exist only over well sampled glaciers that are as well covered by longer periods. For glaciological, we consider glacier-wide time series.

The red shows the hypsometry of glaciers with any kind of glaciological observations (I count 468 glaciers there)? If I understand correctly the amount of glaciers with observed mass-balance time series are described by the "glac anomalies"? How many observations are necessary to be such a glacier? Maybe add that to the caption (it is later described in the text, but maybe good to explain it also here). Also related to that: change the wording of "glac anomalies" and describe that in the Fig. 1 caption, see Sect. 1.8).

- the grey "RGI6" glacier hypsometry is almost not visible (specifically if you print it)

The coverage of geodetic observations (blue) is nearly complete; therefore it hides the grey (full)

hypsometry of the RGI glaciers. This has been clarified adding a statement in the figure's caption.

- explain in caption the meaning of the circles (glacier region area). Location of the circles is sometimes far away from the region's glaciers. For example in CEU, SCA, NZL. In 17-SAN, there are two circles, probably from the two subregions, this needs to be documented further and if the two circles are kept the region should also be split up via the "black" lines.

The meaning of the circles has been added to the caption

**Table 2**

- Why did you exclude these specific glaciers?

Explained previously in this document. For example: Urumqi East and west branches (ASC) correspond to the separation of the Urumqi glacier into two different branches, we use only the full glacier time series because it is longer, and because it is likely more representative of the climatology of the region, since the branches are very small patches of ice.

- How did you choose the complementary glacier mass-balance anomalies?

In non-observed regions like ACS and RUA, we selected the closest and more similar regions time series. For ASW and ASE because they are the longest available time series for HMA, as done in Zemp et al., 2019. For NZL, regional series only start by 2005, we add Martial Este because is well correlated and had similar amplitude, although because of the distance (>7000 km) it won't have much weight by kriging.

- How did you choose the complementary normalized glacier mass-balance anomalies?

By choosing the best correlated glaciological series from neighbouring regions over the common period of observations, i.e. the ones that better represented the climatology of the regional time series during the common period.

- typo: Hinteeisferner → Hintereisferner

Corrected

**Fig. 2**

- It is a bit strange that you show the method for one of the best measured glaciers. Hintereisferner has an annual time series for over 60 years. I think it is very necessary to show at the same time the method for a less well sampled glacier, i.e. a glacier with no in-situ observations, with less and shorter available annual MB anomalies, and with only the Hugonnet et al. 2021 geodetic sample data. This second glacier corresponds better to most glaciers world-wide, I guess? You could add the other glacier in the same figure to have the comparison. Or move the Hintereisferner example in the supplements and add here another glacier.

Figure 2 has been completely remade accounting for the changes in the methodology to kriging. We also now added two more glaciers other than Hinteresiferner, Gulkana in Alaska, a middle sampled region, and Mittivakat in Greenland Periphery, and undersampled region.

- You show here the data of Hintereisferner only from 1952 onwards (i.e., the start of the Hintereisferner observational period). In the dataset that you want to publish, however, the new calibrated time series begins already in 1915. I think you should either mention this in the cap- tion or show it in the figures. From the perspective of Hintereisferner, the period from 1915 to 1952 is the most interesting, as this is the period where your method actually creates new data.

The figure is meant to show methodological steps, not the Hintereisferner time series. We clarify in the legend that the full times series starts in 1915 but for visualization purpose you decided to start in 1950.

a: There is a big red cross in North Africa with the text B*glac* search. I find that rather confusing.

Panel a has been removed

- b: you mention 10 glaciers with anomalies, I guess one of the 10 is HEF itself, correctly? So, what we see are 10 thin lines together with the inverse-distance weighted average and the uncertainties around it? And if I understand correctly, some anomalies are over the entire time period, and others are just over a period of time. It would be interesting to somehow visualise that. Probably this gets easier with a glacier with less glacier anomalies around. I think it would be good to color the line showing the in-situ observed glacier mass-balance anomaly of Hintereisferner, I guess it will be near to the mean annual anomaly? At least it should be clarified in the caption or subplot that one of the lines represents the anomaly of Hintereis- ferner. You do not explain what the grey shading is. You added the equation and from that I assume that the grey shading are the uncertainties from eq. 4, but I believe, both the grey shading and the equation need to be explained (e.g. by refering to eq. 4, and saying in caption that the shaded area corresponds to "two standard errors from the glacier MB anomalies and the glaciological sample uncertainties"?

This comment is no longer relevant due to the change to the kriging method

- c: you write that you only use geodetic data with at least five years. However, in the plot, it seems like you also plotted the geodetic sample data for smaller periods? (even for single years, e.g. 2003). As you do not use them in the calibration, I would not include them in the plot, unless you somehow mark them in another color/style to clarify that these are just used for validation? Please also add how many grey lines there are, i.e. how many geodetic mass change observations were usedFrom Sect. 2.2.3, I understood that every of these "k" lines have their own uncertainty estimated from (b). Although too complex to visualise, you might mention that in the subplot or caption. I would also prefer to see the uncertainties of the geodetic estimates instead of having red/blue filled ares to the zero line.

We removed the geodetic observations shorter or equal to 5 years from the plots. Adding the uncertainties would make the plots too busy, for clarity of visualizations we prefer to keep the figure as simple as it can be.

- d: What are the red and blue lines? I guess this is the same ones as in c? Not sure if it is necessary

  to keep them, but in any case, you need to describe them in the caption or in a legend.
  You just write, that the grey is "uncertainty". Please clarify what kind of uncertainty it is (see comment in Sect. 1.1).I would like to see here how the mean calibration time series changes to the actual in-situ HEF observations with that approach. I guess it is quite near as it is included. I think you should add a colored line with the actual Hintereisferner observations (similar as suggested for subplot b).

Red and blue lines are described in the caption of the figure: "(b) Calibration of the mean annual mb-anomaly over geodetic mass balance observations available for each glacier (Red and blue lines). Grey lines correspond to the individual calibrated time series for each geodetic mass change observation."

- add the corresponding equation numbers to b, c, d; consider adding legends into the subplots to clarify better what the lines mean

The figure is already quite complicated and full. We prefer to keep the figure simple.

- Fig. 2b, d: "mean calibrated time series" and "mean annual anomaly" isn't it mean and "some kind of uncertainty" that you show?

In both cases we show both the means and their calculated uncertainties.

table 3:

- "low confidence" glaciological/geodetic estimates : how is this defined?

Answered and clarified in a previous comment

- empirical function "of"...Hugonnet et al. 2022; Huss and Hugonnet (in prep) → called differently somewhere else, be consistent!

We are now consistent in citing as Huss et al. (in prep)

table 4:

- "uncertainty" : what uncertainty does that describe? Standard error/standard deviation/two times standard deviation?

This issue has been answered elsewhere in this document

- Dataset 1: there is no dataset with the total error. Interested people need to aggregate the uncertainties themselves, which is error-prone.

Added

- Dataset 2: is the dataset really that large if you add all years together, it is easier to download just one file instead of many...

Both formats are possible.

- Dataset 2: here you have mean time series and total error : again, what does total error represent?

Total errors for an individual glacier: EQ 12

- see table XX: -> ref. is missing

corrected

- time series start "of" hydrological year (of was missing)

We mean the starting hydrological year

**2.3 Results**

- **L326** What do the numbers represent? One standard dev. / std. error?

This issue has been clarified elsewhere in the document

- **L333** Maybe refer here to Fig. 3; is it "m" or "m w.e", in Fig. 3 it is in m w.e.

We say it is thickness change not m w.e.

- **L356** attention: your estimates are not "observations" anymore, as you apply basically a model to get the individual glacier MB time series. Consider rephrasing the word "observed".

We disagree here, these are the statistically modeled observed values, we think the observational vs modelled issue has been sufficiently clarified throughout the revised manuscript, and we can use the word observed to refer to our results.

- **L357-366** It is unclear to which figures or tables you refer to in this paragraphs. You write volumes, but write "GT". I would suggest to replace volume everywhere by mass to coincide with the GT unit. Can you really say with confidence that these mass changes occurred on single years? More in Sect. 1.6

This has been rephrased

- **L359-362** I didn't understand the last part of the sentence, consider rephrasing the sentence.

This has been rephrased

Fig. 3:

- Global subplot: move that subplot up as the mass change time series axis is very near to the the Russian Arctic subplot. To make some space, move maybe the global pie to the center left part of the plot

Done

- It is a bit confusing that the global plot is in a different style than the regional subplots

Corrected

- Some of the timeseries are dificult to see due to the bright colors of that region. Consider using black insted and only colo

We prefer color coding to separate regions
- Caption: the area of the pie charts: maybe clarify that you mean the size of the circle
Corrected

Fig. 4:

- Why does RGI19 and RGI05 regional glacier mass increases from 1979 to 2000? Are there any physical explanations for that or other studies showing the same? It seems like Zemp et al., 2019 had less positive MB on these two regions.
This is no longer an issue for RGI05 (corrected thanks to kriging), for RGI19 this is discussed in the text (L580-585)

- You describe here the meaning of m w.e. but this concept is used earlier. Maybe rather describe it somewhere in the methods?
It is described before in Table 1

**2.4 Discussion**

Fig. 5

- More in Sect. 1.6

Most of the issues raised here have been answered elsewhere in this document, mostly section 1.6.

- year 1976: (i) and (ii) look very different. On the individual time series, it looks like all glaciers lost mass, while on the gridded dataset it seems rather that they gained mass. Why?

This was an error in the figure and has been corrected, thanks for spotting this.
- **L398-400** "...selected considering..": but that means you select glaciers where you know it works. Isn't that kind of a bias towards specific glaciers?

- **L403** How many of the 32 glaciers are in CEU? Does a typical "reference" glacier not have much more "glacier anomalies" in their search radius than a "normal glacier"? see Sect. 1.3

This issue is discussed in the text of the leave-block-out cross validation section 5.3 of the revised manuscript and in answers 1.6 to this document.

- **L431** "seven glaciers" : In Fig. 2b, you wrote that there are 10 glaciers used to estimate the "Mean annual anomaly" of HEF. If you remove the anomaly of HEF for the cross-validation, there should be still 9 available glaciers, why is it now seven? In Fig. 8a, there are also 9 glaciers listed.

This comment is no longer relevant due to the change to the kriging method

- **L435-436** To my knowledge, the RMSE (combines variance and systematic errors) and STD-diff

(variability in the errors) do not directly verify whether there is no systematic error for Ba. With-out checking the bias, you cannot confidently rule out systematic errors. I think it would be important to include the bias in Fig. 6,7.

This is addressed in section 1.3 of this document. We now use only the mean of residuals and the standard deviation of residuals as metrics to quantify potential systematic errors and random errors within the cross-validation results, respectively. We do not use the mean absolute error or the RMSE anymore, since they don't provide any additional information.

- **L441-444** Maybe clarify by writing sth. like ... For XX out of XX glaciers, the actual standard deviation is >XX larger than the standard deviation estimated by the cross-validation. Do you believe that in general the interannual variability is underestimated by your approach? This is very important to clarify, as e.g. glacier models interannual variability largely depends on the precipitation factor.
This has been rephrased in the revised manuscript

- **L448-454** I don't fully agree that you can conclude all of that. I am specifically confused about the comparison of two times the standard error vs MAE. (discussed more in detail in comments about Fig. 6d in Sect. 1.3).
The MAE is no longer used as a metric, this has been rephrased in the revised manuscript

- **L454-456** This is a major problem, and should also be accounted for in your cross-validation and uncertainty analysis. This is one of the main reason why I can not accept the conclusions from this paragraph. See Sect. 1.3 for an idea of a data-denial analysis.
This issue has been addressed in 1.3 and the data denial analysis and the statement rephrased in the revised manuscript

- **L476** are these cumulative mean mass losses within the uncertainties of Zemp et al. 2019? Please add uncertainties to these numbers!
Yes, they are, uncertainties added.

- **L486** Deviations of more than XX : please clarify that you compare here to Hugonnet et al. 2021 2000-2019 period?
Why do you have these differences in the regional trends w. Hugonnet et al. 2021 in the period 2000-2019? Do they only come from additional geodetic data used over that period and in these regions? Or do the glacier anomalies (in-situ observations) also influence the regional trend over the period 2000-2019?
In the entire paragraph, the mentioned differences are within the large uncertainty ranges of the regions. If you mention the differences, I believe you should also mention that uncertainties are larger than the differences.
The discussion here has been rephrased. We discuss in this section where the differences between Zemp, Hugonnet and our study might be coming from.

- **L535-537** You apply a "model" by extrapolating the glacier anomalies from reference glaciers to another glacier. For example, you assume that the anomalies are similar for nearby glaciers, you even select a glacier with the most similar climate for glaciers in regions without glacier anomalies. All these choices are somehow like a model. Therefore, you should not call this product a purely observation-based product.
We agree. These statements have been rephrased throughout the revised manuscript.

- **L549** From where do you know if the estimated uncertainties are sufficient. The cross-validation that you applied can not tell you that as none of the glaciers of the cross-validation are e.g. in Southern Andes or Subantarctic and Antarctic Islands.
We agree. The discussion has been enriched with the leave -block-out cross validation and statements adjusted throughout the revised manuscript.

- **L550 onwards** You clearly state the problem, but I am wondering if it is then really valid to still give an annual time series for these regions and periods of extremely high uncertainties

We think is valid, as is the best we can do with the observations we have, we are careful explaining the limitations, and this stands as a clear statement showing where the efforts need to be put on in the future of the observational front.

- **Sect. 5.5.1** I think adding an overview of which geodetic data sources were included before year 2000, and which after year 2000 would help. Which kind of additional geodetic observations were used to compare to Hugonnet et al. (2021) in the period 2000-2019? Did you use any photogrammetric data for geodetic observations in the past? L560: I assume, this is future work, so maybe clarify that.

This has been included in the metadata file

- **Sect. 5.3.2** maybe also discuss potential usage of terminus location (e.g. more used in glacier runoff studies)

We removed all statements of potential uses of the dataset on the revised manuscript we leave this to the user's criteria, they have now all the information to know if the dataset is suitable or not for their purposes.

- **L592** please remove, your dataset is not purely observational

Removed

Fig. 6 (see also Sect. 1.3)

Figure 6 (new figure 7) has been completely remade accounting for the changes in the methodology due to kriging, it now considers a much larger sample of 74 reference and benchmark glaciers and now uses as statistical metrics the mean and the standard deviation of residuals. Many of the following reviewers comments are therefore not relevant anymore.

- maybe add in the legend the amount of considered glacier anomalies for each glacier and the region where they are located in!

We do not see a simple and visually neat way to add that information in the figure for 74 glaciers. This information is available in the metadata file.

- FIg. 6a-d: Please adapt all subplots to have the same scale on the x-axis and y-axis, with equal tick labels and lengths. E.g. in python, you can set: $ax.setaspect('equal')$. I believe this would help a lot to correctly interpret the plots (and would help to understand that all grey dashed lines are 1:1 lines).

We disagree to use same y-axis. The mean error can be positive or negative and the standard deviation is positive, using same y-axis would make us loose too much space and is not a critical aspect in this figure.

- does it work better for those glaciers that have a lot of glacier anomalies nearby?

Results of leave-block-out cross validation show that random errors increase with distant samples. See section 5.3 in the revised manuscript.

- Fig. 6d: Can you please clarify clearly the meaning of the y-axis (more details on that in Sect. 1.3).

Done

- caption L419: I find it hard to interpret "std-diff"? Does every glacier count the same? Does it basically describe whether the differences are similarly large for the different glaciers?

Not relevant for new figure
- caption L422: mass change "trend" (not std. dev)

Changed

- caption L423: mass change "standard deviation" (not trend)

Changed

- caption L424: x-y descriptions does not display the actual Fig. 6d. As all other figures represent on the y-axis the leave-one-out cross-validation results, it would be best to exchange x and y-axes in Fig. 6d.

Not relevant for new figure

Fig. 7

- maybe add the uncertainties of the "leave-one-out" time series

Done

- caption L459: maybe add the word "observed" here. If I understand it correctly the time series that is actually used in the dataset (i.e., e.g. for HEF Fig. 2d) is not shown here, or? Or is it in case of these reference glaciers the same?

It is not clear what the reviewer means here

- caption L459-460: caption description of right and left is reversed! Eventually show the other non-selected reference glaciers in the supplements or appendix

Well spotted, Thanks

**2.5 Data and Code availability**

see Sect. 1.7

**1.6 Conclusions**
**Regarding the Conclusions:**
The conclusion has been fully rewritten accounting for reviewer #1 comments stated here below and reviewer #2 comments.

- **L628** Here again, I would prefer to remove that statement of "independence" and of "purely observational nature".
- **L629-630** With the current cross-validation analysis and figures that I have, I can not yet conclude that. Please recheck, once you refined the cross-validation (see Sect. 1.3).
- **L645-end** These are very broad use cases, maybe a bit more concrete and nuanced use cases would help to clarify what this new dataset can do (i.e., where it adds value) compared to other existing datasets.
- **L657** replace "The" with "the

The last 4 points are not relevant in the revised manuscript. We removed statements on uncertainties and on use cases.

---

## Author Comment (AC4)

**Reply to reviewer #2**

We are very thankful for the anonymous reviewer's constructive review of our manuscript. Please find here our answers to their main and specific comments.

Following reviewer #1 comments (section 1.1), we performed major changes to our algorithm spatial interpolation method. When calculating the mean annual mass-balance anomalies, our previous inverse-distance weighting (IDW) methodology lacked a way to capture the varying errors with distance to the measurements used for interpolation. We decided to replace the IDW spatial interpolation (for which there is no integrated error propagation) by a kriging spatial interpolation (that includes error propagation natively). We invite reviewer #2 to read our answers to reviewer #1 for more details on this. Some of the comments in this review are not relevant anymore due to these changes. We state them accordingly.

The document is color coded as follows:
**Black: reviewer general comment**
**Green: answers to reviewer**
**Blue: extracts of the revised manuscript**

In this study the authors provide mass changes for a large number of world glaciers from 1976 to 2023 by combining data sets based on in situ and space-based measurements. This new product, to be distributed by Copernicus services, will be of invaluable value for climate applications, including sea level rise and land hydrology. It will be definitely of high interest for the scientific community.

While I greatly appreciate the efforts made by the authors in combining different datasets, in performing appropriate calibration and in providing product uncertainties, I find that the manuscript requires major improvements in terms of presentation. The paper is very difficult to read. First of all, it lacks a number of general information about the data used to be understandable to non-experts. Some sections are quite technical and poorly explained. A large number of variables are not defined, and some important information is just provided in tables without explanation in the main text.

I recommend to the authors to consider my comments below and provide in a revised version a text clear enough to be appreciated by both experts and non-experts. As it stands, it is not the case.

We revised the manuscript to make it more accessible to non-experts. Important variables are now clearly defined and explained in the main text (already in the abstract and introduction), technical sections are better explained and, hopefully, will be appreciated by both experts and non-experts.

**General comments:**
- The abstract and introduction are too vague and lack useful information. It is unclear in both the abstract and the introduction which data sets are considered and combined. How have they been obtained? The data section refers to the data sources given by their acronyms, but no information is provided on the methods to obtain the data. What is the proportion of in situ data and remote sensing data? Are the latter only based on ASTER DEMs as described in Hugonnet et al. (2021)? What is the spatio-temporal coverage of each data set? It is insufficient to say (as written in the abstract and introduction) that geodetic and glaciological data are used. It is unclear whether in addition to the Hugonnet et al's data, other remote sensing data are considered.

We agree the abstract and the introduction were not clear enough in showing this specific information. Most of this is communicated later in the input datasets section and Fig. 1 in the data and methods section. More details are now provided in the abstract and introduction, more specifically:

1. The input data sets used with their proper references, where users can find all the information on the methods to obtain the data.
2. The proportion of in situ and remote sensing data used, their spatio-temporal coverage and a summary of their technical details (sensor sources) and references.

We think that the level of detail provided in the revised abstract and introduction, and in the data section is well adapted to the scope of our work. We provide all the necessary references for users to get more information on input datasets, methods, etc. if needed.

- While the paper mentions the percentage of glaciers considered by Hugonnet et al (96%) and the total number of glaciers is never mentioned, the percentage of glaciers considered from in situ measurements.

This information was available from Fig. 1 but as the reviewer correctly states, not in the abstract, introduction or in the manuscript text. This is now corrected.

I suggest to rewrite the abstract and introduction to clarify these issues and provide the reader the missing information. The figures are in general too busy. The figure captions need to be extended and provide the definition of the parameters appearing in the figure.

The abstract and introduction have been re-written accordingly. We edited the figures so that they provide useful and intuitive information properly defined in the captions.

**Line-by-line comments**

- Abstract, line 14: the term 'remote sensing' is too generic. Mention how elevation measurements are obtained (in addition to streo images), laser altimetry and space gravimetry can be also used (the latter given rirectly mass changes).

The abstract has been fully re-written. We mention in the revised introduction how glacier elevation change measurements from DEM differencing are calculated with multiple citations that explain in detail the different sensors from which they can be obtained and how they are calculated. It is out of the scope of this paper to describe fully all the existing methods to measure glacier changes (and even more out of scope to describe them in the abstract). We refer the readers (and reviewer) to a recent review in Reports on Progress in Physics (Berthier et al., 2023).

L55-59: DEM differencing was initially applied to individual glaciers with DEMs derived from maps (Joerg and Zemp, 2014) and aerial photographs (Finsterwalder, 1954; Thibert et al., 2008; Papasodoro et al., 2015; Belart et al., 2019), but has now evolved to include data from airborne Lidar (Echelmeyer et al., 1996; Abermann et al., 2010) spaceborne altimetry (Jakob and Gourmelen, 2023; Menounos et al., 2024) and satellite derived DEMs from multiple sensors (Toutin, 2001; Berthier et al., 2023).

We use glacier-wide (therefore gravimetry estimates are excluded) elevation changes available in the Fluctuation of Glaciers database. All the details of the observations used are available from the citations provided in **section 2.1.2 Glacier elevation and mass change observations:**

L123-127: For more details on the specific input data, auxiliary data, retrieval algorithms and uncertainty estimation of the independent FoG glacier elevation and mass change observations please refer to WGMS (2024). More details on the glaciological method can be found in Østrem and Brugman (1991), Kaser et al. (2003) and Zemp et al. (2013, 2015). For the geodetic method and its error sources see WMO (2023) and about measuring glacier mass changes from space, see Berthier et al. (2023).

- Abstract, line 15: '… resolved information'. What is this information?

This is not relevant for the revised manuscript. The abstract has been fully re-written

- Abstract, line 16: how many world's glaciers?

This number is given in the introduction, we do not think it is needed in the abstract.

Abstract, lines 19-20: sentence unclear. What do you mean by 'leave-one-out cross validation'. This is a technical term. Should it appear in the abstract?

The abstract has been fully re-written. Cross validation is a new and very important aspect of this work. Due to the lack of independent measurements available to compare and validate our glacier-wide mass change assessment (because all the available measurements are used), performing a cross validation is the only way to validate our results and their uncertainties. We believe it is important to note this in the abstract since it highlights the huge efforts put into validation and uncertainty assessment of our dataset. These exercises are explained in more detail in their respective sections **5.2 Leave-one-out cross validation** and **5.3 Leave-block-out cross validation**

- Abstract: Estimated global glacier mass change and inferred sea level rise should appear in the abstract

Agreed. This has been added to the revised abstract

- Introduction, line 54: What is FoG? Please define (in fact it is defined latter, line 74…)

The introduction has been fully re-written. FoG is correctly defined at its first appearance.

- Introduction, line 74: Explain what is an 'hydrological' year.

Hydrological year is explained in the methods, we added a citation to the methods when hydrological year is mentioned in the introduction.

L144-148: For simplicity, throughout this work hydrological years are represented as the last year of the hydrological cycle (e.g. 1976) starting on the 1st October to 30th September in the Northern Hemisphere, and from 1st April of the previous year (e.g. 1975) to 31st March of the year (e.g. 1976) in the Southern Hemisphere. For the Low Latitudes region, we assume the hydrological year to be equal to the calendar year from 1st January to 31st December.

- Introduction, line 78: 'our methodology performs well…'. What is this methodology?

The introduction has been fully re-written. Our methodology is fully explained in the methods section 2.2.

- Introduction, line 82: 'leave-one-out cross validation'. Explain what this consists of.

The leave-one-out cross validation (and the added leave-block-out cross validation) is explained in more detail in the respective sections **5.2 Leave-one-out cross validation** and **5.3 Leave-block-out cross validation**

- - Section 2.1.1, line 91: 'to spatially locate glaciers'. Give some information on their size, number, distribution.

The number of glaciers, their total area (and the area region-by-region) is available in the manuscript in **section 2.1.1 Glacier inventories**. The most relevant information for the scope of our work is illustrated in Fig. 1.

- Section 2.1.1, lines 96-97. Explain what GLIMS consists of.

GLIMS is an initiative from the early 2000s to improve glacier inventories using satellite data (in particular from ASTER). This clarification has been added to the revised manuscript L101.

Section 2.1.1., line 108. Fog already defined

Corrected

- Section 2.1.1, line 144. '96% of all world's glaciers'. See comment above.

This information is given before in the revised text.

- Section 2.1.1, line 118: 'more details on the glaciological methods…'. This sentence should appear earlier. Moreover, it would be useful for non experts to briefly explain what it consists of.

We think it is appropriate to put these references in the data section, this level of detail is too high for an introduction. Describing the glaciological method in detail is out of the scope of this work. Still, it is properly described and referenced in the revised introduction with its strengths and limitations.

- Section 2.2, line 135. What is WGMS-id?

This is the FoG database glacier identifier. We added this clarification in the data section:

L114: Individual glaciers with available observations are identified in the FoG database with a WGMS-Id.

- Section 2.2, line 140. 'low latitudes…'

Glacier region names in RGI have capital letters.

- Figure 2. in the figure caption give the definitition of the parameters beta g, Y, beta cal,g, etc.

We prefer not to have equations in the captions. These parameters are properly defined in the text, with reference to the figure. We added in the figure's caption that "notations are defined in the text".

- Section 2.2.1, line 160. Give the definition of the variables.

Defined

- Section 2.2.1, line 168: What means a 'threshold of 8 years'?

This means that a glacier needs to have at least 8 years of glaciological in-situ observations within the 10-year-reference period to calculate their annual mass-balance anomaly. We clarified this in the text L173-181

- Section 2.2.1, lines 171 to 174: Sentence unclear.

This has been rephrased.

- Section 2.2.1: What means a 'threshold of 1000 km'?

This is no longer relevant with the change to the kriging method (see answers to reviewer #1)

- Section 2.2.1, line 180: Give the definition of the variables.

This is no longer relevant with the change to the kriging method (see answers to reviewer #1)

- Section 2.2.1, lines 191 to 193: Give the definition of the variables.

This is no longer relevant with the change to the kriging method (see answers to reviewer #1)

- Table 2: Define the acronyms CAN, SjM, ALA, SCA, CEU, SAN...

The RGI 19 first order glacier regions can be identified by a region number (1,2,3...19), a region code (ALA, WNa, CAN...ANT) and a region full name. Most of the figures have the number and long name, sometimes using the region codes make figures easier to visualize (Fig. 7, Fig. 8.1, Fig 8.2, Fig 9). All three equivalences are available in Table 5. We changed to full names in table 2 to make it clearer.

- Section 2.2.2, line 219, 224, 229: what means 'k-calibrated'

Clarified in L242-243

- Section 2.2.3, line 234. What is PoR?

PoR = Period of record, defined in L230. This needs to be defined as an acronym since it is used in the equations. We changed the appearances in the text to "period of record".

- Section 2.2.4. This section is very technical. I would suggest to move the maths to an annex and explain in the main text the followed approach. As in the previous section, many variables are not defined...

We agree this section is technical, however, we need to properly describe the methodology of our dataset for publication in a data journal like ESSD. To our knowledge there is no supplement document or annex for this journal. A non-expert can always skip this section of the article if judged too technical. Advice from the editor is welcome here.

- Section 2.2.4, line 295-296. Give a reference for C3S (https://climate.copernicus.eu/)

Added

- Table 3: The content of Table 3 should be summarized in the main text.

We disagree, we intentionally decided to put this information on a table to make comparison simpler. If this is added as text, it is likely to be hard to read and understand.

- Table 4: same comment as above.

As above, we disagree, we intentionally decided to put this information on a table to describe the datasets in a simpler way. A paragraph with all this information would be in any case very hard to read and understand.

- Section 4. The 'Results' section should appear before the section on products

We believe this section is well located after the methods section, we describe the products and where they are located. Then we proceed to analyze the results. We are also open to changing the order if the editor agrees with the reviewer.

- Section 4, line 331: how much sea level rise?

This section has been rewritten and now include the sea level rise

- Figure 3: too busy. The panel on the global mass change should appear separately.

We disagree, we think it should be part of the same figure. We modified it to make it look more similar to the regional panels.

- Table 4, line 366: Should be Table 5.

Corrected

- Section 5.2.1, line 406: Why Ba? Why such a notation for a mass change?

Defined in EQ 1 (L167) and:

L449-450: For each selected glacier, we compare the original 'reference' mass balance time series (reference Ba) as available from the FoG database, with its leave-one-out calibrated mass change time series (Leave-one-out Ba).

- Figure 7: define the acronyms appearing in the figure.

Acronym equivalences are defined in Table 5. We prefer to use acronyms in these plots.

- Table 5, line 470: Should be Table 6. The content of the table should be summarized by a few sentences in the main text.

Corrected. The most relevant part of this information is available in the discussion text of **section 5.4, Improvements with respect to earlier assessments**. We chose the Table format to add extra relevant information in a simple format.

- Figure 8: the 'global' panel should be a separate figure. There is no discussion on the interannual variability? What is its origin? Some quasi periodicity is apparent? Could you provide a spectrum of the detrended time series?

Interannual variability differences are discussed in:

L589-595: Most regions display increased interannual variabilities when compared to both previous studies. The Gaussian regression used to fit the DEM time series in Hugonnet et al. (2021) has a smoothing effect to the point where annual variability is no longer detected (Fig.10). Similarly in Zemp et al. (2019), the variance decomposition model (Eckert et al., 2011; Krzywinski and Altman, 2014) employed to extract the temporal mass change variability for each region has shown

to contribute to a slight smoothing of the annual amplitude signal (Zemp et al., 2020). Our approach allows us to better represent the interannual variability at the individual glacier level, supported by the Leave-one-out cross validation exercise and the effect is inherited to the regional and global level.

We think it is beyond the scope of this data paper to make a deeper analysis of the variability (and periodicities in the signal).

- Section 5.3.1, line 560. What means 'Hexagon corona'? Is it the name of a glacier? Where?

Clarified in the revised manuscript:

L633 "unlocking historical United States spy satellite archives (e.g. KH-9 Hexagon and Corona declassified satellite imagery)"

- Section 5.3.1, line 576: Instead of just quoting 'gravimetry', mention GRACE results and give references.

Added GRACE + references to (e.g. Blazquez et al., 2018; Chen et al., 2022)

- Conclusions, lines 619-620. A reminder of the data used and methodology would be useful. A few sentences on the novelty of the study and the main results should also be added.

The conclusion has been fully rewritten

- Conclusions, line 640. Mention to ECVs and GCOS is too vague. Explain.

The conclusion has been fully rewritten, EVS and GCOS are not cited anymore.

---

## Referee Report (RR1)

**Second round of review of *"Annual mass changes for each glacier in the world from 1976 to 2023"* by *Ines Dussaillant et al.**

Earth System Science Data: essd-2024-323

Dear Ines Dussaillant and co-authors, dear Editor,

The manuscript has improved significantly since the previous submission. I appreciate the considerable effort the authors have invested in addressing my comments and refining the methods, manuscript structure, the story telling, and data description. I would like to thank you for your patience in thoroughly reviewing and carefully considering each of my suggestions. I am glad to see that some of my proposed ideas may have contributed to making the dataset more robust and to articulating further the strengths and limitations of it. I must also apologize for any misunderstandings or oversights on my part during the first review process. I have gained valuable insights into the topic through this exchange and appreciate the opportunity to engage with this work.

I am particularly pleased with the clarified use of "geostatistical modeling," the enhancements made to the methods description, and the updated mass balance (MB) anomaly selection via kriging spatial interpolation. The addition of a leave-block-out cross-validation scheme is another commendable improvement. However, I am no expert in "kriging spatial interpolation" and can not very well judge this new methodological part of the manuscript.

Your research addresses a complex topic, and I believe this paper lays a solid foundation for further advances in this direction. At this stage, I don't ask for any major changes. Nonetheless, I have identified a few aspects that may need to be addressed before the manuscript is ready for publication. I apologize for the inconvenience, but I trust these adjustments will further improve your work.

**1 General Comments**

**1.1 Description of error bars/uncertainties**

In the response to my comment you wrote:

*"Reported uncertainties in the text correspond to $2\sigma$ = 95% confidence. Therefore, the term "uncertainty" corresponds to $1\sigma$ when describing equations and $2\sigma$ for reported values."*

I truly believe it would be important to also add that information into the manuscript. If someone wants to use the total error that you provide in the dataset or e.g. in the abstract as input for their model, how should they interpret your provided error? Is it at "one or two s.e.m (standard error of the mean)" or is this impossible to say as you don't know about the uncertainty levels of your data contributors? Clarifying this would be very important for the data users. From what I have seen the uncertainties are explained only once at the very end at the caption of Fig. 8 but not in the text or any other figure, or did I miss something?

I would suggest that you describe the meaning of the uncertainty in the manuscript's methods by saying

sth. like: "All given uncertainties in the tables, figures and main text are at the $2\sigma$ level (around 95% confidence interval)" [or, if this is true: "... 2 s.e.m."]. I am not sure if this is correct for Fig. 2, L373, Fig. 4, Fig. 10, Table 4, Table 5, Table 8? Though, I guess, Fig. 7 is in "$1\sigma$"? In that case, it may be better to describe it briefly at every caption and once in the text?

I would also add the error description to the data description table where you mention the "error". I would even mention it already once in the abstract (L27, though this may be a question of "taste", so your choice).

**1.2 Interpolating with kriging**

**L185–L215 (including Fig. 3):** This is a very interesting new way to select the glacier annual MB anomalies and to assess associated uncertainties.

I have the following comments on this new approach. Some of those are just "very minor comments", but I thought it is better to gather all "kriging-related" comments here.

- You write that the predicted kriging uncertainty grows with distance. How do the uncertainties increase with the distance? Is it somehow possible to visualise that within/beside Figure 3? Or possible to briefly explain further?

- I am no expert in kriging. I think it would be great if you can give some references to studies that use similar kriging approaches. From your code, I understood that you use "OrdinaryKriging" from the PyKrige package. I think it would be good to cite that package.

- **L196–201**: You write that the observed 5-year anomaly "Hugonnet et al. (2021) spatial correlation patterns" validate the modelled annual MB anomalies. Can you write in your paper another sentence explaining that? They both have the same pattern of a decreasing correlation over the distance (what is expected). Though, apart from that, the two look to me, as non-expert, quite different. For example, the observed 5-year anomaly spatial correlation pattern starts at much lower correlation values (maybe also expected, but something to eventually describe?), and the correlation decreases first stronger and then decreases less with the spatial lag. In comparison, the correlation of the modelled yearly MB anomalies decreases at small spatial lags only minimally, but then decrease at larger spatial lags stronger. What I want to say: the "shape of the curve" is different between the two, or not?

- Related to that, I am missing one sentence of the potential influence of using modelled glacier MB anomaly data to assess the correlations (to add here or in the discussion). If I understand it correctly, you use glacier MB anomalies from GloGEM (Huss and Hock, 2015) which is calibrated with regional geodetic MB data. Each individual glacier's specific mass balance was forced to match the average regional specific MB during the same multi-year time period. In addition, Glo-GEM's modelled interannual mass-balance variability likely depends on the chosen calibration option / calibrated precipitation factor. So I am wondering, does the way how much the precipitation factor varies from one glacier to the next influence the interannual MB variability and with that the Kriging results? I know analysing this is completely out of the scope of this study, but I

think it would be really great to mention this potential model-biased issue very briefly. Or, if you don't think it is an issue, describe why.

- **Eq. 4**: Do I understand it correctly that $\rho_{\beta,y}$ describes the y-axis of Fig. 3 (blue line). In that case, I believe, something has to be wrong with the parameters or the equation. The current equation 4 will give correlations $\rho$ with values above 0.23 for all real "d"-values. Fig. 3, however, shows, that the fitted $\rho$ reaches correlations near to zero. Or do I misunderstand here something, and Eq. 4 shows another "unit/metric" than Fig. 3?
- **Fig. 3**: I was first a bit confused about the 23 crosses for the "empirical variogram". Can you maybe add in the caption one or two words to clarify that? I first thought that the "23" corresponds to the eventually 23 used "glacier sub-periods" (but then understood that this does not make any sense). If I understand it now correctly, the crosses describe the "Average empirical variogram". If yes, consider adding "average" to the label to make it easier understandable.

**1.3 Leave-block-out cross validation**

Thanks a lot for adding that additional analysis. To make the new analysis even more useful, I suggest to consider the following aspects:

- **Table 6**: You provide the ME and S residual. Do these values come from "the yearly results" (same metric as in Fig. 7a)? Would it be easy to add also the metrics for the "Ba variability vs leave-one out BA var. STD" (Fig. 7c)? I would find that interesting to understand whether the interannual variability underestimation increases with the leave-one-block-out estimate. If I understand it correctly, the metrics presented in Figure 9 do not directly describe how the interannual MB variability changes with the leave-block-out cross validation.
- **Figure 9a, b**: I am not completely sure if I understand correctly what is represented. Subplots a, b do have a "violet" color, do they also present the "1km" threshold? I guess no and they rather represent all threshold options. Please clarify.
- **L545–547**: "S is larger than $\sigma$ only in some few cases with distances to the closest glacier >500km, b... .": I guess this estimate comes from a quantitative analysis of the data of Fig. 9d? Figure 9d looks like more than a "few" glaciers, but it is very difficult to check as the dots overlap. Can you add some kind of statistics to the end of that sentence?
- **Fig. 9d y-label**: missing ")" bracket
- **caption**: mass-changeestimate -> mass-change estimate

**2 Specific comments**

I list these points in the order of their appearance in the manuscript, rather than by their significance.

- title: I don't have a strong opinion here, and I am ok with both titles
- **L61**: mountain ranges(Brun et al –> missing "space"

- **L81**: "we use glaciological observations from approximately 500 glaciers" ... please add from how many glaciers you use the glacier MB anomaly. I know I mentioned that already in the last round and you answered that this number is visible for every region in Fig. 1. This is true, though I would really appreciate it if you add just behind this sentence in a bracket (15X glaciers used for the glacier MB anomalies...). Or do you use annual MB glacier observation data from the other around 350 glaciers? I understand L179 clearly in that way, that you use those glaciers with a "glacier MB anomaly", i.e those glaciers with 8 years of MB data within the 10-year reference period.
- **L320**. There are two dots after Zemp et al. (2019)
- **L321**: You assume that all mass change occurs above sea-level. If I understood it correctly, your dataset anyway only describes mass change above sea-level and not the subaqueous mass loss. Maybe you can clarify that in such kind of a sentence, such as : "As our dataset does not capture subaqueous mass loss, we assume that all estimated glacier mass loss occurs above the sea-level..."
- **L415–417**, **L438–439**: "We remind here that, by construction, nearby glaciers share a large fraction of the variance in mass balance variability and are thus not independent"... You write this sentence here at the beginning of the section. And repeat a similar sentence at the end of that section ".. annual mass-balance anomalies are extracted from a handful of glaciers in each region and thus , in each region, individual glaciers share a large fraction of these variabilities" (and in the conclusion). However, I still miss in this section a bit the "uncertainty/error" component. Would it be possible to add one half sentence or sentence on the end ... Something like: "The data user should carefully check the associated errors to decide if the dataset can be used for their specific use case... "
- **L450, Figure 7 caption**: You write sometimes 73 glaciers and sometimes 74 glaciers. I guess it should be 74 everywhere?
- **Fig. 8.1/8.2** There are a few glaciers and years, where the annual MB is not included within the uncertainties of the leave-one-out estimates mass-balance (e.g. Mittivakkat or Djankuat). This shows that the estimated annual MB can also be "completely" off even when considering the provided uncertainties/errors. Eventually consider mentioning that in the discussion.
- **Table 7** Huss and Hugonnet (in prep): Is this the same as the "Huss (in preparation)" somewhere else in the manuscript? If yes, use consistent naming.
- **L564**: is there a word missing? Should it be "35% smaller than the XXX predicted"?

---

## Author Response (AR2)

**Second round of review of "Annual mass changes for each glacier in the world from 1976 to 2024" by Ines Dussaillant et al.**

Earth System Science Data: essd-2024-323

We are very thankful for the second round of minor reviews. We hereby respond point by point to all specific concerns raised by both reviewers. We also profit to notify the editor and the two reviewers that we have recently ingested to the FoG database the latest annual mass balance observations for reference and benchmark glaciers during the hydrological year 2024. This means we were able to update all our results to include the year 2024 before publication. The tittle, total resulting numbers, all result figures and the discussion of the result have been updated accordingly.

**Report reviewer #2:**

This revised version has been significantly improved compared to the initial manuscript. It reads much better. Most of my comments on the initial version have been taken into account, providing the previously lacking information. It remains that sections 2.2.1, 2.2.2, and 2.2.3 are still hard to read for a non expert (like me, Reviewer 2). Several variables are still not defined. To improve understanding of the proposed methodology, I suggest that at the beginning of each of the three subsections, the authors summarize by a few sentences what they intend to do and what is the expected outcome.

I note that Reviewer 1 made substantial comments in the 'Method' section and that the authors responded amply. Thus being unable to judge the details of the calculations, I will rest on Reviewer 1's evaluation. Concerning the rest of the manuscript, I think that significant clarification has been made by the authors. Thus except for the method section, the revised manuscript looks to me now publishable.

We thank reviewer #2 for providing a wider perspective on our work and helping the revised manuscript to be better understood by a broader and non-specialized audience. The method sections 2.2.1, 2.2.2 and 2.2.3 are already introduced in the second paragraph of the methods section with a brief summary of what is done in each step and what is the main outcome. We note that these sentences were previously referring to the panels of Fig. 2 but not to the specific methodological sections to which they correspond. We now added clearer references to these specific sections, however we do not think that it is needed to repeat these summary sentences at the beginning of each section.

L151-158: Our processing algorithm is summarized in three key steps, described in the following sections and in Fig. 2. First, focusing on a specific glacier in the RGI-6.0 inventory, we estimate the detrended temporal variability of annual mass change for the glacier, referred here as the glacier mean annual mass-balance anomaly, using the interannual variability of nearby glaciological time series (Section 2.2.1 and Fig. 2a). Secondly, we calibrate the mean annual mass-balance anomaly to the long-term trends from the geodetic sample available for the respective glacier (Section 2.2.2 and Fig. 2b). Third, we integrate all these calibrated time series into a single, area-weighted average,

producing a data-fused annual mass change time series unique for every individual glacier (Section 2.2.3 and Fig. 2c). All given uncertainties in the tables, figures, main text and reported in the dataset files are at the one\_ $\sigma$  level (68% confidence interval), unless stated otherwise.

**Report reviewer #1:**

Dear Ines Dussaillant and co-authors, dear Editor,

The manuscript has improved significantly since the previous submission. I appreciate the consider able effort the authors have invested in addressing my comments and refining the methods, manuscript structure, the story telling, and data description. I would like to thank you for your patience in thoroughly reviewing and carefully considering each of my suggestions. I am glad to see that some of my proposed ideas may have contributed to making the dataset more robust and to articulating further the strengths and limitations of it. I must also apologize for any misunderstandings or oversights on my part during the first review process. I have gained valuable insights into the topic through this exchange and appreciate the opportunity to engage with this work.

I am particularly pleased with the clarified use of "geostatistical modeling," the enhancements made to the methods description, and the updated mass balance (MB) anomaly selection via kriging spatial interpolation. The addition of a leave-block-out cross-validation scheme is another commendable improvement. However, I am no expert in "kriging spatial interpolation" and can not very well judge this new methodological part of the manuscript.

Your research addresses a complex topic, and I believe this paper lays a solid foundation for further advances in this direction. At this stage, I don't ask for any major changes. Nonetheless, I have identified a few aspects that may need to be addressed before the manuscript is ready for publication. I apologize for the inconvenience, but I trust these adjustments will further improve your work.

**1** General Comments**

**1.1 Description of error bars/uncertainties**

In the response to my comment you wrote:

"Reported uncertainties in the text correspond to  $2\sigma = 95\%$  confidence. Therefore, the term "uncertainty" corresponds to  $1\sigma$  when describing equations and  $2\sigma$  for reported values."

I truly believe it would be important to also add that information into the manuscript. If someone wants to use the total error that you provide in the dataset or e.g. in the abstract as input for their model, how should they interpret your provided error? Is it at "one or two s.e.m (standard error of the mean)" or is this impossible to say as you don't know about the uncertainty levels of your data contributors? Clarifying this would be very important for the data users. From what I have seen the uncertainties are explained only once at the very end at the caption of Fig. 8 but not in the text or any other figure, or did I miss something? I would suggest that you describe the meaning of the

uncertainty in the manuscript's methods by saying 1 sth. like: "All given uncertainties in the tables, figures and main text are at the  $2\sigma$  level (around 95% confidence interval)" [or, if this is true: "... 2 s.e.m."]. I am not sure if this is correct for Fig. 2, L373, Fig. 4, Fig. 10, Table 4, Table 5, Table 8? Though, I guess, Fig. 7 is in "1 $\sigma$ "? In that case, it may be better to describe it briefly at every caption and once in the text?

It seems that there is still some confusion with the use of sometimes one  $\sigma$  and sometimes two  $\sigma$ . For consistency and to avoid further confusions on users and readers, we finally decide to keep all uncertainties at the one  $\sigma$  level across all figures, tables and reported dataset values, unless stated otherwise. Users can decide to consider their preferred confidence intervals when using the dataset. We checked all figures and tables to make sure that uncertainties are at the one  $\sigma$  level.

We added the following statement at the beginning of the method section:

L156-157: All given uncertainties in the tables, figures, main text and reported in the dataset files are at the one  $\sigma$  level (68% confidence interval), unless stated otherwise.

I would also add the error description to the data description table where you mention the "error". I would even mention it already once in the abstract (L27, though this may be a question of "taste", so your choice).

Added to the data description table at the data file names section. Now files corresponding to uncertainties show that they express the one  $\sigma$  level. We do not think it is necessary to add it in the abstract. This has been properly clarified in the method section and across the manuscript.

**1.2 Interpolating with kriging**

**L185–L215 (including Fig. 3):** This is a very interesting new way to select the glacier annual MB anoma lies and to assess associated uncertainties.

I have the following comments on this new approach. Some of those are just "very minor comments", but I thought it is better to gather all "kriging-related" comments here.

• You write that the predicted kriging uncertainty grows with distance. How do the uncertainties increase with the distance? Is it somehow possible to visualise that within/beside Figure 3? Or possible to briefly explain further?

How uncertainties increase with distance was already described in the text:

L192-194: "the predicted kriging uncertainty  $\sigma_{\overline{\beta}_{g,Y}}$  grows with distance, from the measurement error of the inputs  $\sigma_{B_a}$  at close distances from a measured glacier, to the signal variability (spread of  $\beta_Y$ ) at distances far away from any measured glacier, where the prediction is more poorly constrained." Further explanations of the method have been added in the revised manuscript as described in the following answers.

• I am no expert in kriging. I think it would be great if you can give some references to studies

that use similar kriging approaches. From your code, I understood that you use "OrdinaryKriging" from the PyKrige package. I think it would be good to cite that package.

We have added the following statements to the text regarding kriging:

L195-203: Kriging is a core method of spatial statistics (Cressie, 1993), often coined 'best linear unbiased interpolator' due to its non-parametric nature and empirical variance minimization. It emerged in mining applications (Matheron, 1965), and has since become ubiquitous for spatial interpolation across many fields (Webster and Oliver, 2007). In glaciology, kriging has been for instance used to spatially interpolate sparse ablation measurements (Hock and Jensen, 1999) or ice thickness measurements (Fischer, 2009). Recently, the rise of machine learning methods has extended kriging concept to any kind of dimension through Gaussian Processes (Rasmussen and Williams, 2006), which have also found applications in glaciology, from remote sensing time series interpolation (Hugonnet et al., 2021) to model error emulation (Edwards et al., 2021).

**L190: We have added a citation to PyKrige.**

• L196–201: You write that the observed 5-year anomaly "Hugonnet et al. (2021) spatial correla tion patterns" validate the modelled annual MB anomalies. Can you write in your paper another sentence explaining that? They both have the same pattern of a decreasing correlation over the distance (what is expected). Though, apart from that, the two look to me, as non-expert, quite different. For example, the observed 5-year anomaly spatial correlation pattern starts at much lower correlation values (maybe also expected, but something to eventually describe?), and the correlation decreases first stronger and then decreases less with the spatial lag. In comparison, the correlation of the modelled yearly MB anomalies decreases at small spatial lags only mini mally, but then decrease at larger spatial lags stronger. What I want to say: the "shape of the curve" is different between the two, or not?

We agree with the referee that the validation of annual modelled anomalies using 5-year observed ones was not sufficiently explained. We have updated the statement as follow::

L204-2011: In order to estimate the spatial correlation of the annual mass-balance anomaly  $\rho_{-}(\beta, Y)$ (d) to constrain the kriging, we sampled empirical variograms for both local-scale modelled annual mass balance anomalies (Huss and Hock, 2015) and for observational 5-year anomalies (Hugonnet et al., 2021), the latter validating the spatial correlation patterns observed in the modeled estimates (Fig. 3). The 5-year anomalies are used only to validate annual anomalies. As climatic patterns driving correlations in regional anomalies should have a size that is largely consistent in time, we expect 5-year anomalies to be spatially correlated at similar distances than annual anomalies but with a lesser amplitude due to the cancelling of positive and negative anomalies over time. We indeed identify that both anomalies have significant spatial correlation up to 5000 km, with a smaller amplitude for 5-year anomalies (Fig. 3).

Additionally, to address the referee's comment about the form of the correlation function and the

sensitivity to modelling uncertainties (further detailed in next comment), we add the following statement:

L228-235: We note that, because kriging is a non-parametric interpolation method, its prediction primarily depends on the observations themselves, so uncertainties in the correlation function stemming from the modelled estimates of Huss and Hock (2015) have little influence on our results. Furthermore, because our correlations span multiple orders of magnitudes (from 10 km to 5000 km), the choice of functional form of the correlation has been shown to have minimal impact on the prediction (Hugonnet et al., 2022). To exemplify this, we compared kriging with inverse-distance weighting, a different interpolation method altogether, and found almost equal regional estimates as those are primarily driven by the input data. Differences between kriging and inverse-distance weighting only showed at the glacier-scale, where kriging allows to further refine anomalies and derive empirical uncertainties."

Related to that, I am missing one sentence of the potential influence of using modelled glacier MB anomaly data to assess the correlations (to add here or in the discussion). If I understand it correctly, you use glacier MB anomalies from GloGEM (Huss and Hock, 2015) which is calibrated with regional geodetic MB data. Each individual glacier's specific mass balance was forced to match the average regional specific MB during the same multi-year time period. In addition, Glo GEM's modelled interannual mass-balance variability likely depends on the chosen calibration option / calibrated precipitation factor. So I am wondering, does the way how much the precipi tation factor varies from one glacier to the next influence the interannual MB variability and with that the Kriging results? I know analysing this is completely out of the scope of this study, but I think it would be really great to mention this potential model-biased issue very briefly. Or, if you don't think it is an issue, describe why.

**See the previous answer.**

Eq. 4: Do I understand it correctly that ρβ,y describes the y-axis of Fig. 3 (blue line). In that case, I believe, something has to be wrong with the parameters or the equation. The current equation 4 will give correlations ρ with values above 0.23 for all real "d"-values. Fig. 3, however, shows, that the fitted ρ reaches correlations near to zero. Or do I misunderstand here something, and Eq. 4 shows another "unit/metric" than Fig. 3?

Good catch from the referee. This was an old version of the equation, not properly converted from variogram function to correlation function. The correct correlation function does not require a nugget n, and reads:

$$\rho_{\beta,Y}(d) = s_1 e^{-\frac{3d}{r_1}} + s_2 e^{-\frac{3d}{r_2}} if d > 0, else 1$$

where *d* is the distance between two glaciers, s1 = 0.37, s2 = 0.59 are the partial sills and r1 = 200 *km* and r2 = 5000 *km* are the correlation ranges

• Fig. 3: I was first a bit confused about the 23 crosses for the "empirical variogram". Can you

maybe add in the caption one or two words to clarify that? I first thought that the "23" corresponds to the eventually 23 used "glacier sub-periods" (but then understood that this does not make any sense). If I understand it now correctly, the crosses describe the "Average empirical variogram". If yes, consider adding "average" to the label to make it easier understandable.

The confusion must have come from the fact that empirical variogram and correlation were used interchangeably. We have modified the legend and caption to consistently use "correlation" everywhere for describing the graph.

**1.3 Leave-block-out cross validation**

Thanks a lot for adding that additional analysis. To make the new analysis even more useful, I suggest to consider the following aspects:

• **Table 6**: You provide the ME and S residual. Do these values come from "the yearly results" (same metric as in Fig. 7a)? Would it be easy to add also the metrics for the "Ba variability vs leave-one out BA var. STD" (Fig. 7c)? I would find that interesting to understand whether the interannual variability underestimation increases with the leave-one-block-out estimate. If I understand it correctly, the metrics presented in Figure 9 do not directly describe how the interannual MB variability changes with the leave-block-out cross validation.

Yes, this is possible. We added the variability residuals of the leave-block-out experiment to a revised Table 6.

• Figure 9a, b: I am not completely sure if I understand correctly what is represented. Subplots a, b do have a "violet" color, do they also present the "1km" threshold? I guess no and they rather represent all threshold options. Please clarify.

We color-coded the results in panels 9a and 9b to avoid confusion.

L545–547: "S is larger than σ only in some few cases with distances to the closest glacier >500km, b....": I guess this estimate comes from a quantitative analysis of the data of Fig. 9d? Figure 9d looks like more than a "few" glaciers, but it is very difficult to check as the dots overlap. Can you add some kind of statistics to the end of that sentence?

We changed the statement to:

L 572: On average S starts to become larger than  $\sigma$  with distances to the closest glacier larger than 500km, but the large spread suggests this is coming from the randomness of the predictions.

• Fig. 9d y-label: missing ")" bracket

Corrected

• **caption**: mass-change estimate -> mass-change estimate

Corrected

**2 Specific comments**

I list these points in the order of their appearance in the manuscript, rather than by their significance.

• title: I don't have a strong opinion here, and I am ok with both titles

The editor has chosen to keep: Annual mass change of the world's glaciers from 1976 to 2023 by temporal downscaling of satellite data with in-situ observations

**• L61: mountain ranges(Brun et al -> missing "space"**

**Corrected**

• L81: "we use glaciological observations from approximately 500 glaciers" ... please add from how many glaciers you use the glacier MB anomaly. I know I mentioned that already in the last round and you answered that this number is visible for every region in Fig. 1. This is true, though I would really appreciate it if you add just behind this sentence in a bracket (15X glaciers used for the glacier MB anomalies...). Or do you use annual MB glacier observation data from the other around 350 glaciers? I understand L179 clearly in that way, that you use those glaciers with a "glacier MB anomaly", i.e those glaciers with 8 years of MB data within the 10-year reference period.

We added a statement of the total number of glaciological glaciers used in the method section, just after mentioning the rules of selection. We think the introduction is too early to talk about glacier anomalies since we explain that later, therefore there we use the number for the total amount of observation ingested to the processing as input data which is about 500.

L180: We use a total of 158 individual glacier anomalies for the assessment.

• L320. There are two dots after Zemp et al. (2019) Corrected

• L321: You assume that all mass change occurs above sea-level. If I understood it correctly, your dataset anyway only describes mass change above sea-level and not the subaqueous mass loss. Maybe you can clarify that in such kind of a sentence, such as : "As our dataset does not capture subaqueous mass loss, we assume that all estimated glacier mass loss occurs above the sea level..."

This information was added in Table 9 for this study + Hugonnet et al., 2021 and Zemp et al., 2019

• L415–417, L438–439: "We remind here that, by construction, nearby glaciers share a large frac tion of the variance in mass balance variability and are thus not independent"... You write this sentence here at the beginning of the section. And repeat a similar sentence at the end of that section ".. annual mass-balance anomalies are extracted from a handful of glaciers in each region and thus , in each region, individual glaciers share a large fraction of these

variabilities" (and in the conclusion). However, I still miss in this section a bit the "uncertainty/error" component. Would it be possible to add one half sentence or sentence on the end ... Something like: "The data user should carefully check the associated errors to decide if the dataset can be used for their specific use case..."

We think adding a phrase like this one is redundant, it stands more as a recommendation for users than a description or analysis of the dataset itself. We manifest already multiple times and in a transparent way that this aspect is the main limitation of our dataset, and it is also apparent in the large uncertainties at the individual glacier time series. We instead added:

L164: This limitation is, however, well evidenced by large uncertainties on under sampled regions and periods.

• **L450, Figure 7 caption**: You write sometimes 73 glaciers and sometimes 74 glaciers. I guess it should be 74 everywhere?

Corrected, it should be 74 everywhere

• **Fig. 8.1/8.2** There are a few glaciers and years, where the annual MB is not included within the uncertainties of the leave-one-out estimates mass-balance (e.g. Mittivakkat or Djankuat). This shows that the estimated annual MB can also be "completely" off even when considering the provided uncertainties/errors. Eventually consider mentioning that in the discussion.

**We added to the discussion:**

L510-521: Still large differences may occur between reference and predicted values for individual years (outliers in Fig. 7a), also evident on the few annual values where the reference annual mass balance is not included within the uncertainties of the leave-one-out annual estimates (e.g. Mittivakkat, Fig. 8j and Djankuat, Fig. 8n).

• **Table 7** Huss and Hugonnet (in prep): Is this the same as the "Huss (in preparation)" somewhere else in the manuscript? If yes, use consistent naming.

Done and made consistent throughout the text: density propagation based on Hugonnet et al, (2022), and Huss et al. (in prep.)

**L564**: is there a word missing? Should it be "35% smaller than the XXX predicted"? Corrected

---

## Author Response (AR3)

**Annual mass change of the world's glaciers from 1976 to 2024 by temporal downscaling of satellite data with in-situ observations**

**by Ines Dussaillant et al.**

Earth System Science Data: essd-2024-323

Dear editor,

We are very pleased to hear our manuscript has been accepted for publication in ESSD. We are grateful to both the editor and the two anonymous reviewers for their engagement in reviewing this work and all the valuable insights that allowed us to improve the quality of the dataset and manuscript.

In this last version of the manuscript, we have addressed the following few extra points:

- We clean up everywhere in the MS with reference to temporary review links keeping only the publication links and DOI for the final dataset and codes.
- We checked for typos in the text.
- We make sure that all figures are generated at 600 DPI. Figures are larger than their expected final dimensions to avoid quality loss. It is possible to make them at the exact dimensions when we know the size that each figure will have in the final publication.
- To avoid confusion, we decided it is best to remove the citation to the unpublished work from Huss et al. (in prep), and instead briefly explained the methodology for the spatial correlation in density conversion error. For this we added in the revised manuscript:

L323-331: For density conversion, we estimated a spatial correlation function of the uncertainty in the density conversion  $\rho_{\sigma_{f_{\rho}}}(d)$  by performing a similar variogram analysis as detailed for annual anomalies (Section 2.2.1) but instead applied to modelled estimates of annual density of volume change for all glaciers globally. These estimates were obtained by pairing a mass balance model (Huss and Hock, 2015) with a firn densification model (Huss, 2013), calibrated on geodetic mass balances (Hugonnet et al., 2021). We find a spatial correlation function of:

$$\rho_{\sigma_{f_{\rho}}}(d) = s_1 e^{-\frac{3d}{r_1}} + s_2 e^{-\frac{3d}{r_2}} \text{ if } d > 0, \text{ else } 1 \tag{15}$$

Where d is the distance between two glaciers,  $s_1 = 0.12$ ,  $s_2 = 0.72$  are the partial sills and  $r_1 = 200 \text{ km}$  and  $r_2 = 5000 \text{ km}$  are the correlation ranges.

And we changed the citation of Huss et al. (in prep) in Table 3 and Table 7 to: *"Spatial correlation following an empirical function in density conversion error* ( $\rho_{\sigma_{f_o}}(d)$ , EQ 15)"